# Form and function in hillslope hydrology: characterization of subsurface flow based on response observations

Lisa Angermann[1,2], Conrad Jackisch[3], Niklas Allroggen[2], Matthias Sprenger[4,5], Erwin Zehe[3], Jens Tronicke[2], Markus Weiler[4], and Theresa Blume[1]

[1]Helmholtz Centre Potsdam, GFZ German Research Centre for Geosciences, Section Hydrology, Potsdam, Germany
[2]University of Potsdam, Institute of Earth and Environmental Science, Potsdam, Germany
[3]Karlsruhe Institute of Technology (KIT), Institute for Water and River Basin Management, Chair of Hydrology, Karlsruhe, Germany
[4]University of Freiburg, Institute of Geo- and Environmental Natural Sciences, Chair of Hydrology, Freiburg, Germany
[5]University of Aberdeen, School of Geosciences, Geography & Environment, Aberdeen, Scotland, UK

*Correspondence to:* Lisa Angermann (science@lisa-angermann.de)

**Abstract.** The term *form and function* was established in architecture and biology and refers to the idea, that form and functionality are closely correlated, influence each other and co-evolve. We suggest to transfer this idea to hydrological systems to separate and analyze their two main characteristics: Their form, which is equivalent to the spatial structure and static properties, and their function, equivalent to internal responses and hydrological behavior. While this approach is not particularly new to hydrological field research, we want to employ this concept to explicitly pursue the question of what information is most advantageous to understand a hydrological system. We applied this concept to subsurface flow within a hillslope, with a methodological focus on function: We conducted observations during a natural storm event and followed this by a hillslope-scale irrigation experiment. The results are used to infer hydrological processes of the monitored system. Based on these findings, the explanatory power and conclusiveness of the data is discussed. The measurements included basic hydrological monitoring methods, like piezometers, soil moisture and discharge measurements. These were accompanied by isotope sampling and a novel application of 2D time-lapse GPR (ground-penetrating radar). The main finding regarding the processes in the hillslope was, that preferential flow paths established fast, despite unsaturated conditions. These flow paths also caused a detectable signal in the catchment response following a natural rainfall event, showing that these processes are relevant also at the catchment-scale. Thus, we conclude that response observations (dynamics and patterns, i.e. indicators of function) were well suited to describe processes at the observational scale. Especially the use of 2D time-lapse GPR measurements, providing detailed subsurface response patterns, as well as the combination of a stream-centered and a hillslope-centered approach allowed us to link processes and put them in a larger context. Transfer to other scales beyond observational scale and generalizations, however, rely on the knowledge of structures (form) and remain speculative. The complementary approach with a methodological focus on form (i.e. structure exploration) is presented and discussed in the companion paper by Jackisch et al. (this issue).

# 1 Introduction

Characterizing subsurface flow is the aim of many hydrological field and modeling studies. In hillslopes with steep slopes and structured soils, subsurface flow is controlled by high gradients and high heterogeneity of hydraulic properties of the soil, resulting in a highly heterogeneous flow field and preferential flow paths (e.g. Scaini et al., 2017). The specific challenge of investigating preferential flow lies in its manifestation across scales, its high spatial variability, and pronounced temporal dynamics. A considerable number of experimental and model approaches have been proposed to investigate the issue (Beven and Germann, 1982; Šimůnek et al., 2003; Gerke, 2006; Weiler and McDonnell, 2007; Köhne et al., 2009; Beven and Germann, 2013; Germann, 2014). However, rapid flow in structured soils is still a challenge to current means of observation, process understanding and modeling.

In previous studies at the hillslope-scale, the focus was often on lateral flow processes and the establishment of overall connectivity. Hillslope-scale excavations yield information on spatial extent and characteristics of preferential flow paths in 3D (Anderson et al., 2009; Graham et al., 2010), but are highly destructive and lack the temporal component. Hillslope-scale tracer experiments in contrast, resolve temporal dynamics and velocities (Wienhofer et al., 2009; McGuire and McDonnell, 2010), but lack the spatial information. Hillslope-scale experiments are usually very labor intensive and require high technical effort and most studies are concentrated on well-monitored trenches (McGlynn et al., 2002; Tromp-Van Meerveld and McDonnell, 2006; Vogel et al., 2010; Zhao et al., 2013; Bachmair and Weiler, 2012). Blume and van Meerveld (2015) give a thorough review of investigation techniques for subsurface connectivity and find experimental studies on this topic underrepresented in hydrological field research.

In recent years, a trend towards non-invasive methods for hillslope-scale observations emerged (Gerke et al., 2010), which has been an important improvement with regard to repeatability and spatial and temporal flexibility of observations (Beven and Germann, 2013). In this context various geophysical methods have been applied for subsurface exploration (e.g. Wenninger et al., 2008; Garré et al., 2013; Hübner et al., 2015). From all applied geophysical techniques ground-penetrating radar (GPR) is known as the tool providing the highest spatial and temporal resolution. GPR provides information on subsurface structures at minimal invasive cost (e.g. Lambot et al., 2008; Bradford et al., 2009; Jol, 2009; Schmelzbach et al., 2011, 2012; Steelman et al., 2012). Its short measurement times and high sensitivity towards soil moisture predestine GPR for monitoring subsurface flow processes. Nevertheless, only few field studies exist which have successfully applied surface based GPR for the investigation of preferential flow paths or subsurface flow in general (Truss et al., 2007; Haarder et al., 2011; Guo et al., 2014; Allroggen et al., 2015b). Previous GPR monitoring studies rely on two different principles. The first approach relies on interpreting selected reflection surfaces and comparing this interpretation result between the individual recorded GPR surveys (Truss et al., 2007; Haarder et al., 2011). The result is a shift in GPR signal travel time, which can be interpreted in terms of soil moisture changes, using a petrophysical relation (e.g. CRIM model, Allroggen et al., 2015b). The second approach relies on calculating difference images between individual GPR surveys (e.g. Birken and Versteeg, 2000; Trinks et al., 2001; Guo et al., 2014; Allroggen and Tronicke, 2015) and thereby highlighting areas of increased changes in the subsurface. Due to the usually high noise level of

field data, such difference calculations are critical and require sophisticated processing techniques (Guo et al., 2014; Allroggen and Tronicke, 2015).

Especially in structured soils, where subsurface flow is likely dominated by preferential flow paths, methods are required which are capable of covering the existing heterogeneity. Point measurements and integrated observations alone are barely able to meet this requirement. Structural changes of the subsurface as revealed by difference images obtained from GPR measurements, in contrast, reveal spatially discrete flow paths. We therefore applied and tested time-lapse GPR measurements to investigate subsurface flow processes within a hillslope with shallow and highly structured soils.

We chose a combination of conventional hydrological methods and non-invasive GPR measurements to explore flow processes by means of observations at the hillslope (hillslope-centered approach according to Blume and van Meerveld, 2015). This hillslope-centered approach was supported by stream-centered process observations, including a basic hydrograph analysis and surface water stable isotope sampling during the natural rainfall event. Besides the 2D time-lapse GPR measurements, the hydrological methods at the hillslope include surface runoff collectors, a dense network of soil moisture observation profiles, stable isotope samples and piezometers.

All methods and experimental results were subsumed under the framework of form and function as shown in Fig. 1. This framework was developed to analyze the explanatory power of the different observations. The idea of the form and function dualism was established in architecture (*form follows function*, Sullivan, 1896) and is commonly used in biology (e.g. Thompson et al., 1942) and describes the link, mutual influence and co-evolution of the outer appearance and functional purpose of a (research) object.

In our case, form includes all static properties and spatial structures, such as topography, geology and subsurface structures, but also porosity, hydraulic conductivity and stone content of the soil. Function summarizes all dynamics and processes, including soil moisture dynamics, discharge behavior and preferential flow. These two are closely related and co-evolve. Based on this idea, Sivapalan (2005) suggested, that patterns, responses and functions are the basic key to understanding and describing a hydrological system, as they incorporate the morphogenetic processes that led to the spatial structures. While this approach refers to the larger scale and the development of a general theory, our aim is to apply the form-function-framework to observations at the local scale.

Starting at the left side of the spectrum presented in Fig. 1, we focus on the observation of response dynamics and response patterns. The potential of the methods for the investigation of subsurface flow processes at the hillslope-scale and the characterization of typical runoff generation mechanisms are discussed and possible further improvements suggested. Based on these findings, the informative power and conclusiveness of the data will be discussed. To complement the functional perspective on the investigation of subsurface flow, the companion paper by Jackisch et al. (this issue) concentrates on the spatial characteristics of subsurface flow from the point- to hillslope-scale with a specific focus on subsurface structures.

Following the form and function framework, the hypotheses focus on the potential of response observations for hillslope hydrological field research and the application of time-lapse GPR measurements in this context:

**H1** Response observations (discharge, TDR and GPR data) are sufficient to characterize subsurface flow within the hillslope. (function described without form)

**H2** Response patterns can be used to deduce flow-relevant structures in the subsurface. (function reveals form)

**H3** Time-lapse GPR measurements visualize subsurface flow dynamics and patterns and can replace hillslope trenches.

## 2 Methods

### 2.1 Study site

The investigated area is located at the south eastern edge of the Ardennes Massif in western Luxembourg. It consists of a number of nested sub-catchments of the Colpach River catchment, which is part of the Attert River basin. The landscape of this area is characterized by Devonian schist bedrock (Colbach and Maquil, 2003). The soils are young and composed of eolian loess deposits and weathered schist debris. Under periglacial conditions, the weathered rocks were relocated by solifluction, causing an often horizontal or slope parallel orientation of the saprolite (Juilleret et al., 2011). The periglacial deposit layer (basal layer) is overlain by shallow top soil (upper layer). The soil is classified as Haplic Cambisol (CM, IUSS Working Group WRB, 2006). Saturated hydraulic conductivity of the soil was found to be highly heterogeneous, exceeding the measuring range of the capacity of the constant head permeameter of $10^{-8}\,\mathrm{m\,s^{-1}}$ to $10^{-3}\,\mathrm{m\,s^{-1}}$. While depth profiles of hydraulic conductivity measured in the area did not show a specific pattern of conductive layers, measurements at the investigated hillslope indicated higher conductivity in the depth of $0.7\,\mathrm{m}$ (Jackisch et al., this issue).

The schist bedrock below is strongly inclined with almost vertical foliation and considered impermeable but with fractures which can function as a complex flow network with local storage in the rock cracks when saturated (Van den Bos et al., 2006; Kavetski et al., 2011). The subsurface structures are of predominantly geogenic origin and considered temporally persistent.

Within this landscape, a typical hillslope consists of agriculturally used elevated plateaus and forested valleys with steep slopes (15° to 25°). While the headwater catchments of the investigated area are usually narrow with marginal floodplains, the main Colpach River network is characterized by wider valleys with more pronounced floodplains.

The average annual precipitation between 2011 and 2014 was $965\,\mathrm{mm}$, the annual average air temperature was $8.8\,^{\circ}\mathrm{C}$. These data stem from a meteorological station from ASTA (*administration des services techniques de l'agriculture de Luxembourg*) close to Roodt, approximately $2\,\mathrm{km}$ from the experimental site.

The experimental work conducted in the frame of this study focused on a north facing hillslope in the Holtz headwater catchment. The experiment was supplemented by hydrological data from five neighboring headwater catchments of different size. All sub-catchments as well as the location of the irrigation site are shown in Fig. 2.

### 2.2 Experimental approaches

The experimental approach consists of two parts. The hillslope-centered approach concentrates on local observations at the hillslope. It includes soil moisture profile measurements and 2D GPR measurements, pore water and piezometer isotope data, and measurements of surface runoff. These data were collected during a natural summer storm event on June 20, 2013 and a hillslope-scale irrigation experiment one day later on June 21, 2013.

The stream-centered approach focuses on the discharge response and stream water stable isotope signal during the same summer storm event as mentioned above. The stream-centered approach focuses on the integrated response of a catchment. While hydrographs and stream tracer dynamics have been studied and discussed extensively elsewhere (Wrede et al., 2015; Martínez-Carreras et al., 2016, in the same area), we wanted to use these data to position our hillslope observation in the bigger picture of the catchment scale dynamics. An overview over approaches, methods and their foci is given in Fig. 3.

## 2.3 Stream-centered approach

The hydrological response behavior of several nested sub-catchments was investigated. At four locations v-notch or trapezoidal gauges were installed and equipped with pressure transducers, measuring water level, electric conductivity and temperature (CTD sensors, Decagon Devices Inc.). Water levels were measured every $15\,\mathrm{min}$. Precipitation was monitored with tipping buckets (Davis Instruments Corp.) in the Holtz 1 headwater. All data were logged with CR1000 data loggers (Campbell Scientific Inc.).

At the same locations and additionally close to the source of the Holtz River (Holtz 1 in Fig. 2), water samples were taken with auto samplers (ISCO 3700, Teledyne). The bottles of the auto samplers were pre-filled with styrofoam beads to avoid evaporation from the sample bottles. Samples were then transferred to glass bottles and analyzed in the laboratory at the Chair of Hydrology, University of Freiburg. The isotopic composition ($\delta^{18}O$ and $\delta^{2}H$) of the water samples was measured by wavelength-scanned cavity ring-down spectrometry (Picarro L2120-iWS-CRDS). The results are given in the $\delta$-notation in [‰], describing the deviation of the ratio between heavy and light isotopes ($^{2}H/^{1}H$ and $^{18}O/^{16}O$) relative to the ratio of the Vienna Standard Mean Ocean Water (VSMOW). For liquid analysis the accuracy is given as $0.1\,‰$ for $\delta\,^{18}O$ and $0.5\,‰$ for $\delta\,^{2}H$ (according to the manufacturer).

In addition to the stream water, rainfall water was sampled. Bulk samples were collected during the rainfall events right next to the experimental site. The water from the saturated zone was manually sampled on a monthly basis over the course of one year from piezometers close to the sub-catchment gauges. Samples were taken with a peristaltic pump from fresh water flowing into the piezometers, after they had been pumped empty (Fig. 2).

To calculate the event water contribution, we applied a simple hydrograph separation (Pearce et al., 1986). Equation 1 shows the calculation of the discharge attributed to the natural rain event $Q_e$, based on the isotopic composition of the base flow ($c_b$) three days before the storm event, the river water during and after the event ($c_t$), and the rain water ($c_e$) (Leibundgut et al., 2011). $Q_t$ is the total discharge during and after the event, which is constituted of the two components $Q_e$ and $Q_b$, i.e. event water and pre-event water.

$$Q_e = Q_t \cdot \frac{c_t - c_b}{c_e - c_b} \tag{1}$$

## 2.4 Hillslope-centered approach

### 2.4.1 Irrigation setup

The plot of the hillslope-scale irrigation experiment was located on the bottom $8\%$ to $13\%$ of the $238\,\mathrm{m}$ long investigated hillslope, which was defined by a slope of more than $6°$, excluding the plateaus. The plot had a slope of $\sim14°$. While vegetation at the hillsope is dominated by beech forest (*fagus sylvatica*) of mixed age, the irrigation plot is placed in an area with no major trees. Except for few young trees with breast height diameters below $0.1\,\mathrm{m}$ in the downhill monitoring area, all shrubs were cut to facilitate GPR measurements and allow for uniform irrigation. The entire investigated hillslope section covers an area of approximately $260\,\mathrm{m}^2$.

Four circular irrigation sprinklers (Wobbler, Senninger Irrigation Inc.) were arranged in a $5\,\mathrm{m}$ by $5\,\mathrm{m}$ square in the upper part of the experimental site (Fig. 4). The sprinklers had a nominated sprinkler radius of $4\,\mathrm{m}$ and were installed approximately $0.7\,\mathrm{m}$ (two uphill sprinklers) and $1.5\,\mathrm{m}$ (two downhill sprinklers) above ground surface. The level difference between the uphill and downhill sprinklers was $0.5\,\mathrm{m}$. The $25\,\mathrm{m}^2$ area spanned by the four sprinklers is referred to as the core area with homogeneous irrigation intensity of $\sim30.8\,\mathrm{mm\,h^{-1}}$ over 4:35 h. Total water input at the core area was $141\,\mathrm{mm}$. These settings aimed at activating all potential flow paths and were chosen on basis of an *a priori* simulation of the experiment (see appendix of Jackisch et al., this issue). Transfering this amount of water from the irrigated core area ($5\,\mathrm{m}$ hillslope parallel length) to the entire hillslope uphill of the rain shield ($219\,\mathrm{m}$), this intensity compares to a rain event of $3.2\,\mathrm{mm}$ precipitation. While this irrigation settings do not mimic natural conditions, this relation allows us to compare and evaluate observations under experimental and natural conditions regarding lateral subsurface flow.

The surrounding area functioned as buffer of about $4\,\mathrm{m}$ with less intense irrigation, thus mitigating boundary effects. A rain shield defined the lower boundary of the core area as a sharp transition to the non-irrigated area below. The water from the rain shield was collected with a gutter and routed away from the investigated area. The overall irrigation area (including core area and buffer) covered $\sim120\,\mathrm{m}^2$.

To monitor the irrigation we used a flow meter at the main water supply of the irrigation system to measure the absolute water input. Furthermore, one tipping bucket was used to quantify the temporal variability of applied irrigation, and 42 mini rain collectors, evenly distributed across the core area, covered the spatial distribution of the irrigation amount. The topography of the experimental site, as well as all devices and installations were mapped with a total station (Leica Geosystems AG).

The experiment took place on June 21, 2013. After one week of dry weather, two natural rainfall events of $20.2\,\mathrm{mm}$ and $21.2\,\mathrm{mm}$ had occurred on June 20. The first one had a mean intensity of $2.9\,\mathrm{mm\,h^{-1}}$ and ended 29:33 h before the irrigation experiment; the second rainfall event had a mean intensity of $9.0\,\mathrm{mm\,h^{-1}}$ and ended 19:22 h before the experiment.

### 2.4.2 Process monitoring

The monitoring of hydrological processes during and after the irrigation period was accomplished with a combination of methods: a dense array of soil moisture profiles for time domain reflectrometric (TDR) measurements arranged as diverting

transects along the slope line, and four GPR transects located downhill of the core area and oriented parallel to the contour lines and the rain shield for time-lapsed GPR measurements. The latter yielded vertical cross sections of the subsurface.

A surface runoff collector was installed across $2\,\mathrm{m}$ at the lower boundary of the core area. Surface runoff was collected by a plastic sheet installed approximately $1\,\mathrm{cm}$ below the interface between litter layer and the Ah horizon of the soil profile and routed to a tipping bucket.

An array of 16 access tubes for manual soil moisture measurement with TDR probes (Pico IPH, IMKO GmbH) covered the depth down to $1.7\,\mathrm{m}$ below ground. The layout consisted of three diverging transects with four TDR profiles in the lower half of the core area, the highest density of profiles just downslope of the rain shield, and the furthest profile about $9\,\mathrm{m}$ downhill (Fig. 4B). This setup allows for the separate observation of predominantly vertical flow at the core area and lateral flow processes at the downhill monitoring area.

Soil moisture was measured manually. To increase temporal resolution of the measurements, three probes were used in parallel. While these probes were identical with regard to measuring technique and manufacturing, they differed slightly in sensor design: two TDR probes had an integration depth (i.e. sensor head length) of $0.12\,\mathrm{m}$, one probe had an integration depth of $0.18\,\mathrm{m}$. These sensors were manually lowered to different depths into the 16 access tubes, where they measured the dielectric permittivity of the surrounding soil in the time domain through the access tubes. Given a mean penetration depth of $5.5\,\mathrm{cm}$ and a tube diameter of $4.2\,\mathrm{cm}$ this yields an integration volume of $\sim0.72\,\mathrm{L}$ and $1.05\,\mathrm{L}$, respectively. The manual measurements were conducted in $0.1\,\mathrm{m}$ depth increments and followed a flexible measuring routine with regard to the sequence in which the access tubes were measured. Thus, active profiles were covered with higher frequency.

In addition to the hydrological methods, GPR was used to monitor the shallow subsurface. 2D time-lapse GPR measurements were conducted along four transects across the downhill monitoring area. The transects had distances of $\sim2\,\mathrm{m}$, $3\,\mathrm{m}$, $5\,\mathrm{m}$ and $7\,\mathrm{m}$ to the lower boundary of the core area and were arranged approximately perpendicular to the topographic gradient. Each transect was measured nine times. One measurement was taken before irrigation started and the last one about 24:00 h after irrigation start.

The GPR system consisted of a pulseEKKO PRO acquisition unit (Sensors and Software Inc.) equipped with shielded $250\,\mathrm{MHz}$ antennas. The data were recorded using a constant offset of $0.38\,\mathrm{m}$, a sampling interval of $0.2\,\mathrm{ns}$ and a time-window of $250\,\mathrm{ns}$. For accurate positioning, a kinematic survey strategy was employed. The positioning was based on a self-tracking total station (Leica Geosystems AG), which recorded the antenna coordinates as described by Boeniger and Tronicke (2010). To guarantee the repeatability of the 2D time-lapse GPR measurements, all four transects were defined by wooden guides for an exact repositioning of the antennas. The measurement of one transect took approximately $2\,\mathrm{min}$ and measurements of all four transects were taken every $40\,\mathrm{min}$ to $120\,\mathrm{min}$ during and after irrigation as well as 18:00 h and 24:00 h after irrigation start.

### 2.4.3 Isotope sampling

The stable isotope sampling included samples taken from five soil cores (pore water), piezometers (percolating pore water), as well as irrigation and rain water (input water). The soil cores were taken with a percussion drill with a head diameter of $7\,\mathrm{cm}$,

and split into $5\,cm$ increments to get depth profiles of the stable isotopic composition ($\delta^{18}O$ and $\delta^2H$) of the pore water. Two profiles were taken before the rainfall events, one after the first minor rain event on June 20th, and two more after the irrigation experiment (at the core area and the downhill monitoring area). All profiles covered a depth of $\sim 1.7\,m$ below ground.

At the locations of the pre-irrigation soil cores, piezometers were installed. Additionally, three more piezometers were installed in a depth of $\sim 1.0\,m$. This depth was chosen based on observations in the core samples, which showed wet areas in the depths between $0.8\,m$ and $1.2\,m$, right above the $C_v$ horizon. All piezometers consist of PVC tubes of $5\,cm$ diameter and were screened at the bottom $20\,cm$. They were equipped with pressure transducers (CTD sensors, Decagon Devices Inc.). As only a few $mL$ of water were seeping into the piezometers, water tables could not be properly monitored and the data will not be shown. However, the water could be sampled, using a peristaltic pump. In addition to the pore water and piezometer samples, bulk samples of rainfall water were collected during the rainfall events prior to the irrigation experiment and directly next to the irrigation plot. Water samples were also taken from the irrigation water reservoir five times during irrigation.

The soil samples were prepared following the direct equilibration method as proposed by Wassenaar et al. (2008) and described in detail by Sprenger et al. (2016). The precision for the method is reported to be $0.31\,\permil$ for $\delta\,^{18}O$ and $1.16\,\permil$ for $\delta\,^2H$ (Sprenger et al., 2015). All water samples were analyzed following the same procedure as described in Sec. 2.3.

## 2.5 Data analysis

### 2.5.1 TDR data analysis

Almost 5000 individual soil moisture measurements were taken during the irrigation experiment. As the three TDR probes had different integration depths ($0.12\,m$ and $0.18\,m$), the measurements had a different depth offset relative to the ground surface when referenced to the center of the probe and had to be aligned. To do so, the measurements, which were originally taken in $0.1\,m$ increments, were resampled in depths by linear interpolation. Due to the potentially short correlation length of soil moisture (Zehe et al., 2010), inverse distance interpolation between two locations is generally not appropriate. In case of the vertical profiles, however, the integration depths of the probes exceeded the measuring increments. Due to the resulting overlap of the integration volumes, this procedure was assumed to be adequate. The measurement of one depth increment took approximately between $10\,s$ to $30\,s$. While the data was interpolated in time for better visualization, all data analyses were performed with the uninterpolated data.

All TDR measurements were referenced to the last measurement before irrigation. The resulting data set of relative soil moisture changes $\Delta\theta$ was used for the discussion of soil moisture dynamics and response velocity calculation (Sec. 2.5.4). The storage changes [mm] in the top $1.4\,m$ of the core area were estimated based on the four core area TDR profiles TDR1, TDR2, TDR7 and TDR8, by multiplying the $\Delta\theta$ [%] of each depth increment with the respective depth interval [mm]. Together with the time series of water input this data was used to estimate the mass balance dynamics of the core area.

### 2.5.2 Data processing of 2D time-lapse GPR measurements

The time-lapse GPR survey comprised repeated recordings of vertical 2D GPR data along the four transects. The data processing of each measurement relied on a standard processing scheme, including bandpass filtering, zero time correction, exponential amplitude preserving scaling, inline fk-filtering, and a topographic migration approach, as presented by Allroggen et al. (2015a). The GPR data were analyzed using an appropriate constant velocity and gridded to a 2D transect with a regular trace-spacing of $0.02\,\mathrm{m}$.

There is no standard interpretation procedure for the analysis of time-lapse GPR data. Most approaches are based on calculating trace-to-trace differences (Birken and Versteeg, 2000; Trinks et al., 2001) or picking and comparing selected reflection events in the individual time-lapse transects (Allroggen et al., 2015b; Haarder et al., 2011; Truss et al., 2007). In the context of this study, however, both approaches provided only limited interpretable information. Considering the methodological uncertainty, the highly heterogeneous soil did not provide reflectors which were a suitable reference. Therefore, we used a time-lapse structural similarity attribute presented by Allroggen and Tronicke (2015), which is based on the structural similarity index known from image processing (Wang et al., 2004). This approach incorporates a correlation-based attribute for highlighting differences between individual GPR transects and has been shown to improve imaging, especially for noise data and limited survey repeatability.

The calculated structural similarity attributes are a qualitative indicator for relative deviations from the reference state. The GPR data indicated remaining water from the natural rain event when the experiment was started. Therefore, the last acquisition time 24:00 h after irrigation start was chosen as the reference time for all GPR transects. Based on the assumption, that the reference state is the one with the lowest water content, decreasing structural similarity was interpreted as an increase in soil moisture.

For converting GPR two-way travel time (TWT) into depth, we used the average measured GPR propagation velocity of $0.07\,\mathrm{m\,ns^{-1}}$. This velocity is based on additional common midpoint data and the assumption of static conditions during the experiment. Using this velocity, the GPR transects covered a TWT of $120\,\mathrm{ns}$, which corresponds to a depth of $\sim4.2\,\mathrm{m}$ below ground surface. Approximately the first $20\,\mathrm{ns}$ of each transect are influenced by the interfering arrival of the direct wave and the ground wave. Consequently, we observe no interpretable reflected energy in the uppermost time window. Thus, the 2D GPR measurements imaged the subsurface between $\sim0.7\,\mathrm{m}$ to $4.2\,\mathrm{m}$ depth below ground.

### 2.5.3 Comparison of natural event vs. irrigation based on 2D GPR data

To interpret the structural similarity attribute images, we discriminated between the signal of the natural rain event and the irrigation. The discrimination was based on the temporal dynamics of each pixel of the GPR transects (i.e. every single value in the matrix of distance along the GPR transect and depth/TWT). The first GPR measurements were taken 12:52 h after the end of the $2^{nd}$ rainfall event (i.e. 6:30 h before irrigation start) and the observed responses were attributed to the natural rain event. Once the structural similarity attribute value of a pixel decreased more than $0.15$ after irrigation start, the signal of that pixel was attributed to the irrigation. The threshold of $0.15$ was chosen based on the noise of the last measurement 18:00 h

after irrigation start and exceeds the standard deviation of that measurement by a factor of 3. The same procedure was applied to infer the time of first response to the irrigation signal, which was used to calculate response velocities.

This procedure yields 2D maps of response patterns, with each pixel being attributed to either the irrigation or the natural rain event. The structural similarity values are a semi-quantitative measure of soil moisture and thus no reliable indicator to directly compare actual soil moisture responses recorded at different locations or at different times. We therefore used the areal share of the monitored cross sections to compare the impact of the two input events. To do so, all pixels of one of the two categories (natural rainfall or irrigation) which fell below the value of $0.85$ (i.e. maximum similarity 1 minus threshold $0.15$), were counted and expressed as fraction of the entire cross section. The resulting areal share does not represent the actual share of activated flow paths, but is a semi-quantitative indicator for the hillslope cross section impacted by active flow paths.

### 2.5.4  Response velocity calculation

As no tracers were used for irrigation, dynamic processes had to be inferred from changes in state. For TDR measurements, the time of first response was defined as an increase of soil moisture by $2 \, \text{vol}\%$ relative to initial conditions. This threshold was chosen based on the standard deviation of measurements under presumably constant conditions. The time of first response was identified for each TDR profile and depth increment.

Due to the experimental setup, soil moisture dynamics on the core area were dominated by vertical processes, while lateral processes controlled the dynamics at the downhill monitoring area. Accordingly, vertical and lateral response velocities were calculated from core area and downhill monitoring area TDR profiles, respectively.

As a continuous wetting of the soil profile could not be assumed, all response velocities were calculated for the entire depth (or distance) instead of depth increments. Response velocities are therefore integrated values describing processes in the entire soil column above. This procedure also accounts for heterogeneous processes and preferential flow paths, which may bypass shallower depths without leaving a detectable soil moisture signal.

Lateral response velocities account for the depth and distance between soil surface at the rain shield and TDR profile in question and therefore, integrate lateral and vertical flow. They were calculated for every depth of the soil moisture profiles at the downhill monitoring area. The time of the very first response signal measured on the core area was used as reference time $t_0$, which was $15 \, \text{min}$ after irrigation start. Due to the slope-parallel or horizontal orientation of the saprolite, we assumed that the water flows either vertically or laterally rather than diagonally. Based on this assumption, the distances were calculated from the slope parallel distance of each profile from the lower boundary of the core area plus the depth of every measuring point. The distance assumptions for both, vertical and lateral velocity calculations, do not resolve tortuosity of flow paths and therefore, drastically reduce the complexity of the flow path network to its integral behavior. The calculated response velocities are thus not to be interpreted as *in situ* flow velocities in the flow paths, but rather as the minimum necessary velocity explaining the observed arrival of the wetting signal.

The same holds true for the lateral response velocities calculated from GPR data. In accordance to the separation of the natural rain event signal and the irrigation signal, the first decrease in structural similarity of more than $0.15$ was interpreted as the arrival of the irrigation signal. Single structures and flow paths are not the focus of this article and will be discussed

in the companion study by Jackisch et al. (this issue). Here, we therefore simplified the 2D patterns to a depth distribution of occurring response velocities. To do so, all areas that were newly activated at the time of one measurement were accumulated by depth, and given as portion of the entire width of each GPR transect. Comparable to the procedure applied to the TDR data, the response velocities were then calculated from the respective measuring time, the distance between transect and irrigation area and the depth. The resulting patterns show the spatial fraction of the depth increment which is connected to flow paths of the calculated velocity or faster and give an idea about the spatial distribution of the GPR response velocities.

## 3  Results

### 3.1  Response to the natural rainfall

#### 3.1.1  Stream-centered approach: Hydrograph and surface water isotopes

In response to the summer storm event just before the hillslope-scale irrigation experiment, all gauged sub-catchments showed double-peak hydrographs, with one immediate short peak, and one prolonged and by several hours delayed peak ($2^{nd}$ rainfall event, Fig. 5). In the headwater catchments (Holtz 2, Weierbach 1 and 2), the first peak occurred almost instantly, while the more distant Colpach gauge showed a delay of approximately 3:00 h. The second response was prolonged with a maximum approximately 36:00 h after the event. The strength and ratio of the two peaks varied across different sub-catchments and according to hydrological conditions, but the general pattern is characteristic for the hydrological behavior of the Colpach River catchment. Similar behavior was also reported by Fenicia et al. (2014), Wrede et al. (2015) and Martínez-Carreras et al. (2016), whose investigations focused on the Weierbach 1 catchment.

A simple mass balance calculation revealed, that the first peak constituted $7.5\,\%$ of the total event runoff at gauge Holtz 2. The total event runoff coefficient was $0.44$. Referenced to the precipitation amount, about $3.3\,\%$ of the input left the headwater within 7:00 h after the rain event. In the neighboring Weierbach catchment and the Colpach, the first peak contributed more strongly to the total event runoff ($14.2\,\%$ and $12.9\,\%$ at Weierbach 1 and 2, and $19.7\,\%$ at Colpach).

The $\delta^{18}O$ signature of the stream water is indicative for the origin of the water. It showed strong dynamics during the discharge response to the rain events on June 20 (Fig. 5). The results of the hydrograph separation show that the event water contributed up to $67.6\,\%$ to the event runoff during the response to the first rain event in the morning of June 20 (Colpach, 6:00h). After that, the total discharge dropped again, with the event water contribution decreasing to $31.6\,\%$. With the onset of the first peak caused by the $2^{nd}$ rain event at 19:20h, the event water contribution increased again and reached values of over $50\,\%$ ($58.0\,\%$ in the Colpach at 22:00h, $55.2\,\%$ in the Holtz 1 catchment, 21:51h on June 20. Fig. 5). $\delta^{18}O$ values then declined, indicating event water contributions of around $20.0\,\%$ ($24.8\,\%$ at 4:00h in the Colpach, $18.1\,\%$ at 13:03h in Holtz 2 and $16.2\,\%$ at 21:28h in Holtz 1 on June 21). Weierbach 1 and 2 showed the same pattern, with event water contributions well above $50.0\,\%$ for the first peak of the $2^{nd}$ rainfall event.

Uncertainty in hydrograph separation was caused by the uncertainty of the stable isotopic composition of the precipitation input. The uncertainty due to spatial variability of the precipitation input was kept minimal for Holtz 2, by sampling the

precipitation within the small catchment ($45.9\,\text{ha}$). While we could not sample the isotopic input in high temporal frequency, the bulk sample of the precipitation data represents a weighted average of the input isotopic signal.

### 3.1.2 Hillslope-centered approach: Subsurface response patterns

The 2D time-lapse GPR measurements yield images of structural similarity referenced to the last measurement, which were taken $24\!:\!00\,\text{h}$ after irrigation start, which translates to $43\!:\!22\,\text{h}$ after the $2^{nd}$ natural rain event. The first GPR measurements were taken about $12\!:\!52\,\text{h}$ after the $2^{nd}$ rain event and can be interpreted as the subsurface response patterns of this event (Fig. 6A). The subsequent GPR measurements furthermore show the temporal dynamics of the rainfall signal, overlain by the irrigation signal. The high initial signals, as well as the high but decreasing areal share of active regions in all transects during the first measurements until $1\!:\!30\,\text{h}$ after irrigation start (Fig. 6B) indicated free water remaining from the preceding natural rain event which slowly disappeared.

While transect 1 showed only a weak signal of the natural event in the first measurement, transects 2 and 3 exhibited stronger and longer lasting signals. The areal share of active regions of the four transects in the measurement preceding the irrigation experiment was $38.5\,\%$, $51.6\,\%$, $64.4\,\%$ and $50.5\,\%$ from upslope to downslope. Except for transect 2, which even showed a slight increase in areal share of active regions between the first and second measurement, the signal of the natural rain event was continuously vanishing (Fig. 6B).

### 3.2 Hillslope-scale irrigation experiment

### 3.2.1 Core area water balance dynamics

The irrigation intensity was relatively constant over time, with only weak fluctuations due to gradual clogging of the intake filter. The spatial distribution of the irrigation intensity on the core area was influenced by the sprinkler setup and the slope of the experimental site. The mean intensity on the core area was $30.8 \pm 7.3\,\text{mm}\,\text{h}^{-1}$, with slightly higher values in the vicinity of the four sprinklers. Surface runoff at the lower boundary of the core area started $20\,\text{min}$ after irrigation start and ceased with the same time lag. In total, surface runoff amounted to $0.5\,\text{L}$, which equals only $0.02\,\%$ of the water balance.

The core area mass balance is shown in Fig. 7, depicting the storage increase in the top $1.4\,\text{m}$ of the soil. All profiles showed a mass recovery of more than $100\,\%$ (i.e. higher storage increase than water input at measuring time, see Fig. 7) in the first $60\,\text{min}$ of the irrigation period. In profiles TDR1, TDR2 and TDR8 mass recovery then decreased and dropped below $100\,\%$, while TDR7 increased further, with a maximum overshoot of almost $50\,\%$ approximately $2\!:\!00\,\text{h}$ after irrigation start. The last measurement during irrigation was taken approximately $50\,\text{min}$ before the end of the irrigation period. At this time, the average storage increase was more than $20\,\%$ lower than the input mass.

The first measurement after irrigation ($6\,\text{min}$ to $19\,\text{min}$ after irrigation stopped) showed a mean deficit of $54.7\,\%$, indicating that on average $31.2\,\%$ (between $18.6\,\%$ and $43.9\,\%$) of the water that has been recorded at the last measurements before irrigation stop was freely percolating and had left the monitored depth immediately. After this fast instantaneous reaction, the

water content decreased equally. Mean total mass recovery dropped to $8.9\,\%$ after 18:24 h after irrigation start and almost returned to initial conditions ($1.6\,\%$) after 24:00 h after irrigation start.

### 3.2.2 Soil moisture profiles and dynamics

The high variability in soil moisture dynamics observed in the TDR profiles is summarized in Fig. 8. The four uppermost panels (row 1 and 2) show the core area profiles. Columns represent the three diverging TDR transects. The general pattern observed at the core area was characterized by strong and comparably fast response in the top $0.4\,\mathrm{m}$ of the soil, and below the depths of approximately $1.2\,\mathrm{m}$. The response in between these active layers was more diverse and generally weaker.

Soil moisture in the top $0.4\,\mathrm{m}$ of the soil of TDR1, TDR7 and TDR8 quickly stabilized around constant values, indicating the establishment of quasi steady-state conditions. After the end of the irrigation, the soil moisture quickly declined down to a $\Delta\theta$ of $4\,\mathrm{vol\%}$, indicating a very fast response to the dynamics of the water input. In contrast to the fast establishment of steady-state conditions and the fast decline, a slightly increased water content of up to $4\,\mathrm{vol\%}$ above initial conditions was persistent in distinct depth increments and was also measured even 24:00 h after irrigation stopped.

The soil moisture patterns at the downhill monitoring area were more diverse. Profiles located directly below the rain shield (TDR9, TDR3 and TDR10 with a distance to core area of $0.2\,\mathrm{m}$ to $0.5\,\mathrm{m}$, Fig. 8) exhibited dynamics that resemble the reaction at the core area, but with mostly lower intensities and higher variability in depth. More distant TDR profiles however, showed a highly variable picture. Distinct layers in variable depths were activated, while no change of the water content was seen in the other soil depths. Especially noteworthy are TDR10 and TDR11, which showed a strong soil moisture increase of up to $18\,\mathrm{vol\%}$ below $1.4\,\mathrm{m}$ depth and around $10\,\mathrm{vol\%}$ in the top $0.3\,\mathrm{m}$ of the soil. Profiles TDR13, TDR6 and TDR14 showed only weak signals, with the strongest response below $1.4\,\mathrm{m}$ below ground in TDR6. Profiles TDR6, TDR12, TDR13 and TDR14 showed the strongest decrease in soil moisture over the course of the measurments, with $\Delta\theta$ of $-3.7\,\mathrm{vol\%}$, $-4.4\,\mathrm{vol\%}$, $-5.8\,\mathrm{vol\%}$ and $-6.2\,\mathrm{vol\%}$ in certain depths, indicating vanishing free water from the storm event. While the results from the left (TDR7, TDR9 and TDR11) and the right transect (TDR8, TDR10 and TDR12, see Fig. 8) suggested lateral flow in different depths, the central transect (TDR1 through TDR6) did not indicate lateral flow.

### 3.2.3 Time-lapse GPR dynamics

The TDR measurements at the downhill monitoring area were complemented by the 2D time-lapse GPR measurements (see Fig. 4), yielding 2D images of structural similarity attributes referenced to the last measurement 24:00 h after irrigation start (Fig. 9). The first irrigation signals (shown in blue in Fig. 9) appeared in the first measurement after irrigation start (1:28 h), with transect 1 showing the clearest response. After about 3:23 h strong, localized signals occurred and increased in intensity over time. The general maximum was reached approximately 5:18 h after irrigation start, showing distinct activated flow paths. Most signals started to decline after 6:45 h, which is 2:10 h after the end of the irrigation period.

In transect 2 some weak signals appeared in the depth below $2.5\,\mathrm{m}$ 1:30 h after irrigation start. At this time, the signal was close to noise level, but the pattern became stronger and more distinct in the following measurements. At transect 3, the persisting signal of the natural rain event made it difficult to identify the irrigation induced response. However, a weak

irrigation signal appeared after 1:28 h and reached its maximum at 6:45 h after irrigation start. At transect 4 the structural similarity attribute values were generally low, which indicates a low deviation from the reference state. Either the mobile water showed low dynamics (with regard to total mass over time), or water was less confined to specific structures and local changes are less pronounced. Both interpretations suggest that this transect was generally wetter due to its proximity to the river. Overall, the experiment does not appear to have affected this transect much.

The overview over all GPR measurements in Fig. 6B and 9 visualizes the dynamics of the hillslope section. The green natural rainfall signal faded from uphill to downhill, with the highest intensity and duration in transect 3. After irrigation start, the blue irrigation signal appeared, gradually propagating downhill and eventually overpowering the natural rain signal. 18:00 h after irrigation start (i.e. 13:25 h after irrigation ended), no changes in GPR signal could be observed anymore. This suggests steady soil moisture conditions and thus, the absence of highly mobile water in all transects. The mobile water either left the monitored area, or dispersed by diffusion into the matrix surrounding preferential flow paths, where it remained beyond the time of the reference measurement and thus would not have been visible by means of structural similarity attributes.

### 3.2.4 Pore water and piezometer isotope responses

The temporal dynamics of the stable isotope compositions of the pore water (selected depths shown as circles in Fig. 10A) partially traced the signals of the rainfall and irrigation water input (green lines and blue triangles). The high $\delta^2 H$ signal of the first minor rainfall event (dark green) was clearly visible in the top $10 \, \text{cm}$ below ground in the profile sampled at the downhill monitoring area 24:00 h before irrigation and 5:30 h after this rainfall event (Fig. 10A and C). Similarly, the isotope signal of the $2^{nd}$ rainfall event (light green) and the irrigation water (blue triangles) could be seen in the top $10 \, \text{cm}$ of the soil at the core area and the downhill monitoring area, respectively. Especially the isotope profile taken at the core area after irrigation showed an increase in $\delta^2 H$ in the top $0.85 \, \text{m}$ below ground, showing the influence of both, the irrigation and the event water. Below the depth of approximately $1.2 \, \text{m}$ of all profiles, the soil water isotope composition seemed not to be impacted by the rainfall events and irrigation.

Only few $\text{mL}$ of water were seeping into the piezometers, with piezometer B being the only one that could be sampled more than once. Piezometer B was sampled first shortly after the irrigation ended (0:20 h) and showed a composition that was close to the irrigation water. The other two samples 1:32 h and 13:38 h after irrigation ended, showed a decrease in $\delta^2 H$, towards the composition of the soil water (red and orange diamonds in Fig. 10).

Piezometers A, C, G and H were sampled once 13:38 h after irrigation ended. The water sampled from piezometers at the core area (A and C, pink diamonds) showed the same composition as the irrigation water. Piezometers located at the downhill monitoring area (G and H, purple diamonds) in contrast, showed an isotopic composition similar to the rainfall water and different to the pore water in the depth profiles.

### 3.2.5 Response velocities

The results of the calculated response velocities from TDR and GPR measurements are summarized in Fig. 11. The top row shows the depth distribution of observed response velocities for all GPR transects. The bottom row shows the TDR-based

results, separated in core area profiles and the three diverging transects. Additionally to the TDR-based response velocities, GPR-based velocities observed in $0.5\,\mathrm{m}$ wide sections of the GPR transects which were closest to the displayed TDR profiles, are shown in the plot.

At the core area, the dominating vertical response velocity was around $10^{-4}\,\mathrm{m\,s^{-1}}$, with a tendency to increasing velocities with depth (Fig. 11, bottom left). As response velocities were calculated for the entire soil profile above the measuring depth, this increase indicates a bypass of intermediate depths through preferential flow paths, and a limited and slow interaction with the matrix. The highest observed vertical velocity was $10^{-3}\,\mathrm{m\,s^{-1}}$ in the depth of $1.4\,\mathrm{m}$ below ground. The respective soil moisture signal was recorded in the very first measurement after irrigation start, which indicates that we might have even missed the first response.

Similar to the vertical response velocities, the dominant TDR-based response velocity at the downhill monitoring area was in the order of magnitude of $10^{-4}\,\mathrm{m\,s^{-1}}$ (Fig. 11 bottom row). Response velocities of around $10^{-3}\,\mathrm{m\,s^{-1}}$ were observed in six profiles all over the downhill monitoring area, of which the highest values ($1.0$ to $1.6 \times 10^{-3}\,\mathrm{m\,s^{-1}}$, TDR4, TDR6, TDR11 and TDR17) are based on signals observed during the first profile measurements after irrigation start. The fastest response was observed in the top $0.5\,\mathrm{m}$ (TDR4, TDR6, and TDR10) of the soil and below a depth of $1\,\mathrm{m}$ (TDR6, TDR11, TDR17, and TDR18).

The GPR-based response velocities were calculated for the entire width of the GPR transects down to the maximum depth of approximately $4.2\,\mathrm{m}$ (Fig. 11 top), and for single sections close to the TDR profiles down to a depth of $1.7\,\mathrm{m}$ for comparison of the methods (Fig. 11 bottom). All GPR transects showed slight and localized irrigation water signals in the data collected $1{:}28\,\mathrm{h}$ after irrigation start (see Fig. 9). This translates to response velocities between $1.4 \times 10^{-3}\,\mathrm{m\,s^{-1}}$ to $2.2 \times 10^{-3}\,\mathrm{m\,s^{-1}}$, depending on the distance to the irrigation area and signal depth. The data of the next measurements suggest response velocities between $5.4 \times 10^{-4}\,\mathrm{m\,s^{-1}}$ to $8.2 \times 10^{-4}\,\mathrm{m\,s^{-1}}$. Given the fact that this is a conservative estimate due to the even sparser temporal resolution in comparison to the TDR measurements, response velocities are likely to be similar to or even higher than those calculated from TDR results.

### 3.3 Comparison of natural event and irrigation response patterns in 2D GPR images

The signal of the natural rainfall event could be observed throughout the entire transects. The highest signal density (i.e. areal share) was found between $0.9\,\mathrm{m}$ to $1.7\,\mathrm{m}$ depth in all transects (Fig. 12). The strongest response signal (i.e. lowest structural similarity) appeared below the depth of $2.5\,\mathrm{m}$ in transects 2 and 3 (Fig. 6A). Transect 4 showed generally higher structural similarity and thus, lower response signals.

The areal share of the irrigation signal was the highest in transect 1, with $30.5\,\%$, and decreased downhill, with $20.8\,\%$ in transect 2, $16.4\,\%$ in transect 3, and $6.0\,\%$ in transect 4 (Fig. 12). Transect 1 also shows irrigation signals in regions which were not (or not any more) active after the rain event. Especially the depth between $1.7\,\mathrm{m}$ to $2.2\,\mathrm{m}$, which showed a comparably low natural rain signal, was activated by the irrigation. Overall $39.7\,\%$ of the regions activated by irrigation in transect 1 have already been active before irrigation. In contrast, most of the irrigation signals observed in the three downhill transects consist

of re-activated flow paths. Here, 72.5 %, 86.1 % and 76.7 % of the irrigation patterns were activated both by the natural event as well as by the irrigation experiment (Fig. 6 and Fig. 12).

The natural rain signal in the GPR data was overpowered by the irrigation signal at 20:27 h, which is 3:23 h after irrigation start and about 22:45 h after the natural rain event. The dampened dynamics in transect 4 were due to its proximity to the river and therefore generally wetter conditions and less capacity for additional wetting.

The comparison of the two different response patterns shows that both, the natural rain event and the irrigation caused advective flow in discrete flow paths more or less evenly distributed over the hillslope cross-section. An (ephemeral) groundwater body or specific flow layers could not be identified in the top $4.2\,\mathrm{m}$ of the subsurface. Furthermore, the artificial irrigation had only a minor impact in comparison to the natural rain event, despite higher local input. Water that was supplied from upslope areas was therefore more important for the downhill soil moisture response than irrigation intensity or duration.

## 4 Discussion

### 4.1 Process interpretation

#### 4.1.1 Irrigation experiment

The mass balance at the core area showed an overshoot in calculated mass recovery (i.e. higher mass recovery than water input) during the first $1{:}00\,\mathrm{h}$ of the irrigation period (Fig. 7). This might have been related to the spatial heterogeneity in irrigation intensity or lateral redistribution of water in the shallow subsurface. After that, the four core area profiles behaved differently. While the storage change in TDR7 was continuous, the other three profiles showed a stepwise behavior with abrupt stagnation and storage increase. This behavior is interpreted as a stepwise activation of flow paths in vertical or lateral direction. The decrease in mass recovery, which started approximately $1{:}00\,\mathrm{h}$ after irrigation start, signified a loss of water from the core area ($0\,\mathrm{m}$ to $1.4\,\mathrm{m}$ depth). Thus, flow paths towards greater depth and the downhill monitoring area established around this time.

The fast soil moisture response in depth and at the downhill monitoring area (Fig. 8), as well as the immediate drainage after irrigation stop according to the mass balance (Fig. 7) also suggested a high fraction of mobile water. The mobile water is not bound to the matrix and likely subject to advective flow with high velocities of over $10^{-3}\,\mathrm{m\,s^{-1}}$, according to the first response observed in TDR and GPR measurements (Fig. 11). The order of magnitude of the response velocities agreed with the *in situ* measurements of hydraulic conductivity (up to $10^{-3}\,\mathrm{m\,s^{-1}}$ and higher), but clearly exceeded the potential of matrix flow for the silty matrix material (Jackisch et al., this issue). Similarly high preferential flow velocities are reported for the well studied MaiMai hillslope, with initial breakthrough velocities between $6.7 \times 10^{-3}\,\mathrm{m\,s^{-1}}$ and $3.3 \times 10^{-2}\,\mathrm{m\,s^{-1}}$, being at least two orders of magnitude higher than matrix flow, which ranges between $3.8 \times 10^{-6}\,\mathrm{m\,s^{-1}}$ and $1.04 \times 10^{-4}\,\mathrm{m\,s^{-1}}$ (Graham et al., 2010).

Various studies report a concentration of lateral preferential flow at a more or less impermeable bedrock interface for other sites (e.g. Graham et al., 2010; Tromp-Van Meerveld and McDonnell, 2006), which has also been hypothesized for the Colpach River catchment (e.g. Fenicia et al., 2014). However, none of the piezometers showed a significant reaction, despite being

installed in the depths with the highest observed soil moisture responses. Instead, both, TDR profiles and the time-lapse GPR transects revealed very heterogeneous patterns and a soil moisture response in multiple depths, as was also reported by Wienhofer et al. (2009) for a mountainous hillslope with young, structured soils.

The heterogeneous flow patterns observed with both TDR and GPR (Fig. 8 and 9) and the delayed signal in the intermediate depth at the core area (Fig. 8), suggest a heterogeneous network of preferential flow paths, which bypassed a large portion of the unsaturated soil (see also Jackisch et al., this issue). The water either passes through the intermediate depth outside of the monitored soil volume, or through small preferential flow paths. If the volume of these flow path is comparably small in comparison with the soil volume monitored by the TDR probes, they will only become visible (by means of soil moisture changes) if the water leaks into the surrounding matrix and effectively increases the soil moisture content of the integration volume. While we can not distinguish between these processes by means of the data presented here, both are preferential flow processes acting at different scales.

The pore water and piezometer stable isotope composition at the core area showed that the freely percolating water on the core area was predominately constituted of irrigation water (Fig. 10A and B). Piezometer B, which was the only piezometer to be sampled more than once, showed a trend from irrigation water composition shortly after irrigation stop, towards pore water composition, 14:38 h later. This trend indicates that the irrigation water first percolated through the preferential flow paths without significant mixing with old water. After the water supply ended, however, preferential flow is (partially) fed by pore water, suggesting mixing and interaction between matrix and preferential flow as suggested by Klaus et al. (2013).

The piezometers at the downhill monitoring area in contrast, had water with the same isotopic composition as rain water of the $2^{nd}$ rainfall event prior to the irrigation experiment (Fig. 10C). This water has been re-mobilized, as it only seeped into the piezometers after the irrigation, but shows no signs of interaction with the soil matrix or the irrigation water. While this observation has previously been made in other soils with well-developed macropore systems (Leaney et al., 1993), it contradicts the observations at the core area and rather suggests dual flow domains.

### 4.1.2 Natural rainfall event observations

The subsurface response patterns revealed by the GPR measurements after the natural rainfall events were similar to the irrigation induced patterns with regard to their patchiness, but showed a slightly different spatial distribution (Fig. 12). The GPR measurements 12:52 h after the $2^{nd}$ rain event showed the highest density of response signals in the depth between $0.8\,\mathrm{m}$ to $1.7\,\mathrm{m}$ (Fig. 6A and 12). The higher response in the shallow depth could be interpreted as the signal of vertically infiltrating rain water, which did not occur during the irrigation. This depth also correlated with high stone content and the periglacial cover beds, which are characteristic for the area (e.g Juilleret et al., 2011) and have also been observed at the investigated hillslope (Jackisch et al., this issue). While no (transient) water table could be detected in the monitored depth, the patterns indicated a concentration of preferential flow paths in this depth.

The TDR measurements also showed high initial soil moisture content and a strong reaction to irrigation in this depth. However, except for the slight decrease in soil water content in the most downhill located TDR profiles TDR6, TDR13 and TDR14 (Fig. 8), soil moisture values barely fell below the initial values measured between rainfall events and irrigation. This

means that the signal of the natural rainfall events was already gone in the shallow subsurface, and that no information on the natural rain signal in this depth can be derived from the TDR data.

The timing of the response dynamics observed with the GPR measurements and the discharge response also shed light on the prevalent processes. The natural rainfall events ended at 21:35 h on June 20, 2013. The first GPR measurements were taken at 10:42h on June 21, 12:52 h later (Fig. 5). Located at the lower section of the hillslope, they were interpreted to show a declining soil moisture signal, which was mostly gone 37:22 h after the $2^{nd}$ rainfall event (i.e. 18:00 h after irrigation start, see Fig. 6B and 9). Following the hypothesis of a top to bottom drainage of the hillslope, and considering the downslope location of the study site, the recorded signal represented the tailing of the shallow subsurface flow response to the natural rainfall event.

At the time of the first GPR measurements, the first peak of the hydrograph was already gone, while the second peak was on its rising limb and reached its maximum 12:00 h later (24:52 h after rainfall event, Fig. 5). Thus, the following decline in subsurface response was observed after the first peak and coincided with the rise of the second peak. This timing provides strong evidence that the second hydrograph peak was not primarily caused by the activation of the observed preferential flow paths in the shallow subsurface.

Several studies investigated the double-peak hydrographs of the Weierbach catchment. Wrede et al. (2015) used dissolved silica and electrical conductivity and found that the first peak was dominated by event water, while the second peak mainly consisted of pre-event water and strongly depended on antecedent conditions. Based on these observations, the first peak was attributed to fast overland flow from near-stream areas, while the second peak was attributed to subsurface flow where antecedent water was mobilized. Fenicia et al. (2014) came to a similar conclusion and identified a riparian zone reservoir as the origin of the first peak.

The stable isotope data collected during the rainfall event prior to the irrigation experiment, showed the same dynamics as observed by Wrede et al. (2015) (Fig. 5). The isotopic composition of the first peak suggested a mixture of event water and pre-event water, while the composition of the second peak indicated the dominance of pre-event water. A simple water balance revealed, that the total mass of the first peak of the event runoff accounted for about $4\%$ of the precipitation amount in the Holtz 2 catchment. Based on a rough delineation, the existing wetland patches in the catchment amounted to approximately $800\,\mathrm{m^2}$ in the source area. Thus, the specific discharge of the first peak exceeded the existing wetland patches or riparian zones of the headwater catchment by a factor of 27, and suggests that overland flow from near-stream saturated areas could not solely explain the observed discharge response.

### 4.1.3 Synthesis: Functioning of the investigated hillslope

Comparison of the GPR data of the natural rainfall event and the irrigation reveals that the response in the shallow subsurface was stronger after the natural event, even though the input per square meter was much lower than during the irrigation (Fig. 6 and 12). The observed response could only be caused by the accumulated water input of the (entire) hillslope draining through the shallow subsurface. Thus, the hillslope is prone to a substantial amount of lateral flow, which quickly ceases after water supply stops.

In combination with this finding, the high potential response velocities (Fig. 11) and revealed by the TDR and GPR measurements show, that fast, lateral subsurface flow is an important process in the investigated hillslope. The timing of the hydrograph dynamics and the declining response in the GPR measurements described above (Fig. 5), as well as the freely percolating rainfall event and irrigation water shown by the stable isotope data, give further evidence that the activation of preferential flow paths within the shallow subsurface was contributing to the first immediate peak of the stream hydrograph.

Preferential flow paths established quickly, and high response velocities have the potential to route water from the hillslopes towards the river within few hours. The presence of preferential flow paths and the steep slopes in the Colpach River catchment were reported to enable subsurface runoff, even at times when the soil and weathered zone are not yet at field capacity (Van den Bos et al., 2006; Nimmo, 2012).

Many catchments reportedly showing double-peak hydrographs are headwater catchments with predominantly steep slopes and shallow soils, and in many cases with periglacial slope deposits (Burt and Butcher, 1985; Onda et al., 2001; Graeff et al., 2009; Birkinshaw and Webb, 2010; Fenicia et al., 2014; Wrede et al., 2015; Martínez-Carreras et al., 2016). Such systems are characterized by pronounced subsurface structures and therefore, prone to heterogeneous flow patterns and preferential flow at the plot- and hillslope-scale. The results on subsurface structures presented in the companion study by Jackisch et al. (this issue) also support this interpretation. Inter-aggregate flow paths at the scale of $5 \times 10^{-3}$ m to $5 \times 10^{-2}$ m are the reason for the highly variable hydraulic conductivities found in the investigated area and enable such high flow velocities. While these structures could only be revealed at the plot-scale by excavation and direct observation, the related patterns in soil moisture response were similar (with regard to activated depths and response velocities) also for lateral flow at the hillslope-scale.

The processes causing the second peak could not be resolved with this study, but it is hypothesized that deep percolating water from hillslopes and plateaus caused the delayed response. This hypothesis is backed by a study comparing catchments of different geology (Onda et al., 2001), where the prolonged response of double-peak hydrographs was identified as an indicator for deeply percolating subsurface flow through bedrock fissures. This theory might also apply to the here investigated catchment with its fractured schist bedrock (Kavetski et al., 2011). Furthermore, deep subsurface storage overflow (Zillgens et al., 2007) and fast groundwater displacement (Graeff et al., 2009) are processes, which may play a role in the behavior of the Colpach River catchment. These hypotheses are also backed by the isotopic composition of the second peak, suggesting the dominance of pre-event water (Fig. 5), and could explain the dependency of the occurrence of double-peak hydrographs on groundwater storage as described by Wrede et al. (2015) and Martínez-Carreras et al. (2016). They could only observe the second peak if the groundwater storage was sufficiently filled, resulting in a hysteretic threshold behavior for the occurrence of the second, delayed peak.

## 4.2 Form and function in hillslope hydrology

Similar to the categorization of methods and observation data (see Fig. 1 and 3), we also subsumed the results and findings under the form-function-framework. By doing so it becomes clear, that observations and results are not always linearly obtained. Function is not necessarily described best by mere process observations. We thus want to discuss our data with regard to their value for the findings of the different form and function categories.

As elaborated in Sec. 4.1.3, the response observations described the functioning of the investigated hillslope well. Response dynamics and their temporal relation to each other across scales were a valuable source of information, shedding light on the characteristics of subsurface flow processes. Response patterns and spatially distributed point observations helped to develop a conceptual idea of the spatial organization and the functional network of the system. They were necessary to provide the spatial context to calculate response velocities and develop an idea of the establishment of flow paths. As such, response observations were the major key stone towards understanding the investigated system, supporting hypothesis H1: the function of a system can be described by response observations.

In addition to pure response observations, however, basic knowledge about structures and local characteristics strongly improved the interpretability of our data and reduced the ambiguity. This basic knowledge included the presence of periglacial slope deposits, the downslope position at the hillslope, and the hydraulic conductivity of the subsurface. This information was easily obtained from literature and in the field. It allowed us to close gaps in observational scales and link local observations at the hillslope to the overall system responses, and was thus the basis to more reliably relate the hillslope-centered and stream-centered observations.

Without this structural knowledge, the informative scope of response observations is limited to their observation scale. Transfer to other scales remains speculative. This fact is an important aspect, as most *in situ* response observations suffer from limitations in spatial resolution. Soil moisture measurements are restricted by their integration volume, and point measurements in general struggle to cover the entire domain. Here, more detailed information on (flow-relevant) structures might greatly improve our process understanding.

While we were able to describe processes in the investigated hillslope in great detail, new findings on structure inferred from function observations are scarce. Any detail on the form of the investigated hillslope we concluded from our observations, were mere confirmations of previous knowledge. The only knowledge on form that was gained from the presented data, concerns the flow-relevance of structures. We found that substantial lateral flow was occurring in an intermediate depth, most likely associated with the periglacial slope deposits. We also found, that distinct preferential flow paths occurred in greater depth, which suggests that the bedrock might not be as impermeable as previously assumed (Fenicia et al., 2014).

A conclusive picture of subsurface structures could not be drawn from process observations, partially refuting hypothesis H2, stating that function helps to reveal form. However, the observed response patterns were helpful to characterize known structures with regard to their flow-relevance.

## 4.3  2D time-lapse GPR measurements as link between form and function

The interpretation of the time-lapse 2D GPR measurements is difficult and requires a thorough understanding of the expressiveness of the recorded data. While changes in structural similarity at a short time scale can be attributed to variations in soil moisture content, these changes contain no information on where the soil moisture content increases or decreases (i.e. the direction of the change). The data are qualitative information only, and supportive measurements are necessary. Therefore, GPR measurements need to be accompanied by other methods, such as TDR measurements or trenches, to provide a reference for the observed changes.

In the correct setup, however, 2D time-lapse GPR measurements are a very powerful tool and a valuable source of information. Especially in highly heterogeneous hillslopes, the spatial context provided by the GPR measurements is crucial for the investigation of complex preferential flow networks. This spatial context can not be provided by ERT measurements with their comparably low spatial resolution, highly invasive excavated trenches or by point measurements. In contrast to dye tracer excavations, time-lapse GPR measurements yield the temporal component, which is necessary to cover the highly dynamic processes. They allow for repetitions in time and space and avoid the manipulating effect of a trench face on flow through the unsaturated zone (Atkinson, 1978).

Within the form and function framework, the biggest advantage of 2D time-lapse GPR measurements lies in the combination of high resolution spatial response patterns and dynamics. The method provides a direct link between flow-relevant structures and processes. It allows us to map response patterns in high temporal resolution, without manipulating the subsurface flow field. As such, this method has the potential to visualize the gradual establishment of flow paths, localize them and calculate response velocities (hypothesis H3).

## 5  Conclusions

The study site is an example for headwater catchments with steep slopes and young and highly structured soils, typical for landscapes that formed under periglacial conditions. Here, preferential flow paths quickly developed in the unsaturated zone within minutes after the onset of an intense irrigation or rain event, causing lateral flow across the hillslope. In combination with the high response velocities of up to $10^{-3}\,\mathrm{m\,s^{-1}}$ or faster, and the large fraction of mobile water, these flow paths have the potential to quickly route water from the hillslopes towards the stream.

The strong dynamics and high spatial variability of preferential flow challenge the investigation of these processes. While we were able to describe the overall flow dynamics, the spatio-temporal resolution of our monitoring setup was not sufficient to reliably quantify the maximum response velocities. Our study has furthermore shown that causes and importance of observations can only be evaluated if the necessary context is known. This context includes knowledge of the spatio-temporal patterns on the one hand, and relevant process scales on the other.

The spatio-temporal context is provided by a combination of quantitative point measurements (TDR), qualitative mapping of patterns and dynamics (2D time-lapse GPR) and the observation of integrated system response (hydrographs and stable isotopes). Either of these approaches provides a substantial piece to the puzzle, while neither of them on its own would have provided the full picture. The experiment has shown that time-lapse GPR measurements are a powerful tool which provides new perspectives for the investigation of preferential flow processes in hillslopes. The methodology's flexibility and minimally invasive character allow for repetitions in time and space, and thus, the direct observation of processes under driven conditions. Depending on the research question, the method can replace labor-intense trenches and increase the observation density.

The observation of response patterns and dynamics by means of the TDR and GPR measurements were shown to suffice to characterize subsurface flow within the hillslope. Processes were identified and characterized without any concrete information about spatial structures. However, despite the high number of TDR observations and the 2D response patterns obtained from the

time-lapse GPR measurements, our observations were methodologically limited in spatial and temporal resolution. Measuring intervals and integration volumes of the methods are restricted, and interpretations beyond observation scale remain speculative. Conclusions on or links to larger or smaller scales are not reliable. Here, more detailed information on spatial structures and their impact on flow processes might improve our understanding of the investigated area and will improve the ability to transfer findings to other scales by providing the physical basis behind the observed processes.

The observed response patterns, revealed by the GPR and TDR measurements, allowed us to develop a conceptual description of the flow path network, which is linked to subsurface structures. Certain spatial characteristics of the flow path network such as layers prone to preferential flow could be inferred from the response patterns. However, actual structural features, such as the delineation of the deposit layer or the bedrock interface, could not be localized. All structure related conclusions merely confirm previous findings. The topic of structural exploration is taken up in the companion paper by Jackisch et al. (this issue) to further elaborate and discuss the methodological aspects linking form and function in hillslope hydrology.

*Acknowledgements.* We are grateful to Marcel Delock, Lisei Köhn and Marvin Reich for their support during fieldwork, as well as Markus Morgner and Jean Francois Iffly for technical support, Britta Kattenstroth for hydrometeorological data acquisition and isotope sampling and Barbara Herbstritt and Begoña Lorente Sistiaga for laboratory work. Laurent Pfister and Jean-Francois Iffly from the Luxembourg Institute of Science and Technology (LIST) are acknowledged for organizing the permissions for the experiments and providing discharge data for Weierbach 1 and Colpach. We also want to thank the three anonymous reviewers, whose thorough remarks greatly helped to improve the manuscript. This study is part of the DFG funded CAOS project *From Catchments as Organised Systems to Models based on Dynamic Functional Units* (FOR 1598).

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

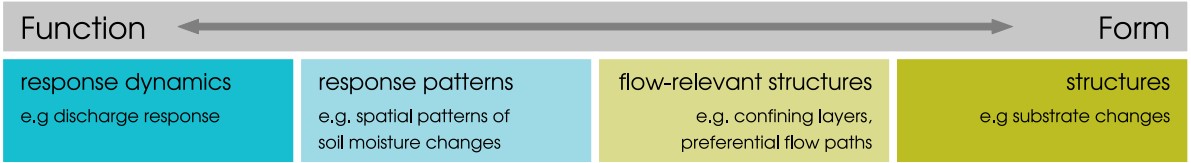

| Function | | | Form |
|---|---|---|---|
| response dynamics e.g discharge response | response patterns e.g. spatial patterns of soil moisture changes | flow-relevant structures e.g. confining layers, preferential flow paths | structures e.g substrate changes |

**Figure 1.** The concept of form and function applied to observations in hillslope hydrology. Four different categories which can be applied to data as well as the data sources.

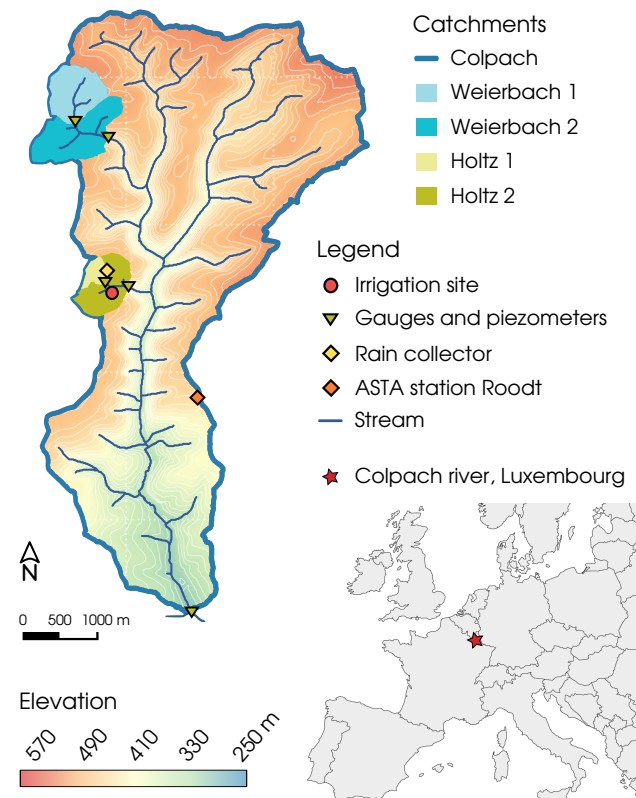

**Figure 2.** Map of the investigated Colpach River catchment and the four gauged sub-catchments. The site of the hillslope-scale irrigation experiment is located in the Holtz 2 catchment and indicated in red.

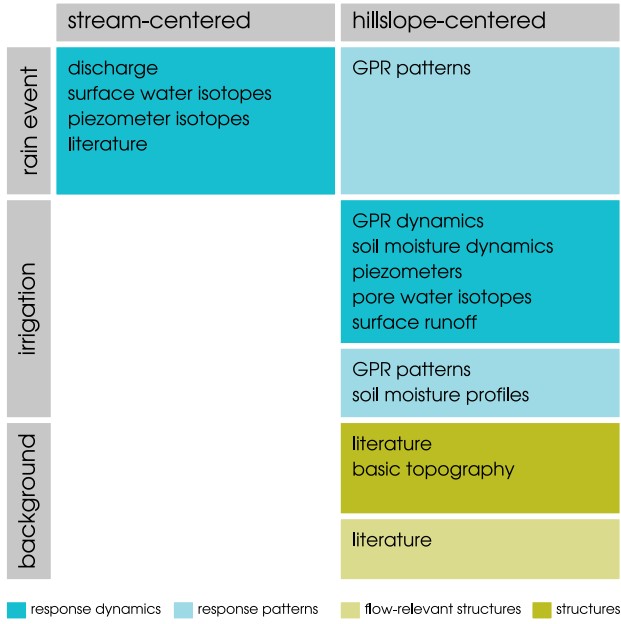

**Figure 3.** Experimental methods applied in the stream-centered and hillslope-centered approaches, divided into the sampling during the natural rain event and the irrigation experiment. Additionally, some structural background information was obtained from the literature and a digital elevation model.

**Table 1.** Overview over sub-catchment size and accumulated specific discharge as percentage of the precipitation amount [%]

| Gauge | size [ha] | accumulated discharge * | |
| --- | --- | --- | --- |
| | | after 7:00 h | after 72:00 h |
| Holtz 1 | 9.2 | - | - |
| Holtz 2 | 45.9 | 3.3 | 43.8 |
| Weierbach 1 | 45.1 | 8.8 | 61.8 |
| Weierbach 2 | 106.3 | 5.3 | 41.4 |
| Colpach | 1903.3 | 12.6 | 64.1 |

* expressed as percentage of the precipitation amount [%]

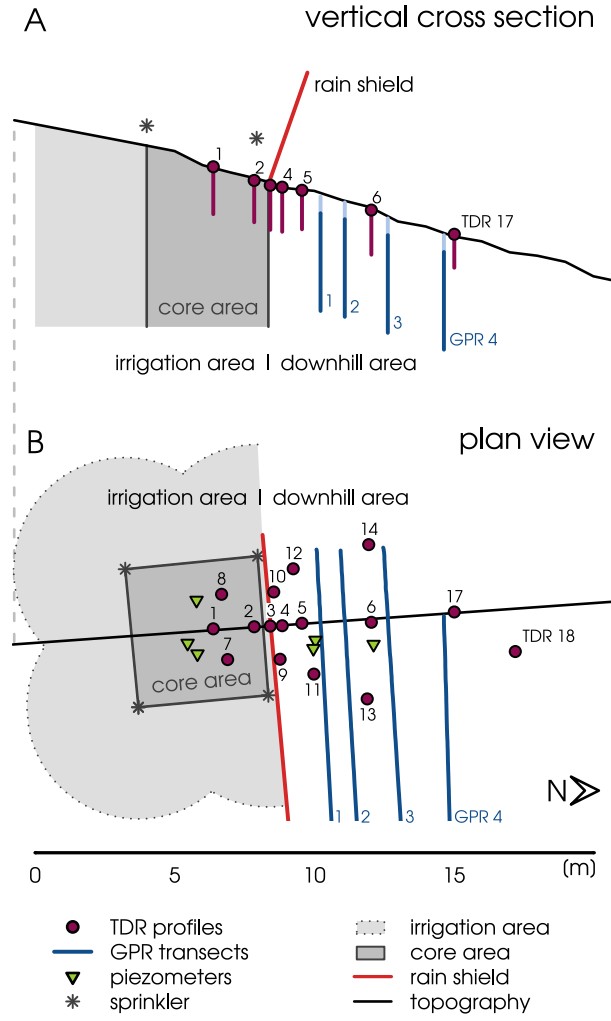

**Figure 4.** Vertical cross section (A) and plan view (B) of the experimental setup and sprinkler array. Location and depth of the TDR profiles and GPR transects are given in purple and blue, respectively. The black line along the central TDR transect in B marks the vertical cross section depicted in A.

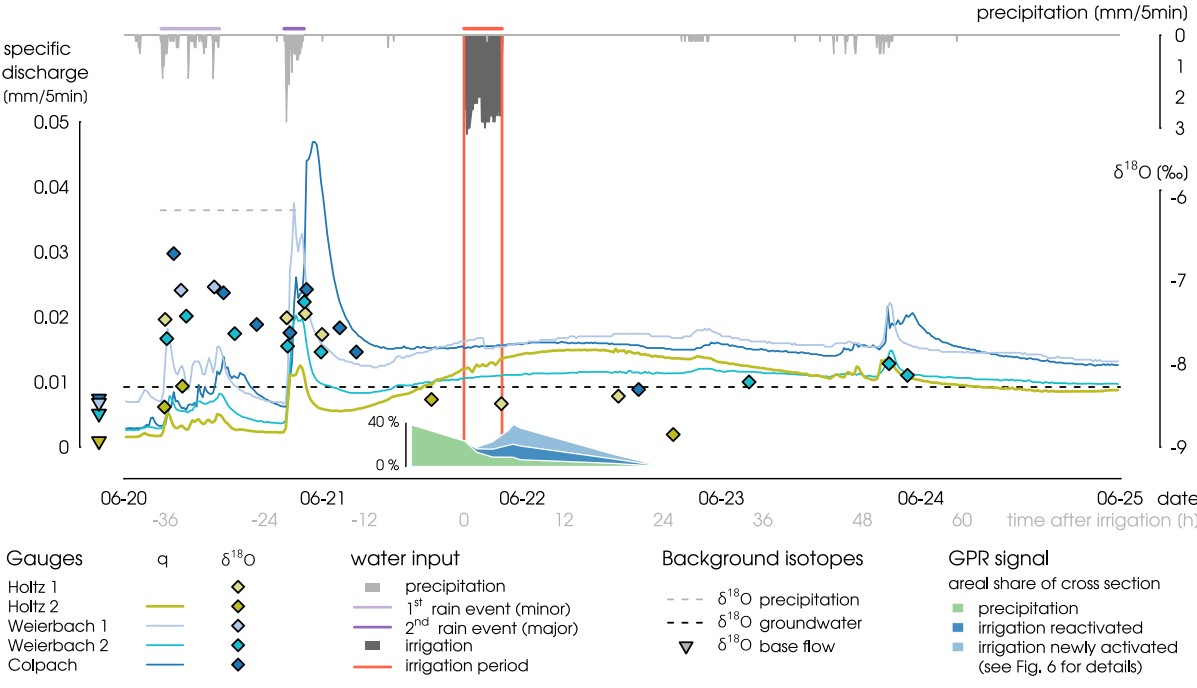

**Figure 5.** The figure shows the natural storm event on June 20, 2013 in the Colpach River catchment and the local intensity of the irrigation on June 21, 2013. The hydrographs below show the discharge response of four nested catchments (solid lines), in combination with the dynamics of the $\delta^{18}O$ isotopic composition of the surface water (dots). The isotopic composition of the groundwater (annual mean) and the precipitation (daily values) are given by the dashed lines. Furthermore, the dynamic response of the GPR signal to natural and artificial rainfall is given in green and blue. While the first minor rain event caused only weak response, the second event caused a double-peak discharge response in all sub-catchments. The irrigation experiment took place 19:22 h after the rain event.

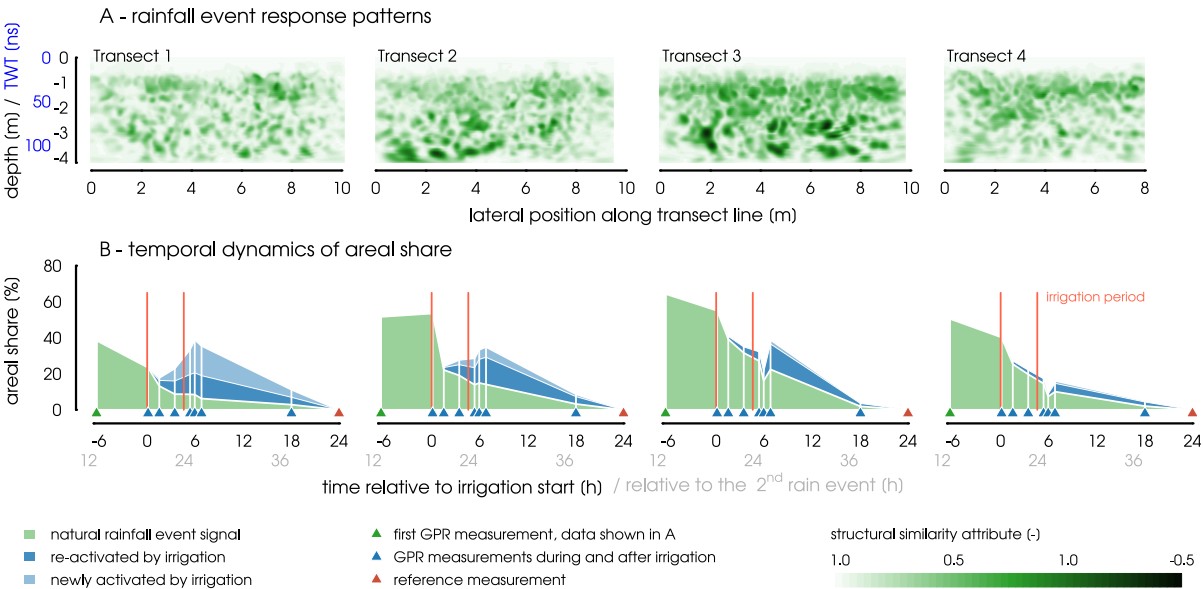

**Figure 6.** A: 2D GPR data showing the subsurface response patterns caused by the natural rainfall event. The data show the structural similarity between the first GPR measurement (approximately $6{:}30$ h before irrigation start and $12{:}52$ h after the $2^{nd}$ rainfall event) and the last one. Low values of structural similarity are interpreted as high changes in soil moisture. B: Temporal dynamics of the areal share of active regions attributed to the natural rain event and the irrigation. Activated regions were identified by a structural similarity attribute of less than $0.85$. Data was interpolated linearly between the measurements for visualization. The measurements shown in A show the data used to calculate the first data point shown in B.

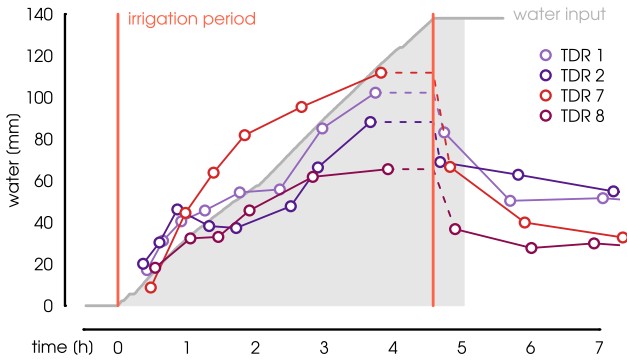

**Figure 7.** Water balance of the top $1.4$ m of the soil column for the four core area TDR profiles. Dashed lines indicate the storage increase at the last measurement before irrigation ended. The variability between the four profiles show the high heterogeneity and cause uncertainty regarding the average mass balance of the core area.

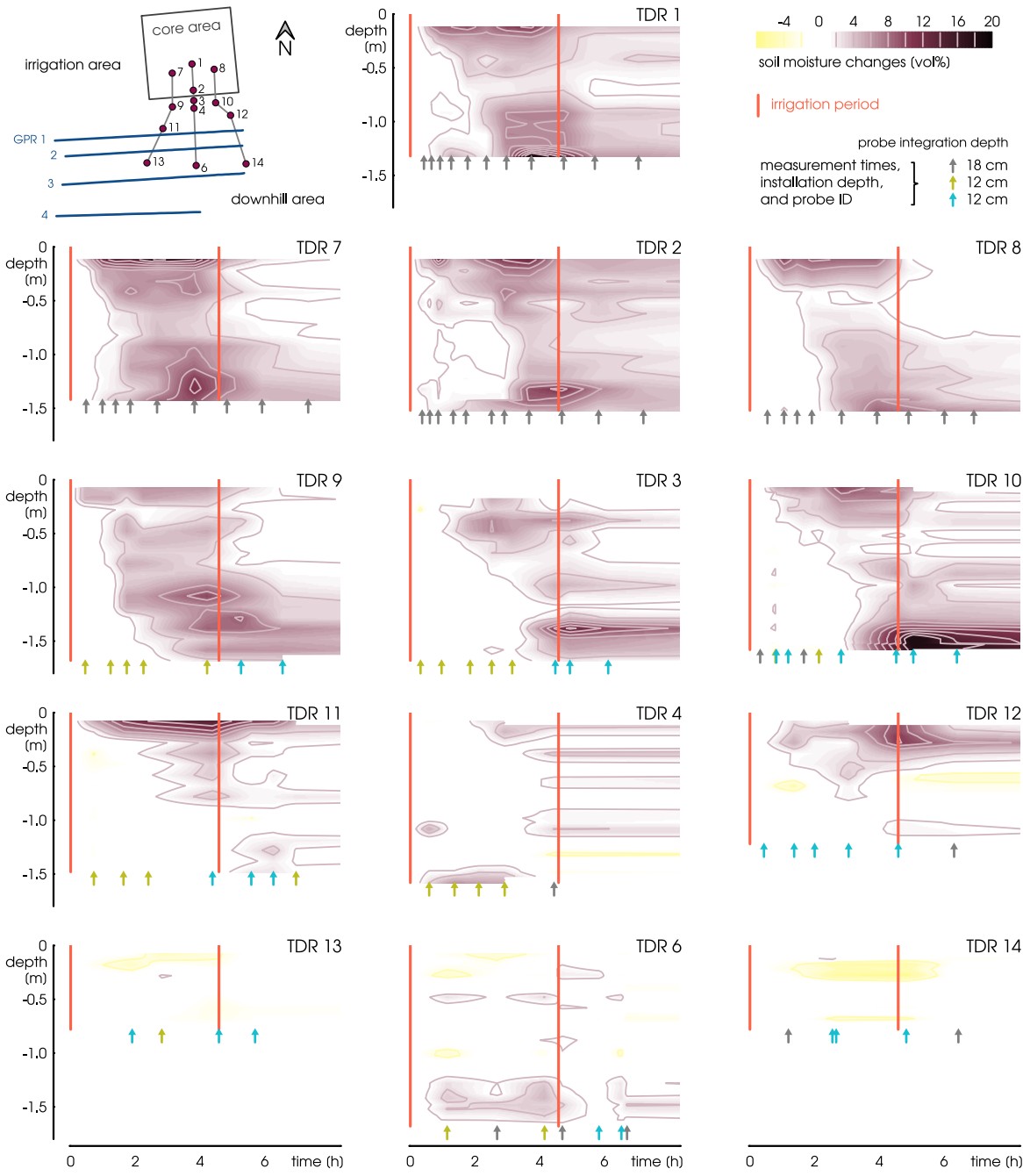

**Figure 8.** The soil moisture data measured at the TDR profiles at the irrigation site, showing the soil moisture dynamics in depth. The top four plots show all four core area profiles, columns are arranged according to the three diverging transects in downhill direction. Rows are approximately at the same contour line. Measurements were taken at $0.1\,\text{m}$ increments. While data analysis was based on non-interpolated data, soil moisture measurements were here interpolated linearly for better visualization. The plots cover the time from irrigation start until $9{:}00\,\text{h}$ after irrigation start to focus on the first soil moisture response. Arrows indicate the measurement times and installation depth of each TDR profile. Time is given in [h] after irrigation start.

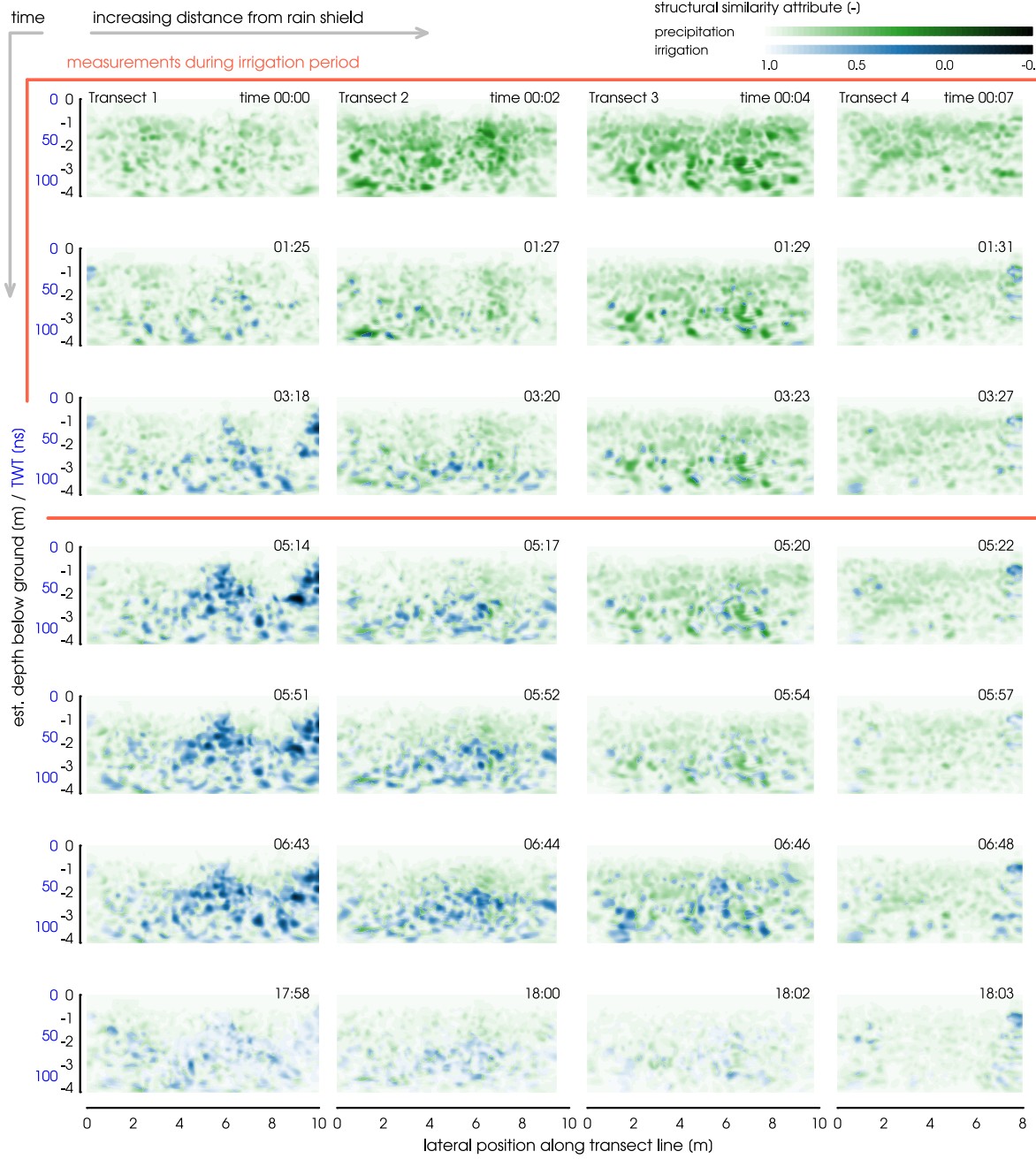

**Figure 9.** Structural similarity attributes calculated from time-lapse GPR data. All measurements were referenced to the last one 24:00 h after irrigation start, indicating changes in the GPR reflection patterns associated with soil moisture changes. A structural similarity attribute value of 1 indicates full similarty, lower values signify higher deviation from the reference state. Water from the preceding natural rain event (green) was identified by constant or increasing structural similarity attributes. Water from the experimental irrigation (blue) was identified by decreasing values after irrigation start by more than 0.15. Within one column the rows give a sequence over time (after irrigation start). Columns proceed downhill, with increasing distance from the rain shield.

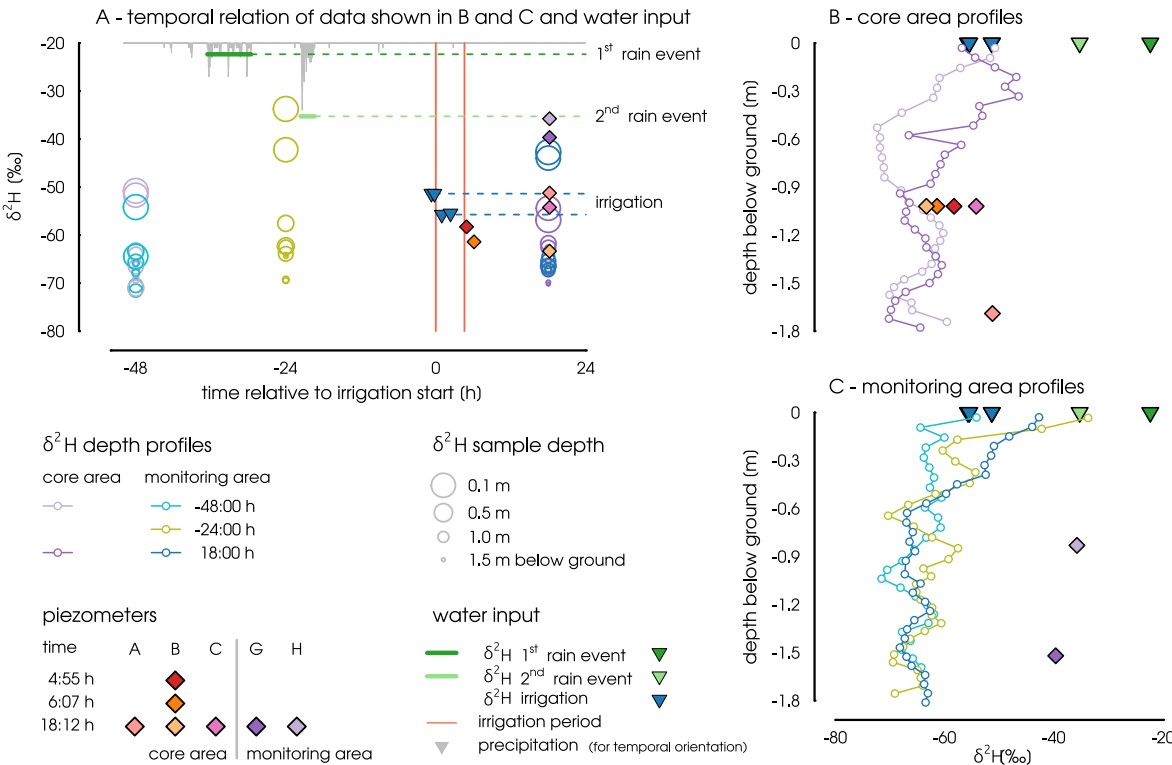

**Figure 10.** Stable isotope data from precipitation, irrigation, piezometers and pore water samples. A: Temporal dynamics of pore water and piezometer $\delta^2 H$ in relation to water input by precipitation (bulk samples) and irrigation. Pore water data is shown only for the depths of piezometer filters (compare with panels B and C for depths) and the top soil (0.1 m below ground). The graph shows the direct impact of the water input (dashed lines) on the pore water isotope composition of the top soil. B and C: Pore water and piezometer data over depths, separated by core area and monitoring area. Water input stable isotope data is indicated at the soil surface.

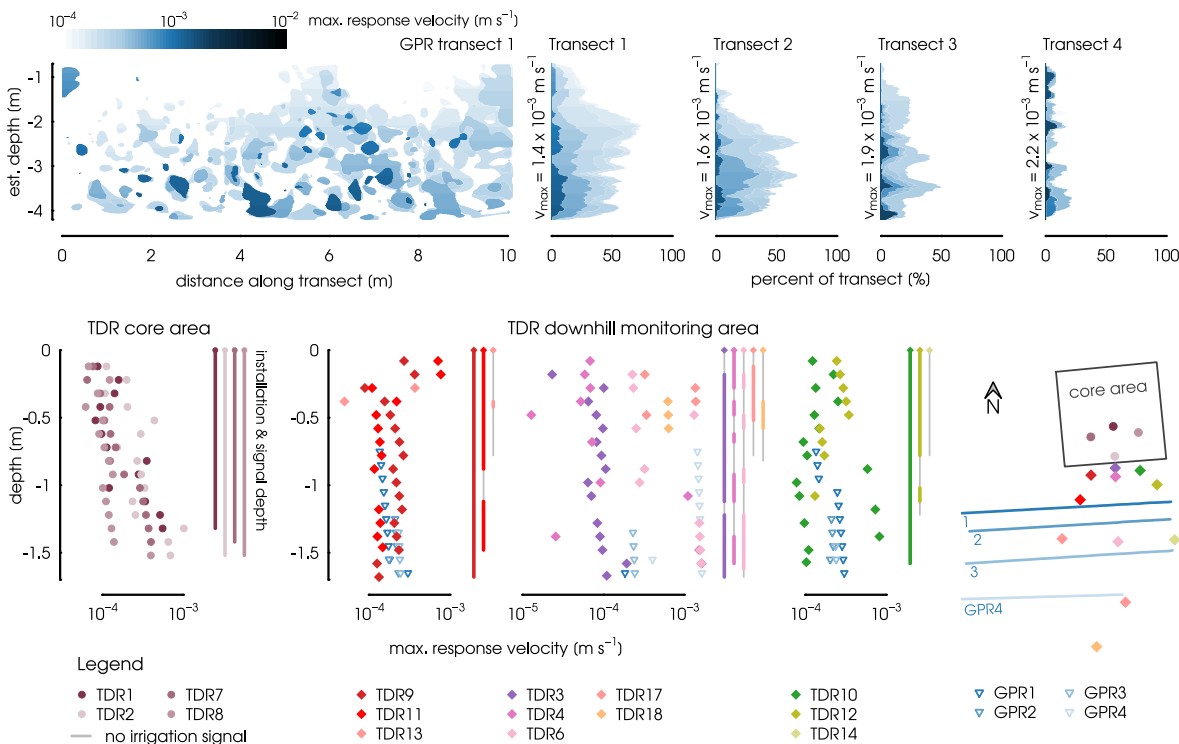

**Figure 11.** Top: Depth distributions of response velocities calculated for the four time-lapse GPR transects. The blue scale indicates the response velocity calculated from the time of first arrival for each pixel. White areas did not show any irrigation water signal. The left plot shows the 2D results for GPR transect 1. The margin plots on the right show the same data accumulated to 1D depth profiles for all four GPR transects. Bottom: Response velocities calculated from TDR measurements at the core area (strictly vertical), and the downhill monitoring area (vertical and lateral). The three plots showing the TDR profiles at the downhill monitoring area are sorted according to the three diverging transects. Lines within the right margins of each TDR plot show the installation depths of the TDR profiles. Grey sections indicate depth increments that did not show a change in soil moisture. Additionally, GPR-based velocities derived from $0.5\,\mathrm{m}$ wide sections of the GPR transects close to the TDR profiles are shown.

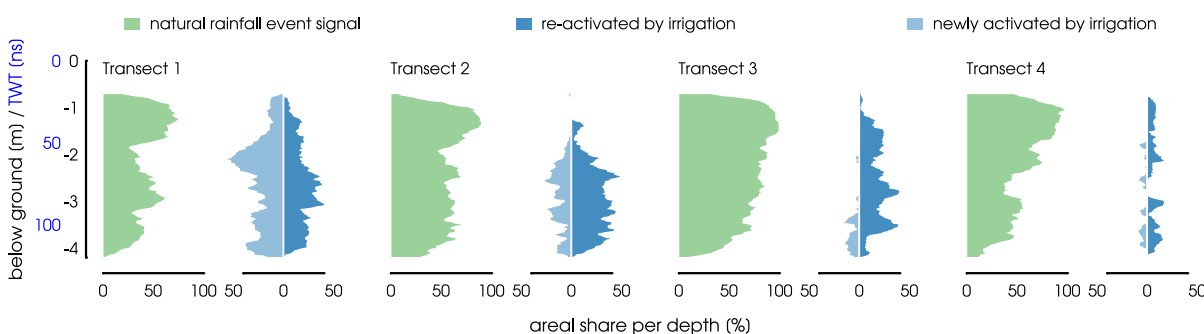

**Figure 12.** Areal share of activated regions per depth for the natural rainfall event and the irrigation. Areal share of activated regions for the natural rain event were calculated from the first measurement only (also depicted in Fig. 6A). Values for the irrigation experiment were accumulated over all GPR measurements, counting every pixel that had been activated after irrigation start. Here, we distinguish between pixels that had been active before irrigation start and were re-activated again, and pixels which have not been active previously and were newly activated by irrigation.