# Peer review of "Form and function in hillslope hydrology: characterization of subsurface flow based on response observations"

_Hydrology and Earth System Sciences, 2016_

## Referee Comment (RC1) · Anonymous Referee #1 · 3 Jun 2016

**General comments**

This paper presents very interesting data from a plot/hillslope scale sprinkling experiment combining TDR observations with time-lapse GPR to investigate soil moisture dynamics. Sadly, the promised (in the title) investigation of subsurface flow and the link to catchment scale is not convincingly presented, interpretations are often overdrawn and not supported by the results, and the paper currently reads more like a patchwork of data and ideas and not like a well aligned story. During the read, it felt as if the authors were not sure of what to present and what the key message of the work can/should be. The data is visually very well presented, but the actual analysis of the experimental data remains poor, eventually supplying temporal evolution of soil

moisture (although the method combination gives an interesting depth distribution to 4 m below surface) and flow velocities for the first respond. Moreover, these velocities are not really used. From a water balance perspective it is interesting, if the occurring amounts of PF can be, by any means, linked to the generated runoff volume in events. I also think that the inclusion of one event based hydrograph separation is a little bit thin.

Stated by authors, the runoff ratio of 4% during the rapid peak (that consists of event water) cannot be explainable by direct rainfall on saturated areas. Nevertheless, the experimental observations and the fact that rapid flow downslope of the sprinkling site can be detected do not, in my opinion, convincingly explain the rapid stream response that is observed during rainfall in the study catchment. And this is why: I think that the method of the sprinkling experiment, with the use of a sprinkling intensity of 30mm/h, is simply not appropriate for investigating if preferential flow reaches the stream and generates the hydrograph under natural conditions.

I do understand that the purpose was to initiate the preferential flow with this rate, but I disagree with the authors that any explanation for the stream response can be done in this way. Such high intensities will of course lead lateral flow out of the sprinkling site as soon non-vertically oriented preferential flow paths are filled, or some low permeable lense or layer is reached. It would be interesting to report the natural rainfall intensities in the area and for the particular shown runoff event. As only a total of around 20 mm occurred. I am wondering if natural intensities would be indeed able to establish, supply, and connect a preferential flow system from the hillslope to the stream under unsaturated conditions on that time scale. This would be very important, since initiation of preferential flow strongly depends on intensities. I do not doubt that PF flow might be active at the hillslope and connecting to the stream in events with 30 mm/h rainfall intensity over hours, but when do such events occur? I think that there are no data and results provided in this manuscript that showed that preferential flow at natural rainfall intensities, amounts, and duration can explain the observed behaviour. Furthermore,
early work of McDonnell and various follow up studies showed that much of the water in preferential flow actually is not event water. This adds further doubt to the authors' interpretation, since the observed peaks in the stream in the area are dominated by event water.

Leaving the interpretation of the catchment scale interpretation and the lateral preferential flow aside, the paper does not provide much novelty from a process perspective. The depth distribution data of soil moisture is interesting from methodological perspective, but work on this was published previously (see the Allroggen references). The discussion about the processes is not supported by the results of the work, and the methodological discussion is overly long, repetitive, lacks nearly any citations, and I was not struck by a major scientific message. That just leaves a case study of a sprinkling experiment, which is - in the current form - not sufficient to be published in HESS. That said, the dataset is very interesting and could supply interesting insights, but a clear idea of what should be presented, and what conclusions can be supported by the data is needed. Following, an in-depth analysis is needed that goes beyond the very nice visual presentation of soil moisture data and calculated flow velocities.

Detailed comments: The syntax is often awkward (except in the introduction) and could highly benefit from a very careful revision (or editing by a native speaker). P1L10: "hillslope-scale connectivity". There is no evidence provided for connectivity. There is evidence that some water moves downward from the sprinkling site. This neither indicates connectivity in this case nor connectivity under natural conditions. P1L11-12: "These processes" is somewhat vague. P2L13: Change: "is on" to "was on" P2L26: Change: "At minimal..." to: "GPR provides..." P3L7ff. I am not sure, if introducing multi-modal transit time distributions here and in the following parts is beneficial for the manuscript. If the authors introduce it and eventually mention it at several points, they should start calculating them for the mentioned events, rather than doing a simple two component end member mixing (although this seems not to be possible with the isotope sampling). References are needed after "flow process". P3L16ff. References
for the claims are needed. Why vague? The research need that is outlined here, is not convincingly derived and link to the majority of the introduction. The introduction exclusively deals with preferential flow processes, before some double-peaked hydrograph behaviour is mentioned. After that, the research needs is only related to these hydrographs. As mentioned earlier, I miss the well thought out story line. I also missed references to important connectivity work, since the connectivity (hillslope-stream) part seemed to be an important part of the paper. P6L10: What is the uncertainty that is associated with this very simple hydrograph separation, especially when accounting for the high variability of stable isotopes in rainfall? P7L3: Please provide the relative location of the irrigation site, when hillfoot is 0 and hilltop is 1. What is the length of the hillslope? Language and syntax are improvable. P7L10: What is the return period of the sprinkling event (intensity, amount)? P7L11: Here it is stated that it is not intended to mimic natural conditions, but then the authors used the results to infer flow processes under natural conditions. This does not fit together. P7L17ff: Information about size and height above ground of the samplers are needed. What were the intensities of the natural rainfall events? This could be a good comparison to the experiment. P9L26-28: "The..." Better delete? Not needed. Another suggesting: better not start sentences with "Figure 2 shows/presents/" or "This/that is presented in Figure 2"; rather report the finding or what the reader show see and then reference the figure. P10L1: "Similar". So what is the difference to the cited work? P10L14: "suggested" Is certain or uncertain? I am also wondering where the separation between mobile and stagnant water was made? P10L31: Why was the 2% value chosen? P10L4ff: "were calculated for the entire depth": Do you mean, that if you give the value for 10 cm it is the velocity for 0-10 cm, if you report the velocity for 20 cm depth, it is the velocity from 0-20cm? P10L9: "main stem". Delete? P12L13ff: How are these hydrographs separated? What method was chosen? This needs to be mentioned in the methods. What are the uncertainties associated with the isotope hydrograph separation? Why was only one event chosen for analysis? P13: Strange font size/type of "20 mm" in the figure captions of fig.3. P13L2: 0.5 L. This was over the whole experiment? So more or less unimportant in

**HESSD**
the water balance. P13L2 "relatively". Please provide a objective description, rather than using subjective terms like small, relative, best with numbers. Do that throughout the manuscript. P13L8: Can these 20% be the uncertainty of the water balance? Or what is the actual uncertainty and how does that influence the results and interpretation? I am not sure what "average storage deficit of 20%" means. P14: Fig4. Uncertainty needed. P15L4-7: Move to methods. P15L13: "generally low dynamics". Please change P15L31: "all fast dynamics" P15L32-33: Is there a possibility to differentiate if it is in the matrix or left the monitored area? How about working more with the water balance? P17L2: This is more discussion of processes then results. How was this inferred? It seems like a statement, but not supported by data. There is no evidence for this. P17L4: Again, there is no proof for this. It is just speculative. How deep below the soil surface is the bedrock? P18: Can velocity distributions be estimated? They might be more interesting/informative than just the first response time. One could see how the majority of the water moves, as the response time of the preferential flow should be somehow consistent with the time of the hydrograph response, especially if the preferential flow should explain the delivery of event water to the stream. P22L6: This is commonly observed. Soil moisture variability increases with decreasing moisture contents P22L12: "high portion of mobile water". Please quantify. This is important P22L13: "Velocities over..." But this only relates to the fastest component. The water itself should travel with a velocity distribution, which was not estimated. I would argue that the median or average speed to the "mobile" water (the matrix water is mobile too) is

some preferential flow out of the sprinkling site occurred. P22L30: Every hydrograph is likely a result of multi-modal distribution of travel times..... P23L20: How were the riparian zones and wetland patches delineated? What is the uncertainty? Is there a contraction/extension during events that could change this? P23L22ff. Again, there is no evidence shown in this work that there was any connectivity between hillslope and stream. This is even more true when natural conditions are considered. P24L26ff. This methodological discussion is overly long, repetitive, and lacking references to work of others. P24L27-P25L3: Here I disagree. As said, the experiment was not designed to mimic natural conditions. This is completely fine, since it seems to me that the original idea was to employ the GPR system. Nevertheless, the setup does not allow inferring natural flow processes that explain hydrograph response. P25L10: Explain: "sufficiently steady state" P25L24: Sat. conditions are most likely occurring at the bedrock interface. Technically the periglacial layers are not "soils" anymore. Try to have a look at the work of Uhlenbrook et al. on periglacial slopes. P27L31ff. This seems to be unnecessary - no important point was made in this section. P29L4: The resulting connectivity was not shown and remains highly speculative. P29L5ff: This is speculative. Furthermore, the link to transit time distributions is somewhat strange. P29L20: Again, I think this was not shown here.

---

## Referee Comment (RC2) · Anonymous Referee #2 · 9 Jun 2016

General comments: This paper presents an interesting experimental approach, combining different types of measurements and techniques (soil moisture monitoring, time-lapse GPR, discharge and tracer measurements) to investigate mechanisms of rapid subsurface flow. Such combinations are often very promising as they can yield new interpretation of observations and new insight. The visual presentation of results, i.e. the figures, is very well done, the figures are informative and mostly well structured. The language is mostly suitable, with sentences sometimes being quite long. Here a language revision would certainly help to make the text more pleasant to read. However, I do not think that the authors achieve their objective of investigating and providing new insights into rapid subsurface flow processes. As they themselves state in the Conclusions, the link between processes and structures is still missing, many statements remain speculations and thus, an exciting new insight is not really presented. The experiments were no doubt labor-intensive and comprehensive, and I can understand that the authors would like to present them in their entirety. I have the impression, though, that the authors were not quite sure about the focus of the manuscript. Title and abstract speak of rapid subsurface flow and identifying preferential flow features whereas in the introduction and then later in the discussion the focus shifts to double-peak hydrographs as if the explanation of these had been the aim of the paper. The abstract does not indicate this. In-between also (multi-modal) transit-time distributions are mentioned several times but surely the experiment was not designed to investigate transit time distributions. The discussion is lengthy and focuses on methodological aspects. At the moment I doubt that the manuscript presents sufficient new data and findings beyond speculations to justify a publication. Maybe it would be an option to combine this manuscript and the mentioned companion paper by Jackisch et al as they present the missing link between processes and structures as stated in the Conclusions, last paragraph.

Specific comments: - P. 2, L 3: preferential flow and rapid subsurface flow are not per se synonyms - P. 3, L 12: if the explanation of double-peak hydrographs is indeed the motivation, this should be mentioned in the abstract - Section 2.1/2.2 and 3.1: I was a little confused that first it seemed to be a study focusing on the irrigation experiment whereas later on also longer-term behavior is described as result. I would only focus on description of the event and potentially, move descriptions of the general hydrological behavior of the catchments to the site description. - Section 2.3.2: I had difficulties to exactly understand the TDR measurements with access tubes and would recommend to describe the soil moisture monitoring more clearly. Could you insert the prongs of the moisture probe via the access tubes in different depths? - P. 8, L 1: predominantly - P. 8, L 6: which three TDR probes? Unclear - Fig. 2: I would not show the location of the piezometers here as you do not discuss any results. That may only be confusing. You can still mention in the text that you installed them but they measured any transient

water tables. Also, it would help if you could indicate the three diverging transects in the figure. - P. 9, L1:...transect of 6 piezometers "intended to observe transient water levels"... - P. 9, L 28-29: add numbers of soil moisture profiles here to make sure it is clear to the reader which profiles you are referring to - P. 13, L 1: intensity of irrigation stated here is different to intensity stated in the methods section - P. 13, L 1: when did surface runoff cease? - P. 13, L 4-5: I am not sure what is meant with "overshoot". Is this shown in Fig 4? Please explain more clearly. - P. 15, L 12: I would recommend to make the way times are stated consistent between text and figure (3.3 hours vs 03:18) - Fig. 5: where is the map for TDR 5? And the decreases in soil moisture shown e.g. in the lowest row of profiles is not really discussed in the text. - I do not fully understand how the authors distinguish in the GPR maps between natural event water and irrigation water. Can that be done with the GPR signal? Please explain in a little more detail. - P. 17, L 2-3: is this a speculation or really supported by the results? - P. 17, L 5-6: maybe better say "water that was supplied from upslope areas was more important..." - Fig. 6: there seems to be one map missing of transect 4 in the lower right corner where the legend is located? - P. 22, L 13: agreed instead of fitted - P. 22, L 24-25: again, is this a speculation? - P. 22, L 30: here now it sounds as if the investigation of double-peak hydrographs is the main objective of the paper. This comes as a surprise as it was not mentioned neither in the abstract nor in the title. - I am not sure that from these measurements of soil moisture at discrete points in time lateral flow can really be inferred. - P. 24, L 28-33: I do not think the reasoning here for the high input rates in convincing. I wonder how useful the observed patterns are for more realistic rainfall conditions. - P 26, L 11: soil moisture should be used with caution as indication for flow, and this has been discussed in the literature before - P 26, L 19-24: this is not new - Conclusions: I would not mention transit-time distributions here. That surely goes beyond the focus of the paper and may be a stretch to infer. - P 29, L 14: stable

---

## Referee Comment (RC3) · Anonymous Referee #3 · 14 Jun 2016

General comments. This study is designed to investigate subsurface flow processes, and in specific preferential flow, at two different scales. The authors chose a multimethod approach including quantitative soil moisture measurements using TDR (time domain reflectometry) and qualitative soil moisture measurements using GPR (ground penetrating radar) during an irrigation experiment at the hillslope scale. The experiment was complemented with common hydrometric and tracers techniques (interpretation of the rainfall runoff graph and stable water isotopes of rainfall and runoff) at different catchment scales. The study addresses an important topic in the hydrological sciences. The manuscript is clearly structured following common scientific standards. The data are presented in (mostly) clearly arranged, high quality figures. The combination of TDR and GPR measurements provide a very promising approach to monitor subsurface processes. While this approach is relatively novel, it bears a number of uncertainties and disadvantages, which are discussed at length (although the methodological discussion lacks a bit in references). However, I do have some concerns. First of all, clear objectives were not formulated in the beginning of the manuscript and it follows that the rest of the story misses a central theme. I do understand though, that the catchment scale analyses revealed that the system responded with a double-peak hydrograph to water input with the first, steeply and guickly responding hydrograph consisting of a mixture of new and old water and the second, dampened hydrograph consisting dominantly of old water. The results from the irrigation experiment were then used to explain the hillslope processes that may potentially lead to this catchment scale response. This leads to my second and third concerns: (2) I feel that only one hydrograph separation to calculate the event and pre-event water fractions of a catchment is not representative. Many studies have shown in the past that the event water fraction may vary largely in one catchment depending on various factors, e.g. water input and/or antecedent moisture conditions [e.g. Munoz-Villers & McDonnell, 2012]. Hence, I suggest including more data and/or an uncertainty analysis to add representability to the HS. And (3) the authors clearly point out that they did not intend to mimic natural conditions with the irrigation experiment, however, they use the results to explain the natural response at the catchment scale. While this shortcoming is mentioned in the methodological discussion, I don't feel that this justification is sufficient to link the observed processes in the irrigation experiment to the natural conditions at the catchment scale. At last, I feel that the novelty of this study is not clearly conveyed to the reader. I think it is the combined use of TDR and GPR measurements which bear a large potential to move forward in subsurface flow process understanding. However, this message needs to be presented more clearly. Specific comments. Objectives and experimental approach: I think that this subsection needs a bit more structure and/or more precise phrasing. Maybe name all objectives at the beginning of the paragraph and then list the approaches that you chose to address them in addition to the explana-
tions why you chose said approaches (advantages vs disadvantages of others). Page 3, line 24: the role of what? Please add information because otherwise this sentence and subsequent paragraph come a bit out of the blue, i.e. raise further questions, e.g. why is it necessary to use a multi-scale approach? Line 25: What are the conventional hydrological methods that you mention? I suppose the TDR measurements that you chose as approach in your experiment? Please add this information to the text or rephrase. Line 27: you only propose GPR, why do you not mention the other techniques? (Maybe use this 3 comments to elaborate your objectives.)

Methods. Hydrological response monitoring: Page 6, line 22: How exactly was rainfall water sampled for isotope analysis? Was a sequential sampler used or is it a bulk sample? Please add information. Line 24 (and everywhere else in the ms): I suggest to use "hydrograph separation" instead of "mixing model", the latter implies that an EMMA was performed which is not the case.

Process monitoring: Page 8, line 3: Please add information why different TDR sensors were used. Page 8, line 6: Which 3 TDR tubes were used? Sounds like they were mentioned before which they aren't. Please rephrase or add information for better understanding.

Results. Figure 3: there is too much information in this Figure. I suggest moving some of the information to e.g. a separate table. For example, the information of the catchment sizes and the % of the precipitation amount should be placed somewhere else, maybe together in one table. You may mark the 7th and 72th hr in the RR-Graph and refer the reader to the table. Ii also remains unclear to me why the information about the irrigation experiment (i.e. the mean structural similarity attribute) are placed in the rainfall-runoff and chemographs of the headwater catchments. Which y-axis does the mean structural attribute refer to? And what do the vertical bars mean? Please provide this information in the legend. I suggest considering making two different graphs, for example below each other in the manner of Figure a) catchment scale and b) hillslope scale. This should help to understand this figure faster and identify/extract important

HESSD
information more easily. Please also provide information about where groundwater was sampled for d18O, e.g. in section 2.2. Page 12, lines 7: Since there are several peaks in Figure 3, it would make it easier for the reader if you added the date of the described rain/hydrograph event. Page 13, line 5: What is an "overshoot" in mass recovery? Please add a bit more information to explain this. Page 14, lines19-20 and Figure 5: The weak signal of soil moisture dynamics in TDR 13, 6 and 14 is apparent in Figure 5. While an explanation is provided for the signal in TDR 6, I am missing explanation for TDR 13 and 14. How do you explain this? I suggest including this in the discussion section 4.1. Figure 5: Soil moisture change is interpolated over time for visual reasons (according to figure caption), I think it would be helpful to add this information somewhere in the methods section (e.g. 2.4.1. TDR data analysis).

Discussion. Process interpretation: What is the novelty about this study? I suggest elaborating this e.g. add a paragraph to emphasize this a bit more clearly. Page 22. lines 22 – 25: I do not understand the link between delayed signal in the intermediate depth and the network of preferential flow paths. Please add a bit more text. Page 23, lines 22 - 29: So, does that mean that the first peak is generated by overland flow (as observed by Wrede et al.) AND shallow subsurface flow? Or is all of the event water conveyed via subsurface flow? Please express your interpretation a bit more clearly. Technical corrections. Figure 4: Can you move the y-axis name and unit away from the numbers, e.g. to the very top or remove 140 and insert 120 instead? Typing errors: Throughout the whole manuscript use either one of Fig. X, Fig X or Figure X. Page 3, line 8: empirically Page 9, line 23: Add "of" in "In case of the vertical profiles..." Page 12, line19: use "to" instead of "of" in "... up to 67.6 % to the event runoff..." Page 13, Figure 3, legend: "attribute" instead of "atribute" Page 14, line 3: delete "to" in "... returned to similar to initial conditions..." Line 15: insert "(" in "Fig.5)" Page 15, line 2: use plural instead of singular: "flow paths" Page 19, Figure 6 caption, line 1: time-lapse GPR Page 22, line 13: I suggest to use "...match..." instead of "...fitted with..." Page 24, line 22: delete redundant "the" Page 27, line 17: substitute "is" with "are" in "The dynamics of preferential flow are often characterized..." Page 29, line 14:

---

## Author Comment (AC2) · 3 Aug 2016

**Response to comments by reviewer#2**

We highly appreciate the thorough review and constructive comments of the anonymous reviewer. We are happy to implement and discuss the cogent remarks on data analysis and presentation of our findings and developed a concept to tackle the basic criticism regarding the story line and the conclusiveness of our results. This concept is based on the comments by all three reviewers and provides a common theme as well as overarching yet standalone story lines for the two companion papers. We will shortly present the main idea of the new concept, the revised hypotheses as well as the restructured story line of the first manuscript (MS1). As these improvements are relevant for our response to the comments by all reviewers, this part is included in all replies. Afterwards, we will address the general and specific comments made by reviewer#2, with our replies inserted in the original review.

**Conceptual framework**

To better elaborate the methodological aspects of the study and to provide a common theme for the two companion papers, we want to employ the concept of form vs. function. The original term 'form follows function' was first established in architecture and soon was adopted by biologists. It refers to the idea, that form and functionality are closely correlated, influence each other and co-evolve. We suggest to transfer the same idea to hydrological systems. This allows us to separate and analyze their two main characteristics: Their form, which is equivalent to the spatial structure and static properties, and their function, equivalent to internal responses and hydrological behavior. While this approach itself is not particularly new to hydrological field research, we want to employ this concept to explicitly pursue the question of what information is most advantageous to understand a hydrological system.

Accordingly, we developed different categories to organize and describe the data presented in the two manuscripts: Structural data summarizes all sorts of data which focus on direct exploration of form, e.g. soil cores. Response dynamics, on the other hand, are observations of function. They represent processes deprived of their spatial context and include soil moisture dynamics and discharge responses. In between these two categories are flow-relevant structures and response patterns, which may contain information on both, form and function of the system.

In the presented study, we apply this concept to subsurface flow within a hillslope. The first part of the study (MS1) methodologically focuses on function: We observed response patterns and dynamics from a natural rainfall event and during an irrigation experiment. The results are used to infer hydrological processes and the spatial organization of the monitored system. Based on these findings, the informative power and conclusiveness of the data will be discussed.

The second manuscript (MS2) focuses on form and starts off with a thorough structural exploration of the subsurface. It then proceeds towards observations of flow-relevant structures and response patterns and analyzes the information gain along this path.

**Hypotheses and story line**

The hypotheses of both manuscripts can be aligned according to the form/function framework and will clearly be stated at in the beginning. The hypotheses of MS1 will focus on the potential of response observations for hillslope hydrological field research and the application of time-lapse GPR measurements in this context:

- **H1.1** Response observations (discharge, TDR & GPR data) are sufficient to characterize subsurface flow ywithin the hillslope.

- **H1.2** Response patterns can be used to deduce flow-relevant structures in the subsurface.

- **H1.3** Time-lapse GPR measurements visualize subsurface flow dynamics and patterns and can replace hillslope trenches.

The story line will be streamlined and arranged along these hypotheses. This will help to make the manuscript easier to follow and to better elaborate the important and novel key points of our study. In the following, new or restructured sections are marked in brown.

**1 Introduction**

- Concept of form and function: form and function are the main defining features of a system. Applied to catchments, the concept describes any kind of spatial structure (from topography to macropores) as form, and the sum of all processes defining the hydrological behavior of the catchment as function. Both strongly influence and determine each other.

- Example subsurface flow at the hillslope: structures and heterogeneity control flow patterns and velocities and thus the occurrence of preferential flow. Which one is better suited to characterize a hillslope, form or function?

- Focus on function: What does it need to describe subsurface flow and preferential flow at the hillslope? How to observe subsurface flow processes? What do response patterns and dynamics tell us about form and function of a hillslope?

- Methodological challenge of preferential flow: former approaches at different scales: plot, hillslope- and catchment-scale.

- Hypotheses as stated above

**2 Methods**

- Study site description

- Hydrological response monitoring: Hydrograph and surface water isotopes

- Hillslope-scale irrigation experiment

  - Setup
  - Process monitoring
  - Piezometer isotope sampling

- Data analysis

  - TDR data analysis
  - GPR data analysis
  - Comparison natural event vs. irrigation: Distinguish the signals and calculate areal share of activated cross section
  - Response velocity calculation

**3 Results**

- Response to the natural rainfall

  - Hydrograph and surface water isotopes
  - Subsurface response patterns (green GPR reflection patterns)

- Irrigation experiment

  - Core area water balance
  - Soil moisture dynamics
  - 2D time-lapse GPR
  - Soil and piezometer isotopes
  - Combination of TDR and GPR
  - Response velocities

- Comparison of natural event and irrigation: Areal share and signal strength of GPR measurements before and after irrigation

**4 Discussion**

- Process interpretation

  - Interpretation of artificially induced response observations during and after the irrigation
  - Interpretation of the natural response observations after the rainfall event prior to the irrigation experiment
  - Identification of (flow-relevant) structures from response patterns

- Methodological discussion

**5 Conclusions**

- **H1.1** 'Response observations (discharge, TDR & GPR data) are sufficient to characterize subsurface flow within the hillslope.'

  → Processes can be identified and characterized without any concrete information about spatial structures. However, observations are limited in spatial (and temporal) resolution and interpretations beyond observation scale remain speculative.

- **H1.2** 'Response patterns can be used to deduce flow-relevant structures in the subsurface.'

  → Response patterns allowed to develop a conceptual description of the flow paths network, which is linked to subsurface structures. However, actual structural features, such as the deposit layer or the bedrock interface, could not be located.

- **H1.3** Time-lapse GPR measurements visualize subsurface flow dynamics and patterns and can replace hillslope trenches.'

  → Time-lapse GPR measurements lack the quantitative power and the direct link to structures (obtained by excavation) of trenches. Their spatial and temporal flexibility as well as their non-invasive character, however, are very advantageous for the investigation of highly dynamics and spatially distributed flow processes. Thus, the application of time-lapse GPR measurements in combination with soil moisture measurements are a powerful tool for the observation of hydrometric responses. Depending on the research question, the method can replace labor-intense trenches and even increase the observation density due to its spatial flexibility.

The titles of the two manuscripts will be adapted accordingly. They will be rephrased to emphasize the methodological aspects of the two papers, while keeping the focus on hillslope processes. The final versions of the titles are still subject to discussion. A possible suggestion for the title of this manuscript is:

FORM AND FUNCTION IN HILLSLOPE HYDROLOGY:
IN SITU CHARACTERIZATION OF SUBSURFACE FLOW BASED ON
RESPONSE OBSERVATIONS

We hope to have given a good overview over the anticipated revisions of the manuscript. While the elaboration above was meant to provide the 'big picture', our answers to the first reviewer's general and specific comments will illustrate how this concept will help to mitigate the reviewer's concerns. In the following, our replies are inserted into the original text by reviewer#2 and marked in purple.

**Answers to the general comments**

This paper presents an interesting experimental approach, combining different types of measurements and techniques (soil moisture monitoring, time-lapse GPR, discharge and tracer measurements) to investigate mechanisms of rapid subsurface flow. Such combinations are often very promising as they can yield new interpretation of observations and new insight. The visual presentation of results, i.e. the figures, is very well done, the figures are informative and mostly well structured. The language is mostly suitable, with sentences sometimes being quite long. Here a language revision would certainly help to make the text more pleasant to read.

However, I do not think that the authors achieve their objective of investigating and providing new insights into rapid subsurface flow processes. As they themselves state in the Conclusions, the link between processes and structures is still missing, many statements remain speculations and thus, an exciting new insight is not really presented.

In their current version, the two companion papers were separated by their foci on temporal dynamics on the one hand (first manuscript at hand, MS1), and spatial structures on the other (second manuscript, MS2). Therefore, the discussion of the link between processes and structures was deliberately avoided in MS1 and left to the companion paper.

In the course of the revision, the foci of the two papers will be shifted towards a more methodological perspective. While we will pursue the question of how much information we can get from pure response observations in MS1, MS2 starts off with structural explorations of the subsurface. We subsume this conceptual approach under the term 'form and function', applied to hydrological systems. According to this concept, the link between processes and structures will still be part of the findings of MS2. However, the informative value of observed response dynamics and patterns and the usefulness of structural information will be discussed more thoroughly also in the revised manuscript MS1.

The experiments were no doubt labor-intensive and comprehensive, and I can understand that the authors would like to present them in their entirety. I have the impression, though, that the authors were not quite sure about the focus of the manuscript. Title and abstract speak of rapid subsurface flow and identifying preferential flow features whereas in the introduction and then later in the discussion the focus shifts to double-peak hydrographs as if the explanation of these had been the aim of the paper. The abstract does not indicate this. In-between also (multi-modal) transit-time distributions are mentioned several times but surely the experiment was not designed to investigate transit time distributions. The discussion is lengthy and focuses on methodological aspects. We understand that the story was not well presented in the current version of the manuscript and needs to be improved. Based on the comments of the three anonymous reviewers, we decided on a common theme for the two companion papers and streamlined the story line of MS1. The revised manuscript will focus on the investigation of subsurface flow processes within the hillslope with a strong methodological perspective. The presented hydrographs will still be a minor part of the study to provide a temporal framework the subsurface flow processes observed after the rain event. The discussion of double peaks and (multi-modal) travel time distributions, however, will be minimized or avoided.

At the moment I doubt that the manuscript presents sufficient new data and findings beyond speculations to justify a publication. Maybe it would be an option to combine this manuscript and the mentioned companion paper by Jackisch et al as they present the missing link between processes and structures as stated in the Conclusions, last paragraph.

The companion paper by Jackisch et al. presents many experiments and results which were not introduced in the manuscript at hand and does not include all of the data presented here. The presentation of the entire set of methods, results and conclusions clearly exceeds the frame of one paper. We therefore suggest to revise the two manuscripts. We will clearly elaborate their specific and distinct perspectives, while also allowing slightly more overlap than in the original version so that the links between processes and structures can be pointed out.

**Answers to line-by-line comments**

- P. 2, L 3: preferential flow and rapid subsurface flow are not per se synonyms
  'rapid subsurface flow' will be removed and the use of these terms checked throughout the manuscript.

- P. 3, L 12: if the explanation of double-peak hydrographs is indeed the motivation, this should be mentioned in the abstract
  As stated above, the story line of the manuscript will be revised in a way, that the focus is on the hillslope response. The abstract will be revised accordingly.

- Section 2.1/2.2 and 3.1: I was a little confused that first it seemed to be a study focusing on the irrigation experiment whereas later on also longer-term behavior is described as result. I would only focus on description of the event and potentially, move descriptions of the general hydrological behavior of the catchments to the site description.
  As suggested, the first three sentences of Section 3.1 will be removed and the necessary information will be given as part of the site description.

- Section 2.3.2: I had difficulties to exactly understand the TDR measurements with access tubes and would recommend to describe the soil moisture monitoring more clearly. Could you insert the prongs of the moisture probe via the access tubes in different depths?
  The used Trime Pico IPH probes (IMKO GmbH) do not have prongs, but aluminum plates as TDR wave guides. These plates are mounted on opposite sides of the probe, pressing against the access tube and measuring through the PVC access tube. We will clarify our description of the procedure, possibly also including a sketch.

- P. 8, L 1: predominantly
  Will be corrected in the entire manuscript.

- P. 8, L 6: which three TDR probes? Unclear
  This will be clarified three lines above:
  - before: 'Two versions of the TDR sensor were employed, with either

0.12 m or 0.18 m integration depth.'
- now: 'Soil moisture was measured manually. To enable parallel measurements and increase temporal resolution of the measurements, three probes were used in parallel. These probes differed slightly in their sensor design: Two TDR probes had an integration depth (i.e. sensor length) of 0.12 m and one probe had an integration depth of 0.18 m. '

- Fig. 2: I would not show the location of the piezometers here as you do not discuss any results. That may only be confusing. You can still mention in the text that you installed them but they measured any transient water tables. Also, it would help if you could indicate the three diverging transects in the figure.
  In the revised manuscript we will include the isotope data from water seeping into these piezometers as well as soil core samples taken during installation, which are currently presented in MS2. Due to these changes, the piezometers' positions will be of relevance in the revised manuscript. We therefore suggest to include them in the map of the revised manuscript. We will add lines to indicate the three diverging transects.

- P. 9, L1: ...transect of 6 piezometers 'intended to observe transient water levels'...
  We will change the sentence as suggested.

- P. 9, L 28-29: add numbers of soil moisture profiles here to make sure it is clear to the reader which profiles you are referring to
  Will be added.

- P. 13, L 1: intensity of irrigation stated here is different to intensity stated in the methods section
  This discrepancy is based on the three different monitoring methods we applied: a flow meter, a tipping bucket, and 42 rain collectors. While the flow meter and tipping bucket are considered more reliable with regard to absolute amount, the rain collectors were mainly used to evaluate the homogeneity of the irrigation intensity. We will relate the measurements to each other and correct this discrepancy.

- P. 13, L 1: when did surface runoff cease?
  Surface runoff ceased approximately 20 min after the end of irrigation. We will include this information in the revised manuscript.

- P. 13, L 4-5: I am not sure what is meant with 'overshoot'. Is this shown in Fig 4? Please explain more clearly.
  We will rephrase the sentence and explain the observation: 'All profiles showed a mass recovery of more than 100 % (i.e. higher storage increase than water input at measuring time, see Fig. 4) in the first 60 min of the irrigation period.'

- P. 15, L 12: I would recommend to make the way times are stated consistent between text and figure (3.3 hours vs 03:18)
  Will be done.

- Fig. 5: where is the map for TDR 5?
  TDR 5 showed only very low soil moisture dynamics. It was left out to

leave more space in the whole-page figure for the 'more interesting' profiles. We will try to find a way to put all profiles in that figure.

- Fig. 5: And the decreases in soil moisture shown e.g. in the lowest row of profiles is not really discussed in the text.
  The decrease in soil moisture can be attributed to the fading signal of the natural rain event. However, it is minimal compared to the soil moisture increase observed elsewhere and close to noise level. We will add this information to the manuscript and discuss it accordingly.

- I do not fully understand how the authors distinguish in the GPR maps between natural event water and irrigation water. Can that be done with the GPR signal? Please explain in a little more detail.
  The discrimination was made based on the dynamics of the GPR signal. We assume that the irrigation is causing any (significant) increase in soil water content after irrigation start. This assumption is not per se valid, but in our case supported by the increasing signal strength and the dynamics after irrigation start. On the other hand, measurements taken before irrigation start show all areas activated by the natural rainfall event. Thus, areas where soil water content is decreasing during the irrigation are interpreted as flow paths that were only activated during the natural rainfall event, but not during irrigation. The procedure is described in the Methods section (P. 10, L. 24-28 of the original manuscript) as well as in the figure caption of Fig. 6. We will improve the description and mention the necessary assumptions.

- P. 17, L 2-3: is this a speculation or really supported by the results?
  We clearly see the fading signal after the natural rain event as well as the increase and decrease initiated by the irrigation. Also, the patterns are distributed across the GPR transects. While we cannot specify the size and characteristics of these distributed structures, a groundwater table or flow layer would have caused a clear and more organized signal in a specific depth. In the revised manuscript we will provide more detail on these findings and also discuss the informative power of the observed response patterns. We will also rephrase the annotated sentence to avoid over-interpretation.

- P. 17, L 5-6: maybe better say 'water that was supplied from upslope areas was more important...'
  Will be changed.

- Fig. 6: there seems to be one map missing of transect 4 in the lower right corner where the legend is located?
  The measurement was omitted to fit in the legend while showing the other measurements in bigger size. As described in the text, transect 4 seemed not to be affected by the irrigation experiment much and shows generally low dynamics. The figure is at maximum possible size already and we feel that it is not appropriate to further shrink the size of the subplots. However, we will try to find a different solution or explain the reason for omitting the last sub-plot in the figure caption.

- P. 22, L 13: agreed instead of fitted
  Will be changed.

- P. 22, L 24-25: again, is this a speculation?
  This is the interpretation of the response patterns in both, GPR and TDR results. The 2D GPR reflection patterns clearly show that there is a heterogeneous soil moisture response, but no groundwater table or defined flow layer. The heterogeneity between the TDR profiles as well as the delayed signal in the intermediate depths of the core area profiles also imply preferential flow. We agree, that any interpretations beyond these findings are speculative, and will leave them to the second manuscript where they are backed by structural information. The focus of the revised manuscript will be shifted towards the methodological aspects of interpreting response patterns and dynamics. Thus, we will highlight the informative value of the presented data rather than their interpretation.

- P. 22, L 30: here now it sounds as if the investigation of double-peak hydrographs is the main objective of the paper. This comes as a surprise as it was not mentioned neither in the abstract nor in the title.
  The catchment response was meant as a reference to evaluate the hillslope-scale observations, but was discussed too thoroughly. We will revise the story line, focus it on the investigation of hillslope responses and reduce the discussion of the catchment-scale observations to the necessary and helpful amount.

- I am not sure that from these measurements of soil moisture at discrete points in time lateral flow can really be inferred.
  We agree that discrete measurements in time alone are per se not useful to infer lateral flow. However, our experimental setup allows for a clear separation between observations at the irrigated area and the downhill monitoring are. Due to the lack of water input at the downhill monitoring area, all increases in soil moisture downslope of the rain shield are the result of lateral flow or a combination of lateral and vertical flow. Thus, the spatial arrangement of measurements provides the necessary context to interpret the observed dynamics.

- P. 24, L 28-33: I do not think the reasoning here for the high input rates in convincing. I wonder how useful the observed patterns are for more realistic rainfall conditions.
  In the frame of the anticipated revision, we will restructure the discussion. The general discussion on the experimental design, including the annotated paragraph, and different aspects of soil moisture monitoring in general will be drastically shortened and the sentence deleted.

- P 26, L 11: soil moisture should be used with caution as indication for flow, and this has been discussed in the literature before
  We agree and will rephrase the sentence accordingly

- P26, L 19-24: this is not new
  As mentioned earlier, the general methodological discussion will be drastically shortened. We will avoid such general statements or provide the necessary references, if required. The revised discussion will focus on the (methodological) link between form and function of hydrological systems and the explanatory power of response patterns and dynamics in that context.

- Conclusions: I would not mention transit-time distributions here. That surely goes beyond the focus of the paper and may be a stretch to infer.
The discussion of catchment-scale observations will be reduced to the aspects which are important to evaluate the hillslope-scale interpretations. These aspects do not include transit-time distributions, nor will they be part of the conclusions of the revised manuscript. Thanks for pointing this out.

- P29, L 14: stable
Will be corrected.

---

## Author Comment (AC1)

**Response to comments by reviewer#1**

We highly appreciate the thorough review and constructive comments of the anonymous reviewer. We are happy to implement and discuss the cogent remarks on data analysis and presentation of our findings and developed a concept to tackle the basic criticism regarding the story line and the conclusiveness of our results. This concept is based on the comments by all three reviewers and provides a common theme as well as overarching yet standalone story lines for the two companion papers. We will shortly present the main idea of the new concept, the revised hypotheses as well as the restructured story line of the first manuscript (MS1). As these improvements are relevant for our response to the comments by all reviewers, this part is included in all replies. Afterwards, we will address the general and specific comments made by reviewer#1, with our replies inserted in the original review.

**Conceptual framework**

To better elaborate the methodological aspects of the study and to provide a common theme for the two companion papers, we want to employ the concept of form vs. function. The original term 'form follows function' was first established in architecture and soon was adopted by biologists. It refers to the idea, that form and functionality are closely correlated, influence each other and co-evolve. We suggest to transfer the same idea to hydrological systems. This allows us to separate and analyze their two main characteristics: Their form, which is equivalent to the spatial structure and static properties, and their function, equivalent to internal responses and hydrological behavior. While this approach itself is not particularly new to hydrological field research, we want to employ this concept to explicitly pursue the question of what information is most advantageous to understand a hydrological system.

Accordingly, we developed different categories to organize and describe the data presented in the two manuscripts: Structural data summarizes all sorts of data which focus on direct exploration of form, e.g. soil cores. Response dynamics, on the other hand, are observations of function. They represent processes deprived of their spatial context and include soil moisture dynamics and discharge responses. In between these two categories are flow-relevant structures and response patterns, which may contain information on both, form and function of the system.

In the presented study, we apply this concept to subsurface flow within a hillslope. The first part of the study (MS1) methodologically focuses on function: We observed response patterns and dynamics from a natural rainfall event and during an irrigation experiment. The results are used to infer hydrological processes and the spatial organization of the monitored system. Based on these findings, the informative power and conclusiveness of the data will be discussed.

The second manuscript (MS2) focuses on form and starts off with a thorough structural exploration of the subsurface. It then proceeds towards observations of flow-relevant structures and response patterns and analyzes the information gain along this path.

**Hypotheses and story line**

The hypotheses of both manuscripts can be aligned according to the form/function framework and will clearly be stated at in the beginning. The hypotheses of MS1 will focus on the potential of response observations for hillslope hydrological field research and the application of time-lapse GPR measurements in this context:

- **H1.1** Response observations (discharge, TDR & GPR data) are sufficient to characterize subsurface flow ywithin the hillslope.

- **H1.2** Response patterns can be used to deduce flow-relevant structures in the subsurface.

- **H1.3** Time-lapse GPR measurements visualize subsurface flow dynamics and patterns and can replace hillslope trenches.

The story line will be streamlined and arranged along these hypotheses. This will help to make the manuscript easier to follow and to better elaborate the important and novel key points of our study. In the following, new or restructured sections are marked in brown.

**1 Introduction**

- Concept of form and function: form and function are the main defining features of a system. Applied to catchments, the concept describes any kind of spatial structure (from topography to macropores) as form, and the sum of all processes defining the hydrological behavior of the catchment as function. Both strongly influence and determine each other.

- Example subsurface flow at the hillslope: structures and heterogeneity control flow patterns and velocities and thus the occurrence of preferential flow. Which one is better suited to characterize a hillslope, form or function?

- Focus on function: What does it need to describe subsurface flow and preferential flow at the hillslope? How to observe subsurface flow processes? What do response patterns and dynamics tell us about form and function of a hillslope?

- Methodological challenge of preferential flow: former approaches at different scales: plot, hillslope- and catchment-scale.

- Hypotheses as stated above

**2 Methods**

- Study site description

- Hydrological response monitoring: Hydrograph and surface water isotopes

- Hillslope-scale irrigation experiment

    - Setup
    - Process monitoring
    - Piezometer isotope sampling

- Data analysis

    - TDR data analysis
    - GPR data analysis
    - Comparison natural event vs. irrigation: Distinguish the signals and calculate areal share of activated cross section
    - Response velocity calculation

**3 Results**

- Response to the natural rainfall

    - Hydrograph and surface water isotopes
    - Subsurface response patterns (green GPR reflection patterns)

- Irrigation experiment

    - Core area water balance
    - Soil moisture dynamics
    - 2D time-lapse GPR
    - Soil and piezometer isotopes
    - Combination of TDR and GPR
    - Response velocities

- Comparison of natural event and irrigation: Areal share and signal strength of GPR measurements before and after irrigation

**4 Discussion**

- Process interpretation

    - Interpretation of artificially induced response observations during and after the irrigation
    - Interpretation of the natural response observations after the rainfall event prior to the irrigation experiment
    - Identification of (flow-relevant) structures from response patterns

- Methodological discussion

- – Conclusiveness of function observations without structural knowledge with regard to process identification and the characterization of a system
- – Conclusiveness of function observations without structural knowledge with regard to transferability and regionalization of results
- – Evaluation of GPR as trench replacement

**5 Conclusions**

- **H1.1** 'Response observations (discharge, TDR & GPR data) are sufficient to characterize subsurface flow within the hillslope.'

  → Processes can be identified and characterized without any concrete information about spatial structures. However, observations are limited in spatial (and temporal) resolution and interpretations beyond observation scale remain speculative.

- **H1.2** 'Response patterns can be used to deduce flow-relevant structures in the subsurface.'

  → Response patterns allowed to develop a conceptual description of the flow paths network, which is linked to subsurface structures. However, actual structural features, such as the deposit layer or the bedrock interface, could not be located.

- **H1.3** Time-lapse GPR measurements visualize subsurface flow dynamics and patterns and can replace hillslope trenches.'

  → Time-lapse GPR measurements lack the quantitative power and the direct link to structures (obtained by excavation) of trenches. Their spatial and temporal flexibility as well as their non-invasive character, however, are very advantageous for the investigation of highly dynamics and spatially distributed flow processes. Thus, the application of time-lapse GPR measurements in combination with soil moisture measurements are a powerful tool for the observation of hydrometric responses. Depending on the research question, the method can replace labor-intense trenches and even increase the observation density due to its spatial flexibility.

The titles of the two manuscripts will be adapted accordingly. They will be rephrased to emphasize the methodological aspects of the two papers, while keeping the focus on hillslope processes. The final versions of the titles are still subject to discussion. A possible suggestion for the title of this manuscript is:

FORM AND FUNCTION IN HILLSLOPE HYDROLOGY:
IN SITU CHARACTERIZATION OF SUBSURFACE FLOW BASED ON
RESPONSE OBSERVATIONS

We hope to have given a good overview over the anticipated revisions of the manuscript. While the elaboration above was meant to provide the 'big picture', our answers to the first reviewer's general and specific comments will illustrate how this concept will help to mitigate the reviewer's concerns. In the following, our replies are inserted into the original text by reviewer#1 and marked in blue.

**Answers to the general comments**

This paper presents very interesting data from a plot/hillslope scale sprinkling experiment combining TDR observations with time-lapse GPR to investigate soil moisture dynamics. Sadly, the promised (in the title) investigation of subsurface flow and the link to catchment scale is not convincingly presented, interpretations are often overdrawn and not supported by the results, and the paper currently reads more like a patchwork of data and ideas and not like a well aligned story. During the read, it felt as if the authors were not sure of what to present and what the key message of the work can/should be.

To set the frame for a more clear story line, we elaborated a common theme for the two companion papers: The theme focuses on the informative value of observations of 'form and function' for the hydrological investigation of hillslopes. While the first manuscript (MS1, at hand) uses process observations (i.e. response patterns and dynamics) to characterize the investigated hillslope, the second manuscript (MS2) starts with structural exploration of the subsurface only and complements these with the observed responses. This frame allows us to formulate clear and distinct hypotheses for both manuscripts and to align the story lines along the overarching and coherent theme without relying on the other manuscript too much.

For MS1 this means a thorough revision of the methodological discussion: The general aspects of subsurface process monitoring as presented in the current version will be drastically shortened. Instead, the discussion will focus on the value and shortcoming of response observations and the lack of structural information. Furthermore, the potential of time-lapse GPR measurements will be discussed in this context.

Finally, we will restructure the presentation and discussion of the hydrological response data in a way to to more clearly elaborate the logical link between observations and conclusions. Namely, the current separation in hillslope-scale and catchment-scale flow dynamics will be transferred into a separation of observations of the irrigation experiment and of the natural rainfall event. These changes will help to focus on the relevant aspects of the study and the novelty of the presented approach.

The data is visually very well presented, but the actual analysis of the experimental data remains poor, eventually supplying temporal evolution of soil moisture (although the method combination gives an interesting depth distribution to 4 m below surface) and flow velocities for the first respond[a)]. Moreover, these velocities are not really used[b)]. From a water balance perspective it is interesting, if the occurring amounts of PF can be, by any means, linked to the generated runoff volume in events[c)]. I also think that the inclusion of one event based hydrograph separation is a little bit thin[d)].

a) In this manuscript (MS1), we originally focused on the temporal evolution of the responses and deliberately minimized the discussion of spatial response patterns, as this was to be discussed in the companion paper. While we would like to keep the general separation of the two parts, the foci will be adapted to the new concept of 'form and function' of a hydrological system. This allows us to discuss certain aspects of the response patterns in more detail in MS1 already, without impairing the story line and arguments of MS2. We will for example provide a quantitative comparison between the responses to the natural rain event and the irrigation.

b) The response velocities were calculated to evaluate the observations of the response to the natural rainfall event: If the experimentally quantified response velocities are also occurring in flow paths, which were observed under natural conditions, these flow processes could be linked to the observed runoff response. In the revised manuscript, we will elaborate the line of argumentation and discuss its explanatory power as well as the calculated response velocities in more detail.

c) Unfortunately, the GPR signal can not be translated into quantitative soil moisture changes. The structural similarity attributes indicate areas of structural changes and can only be used to obtain spatial patterns and qualitative estimates of soil moisture dynamics. Although strong dissimilarity speaks for large changes in soil moisture, a quantitative relationship cannot be derived. To comply with the reviewers remark, we will improve the interpretation of the GPR data, e.g. by quantifying the spatial portions of the GPR transects activated during by the natural rainfall event and the irrigation experiment. We will also include isotope data sampled from piezometers at the irrigation site, which provide a basis for the discussion of flow proceses within the hillslope and the link between hillslope observations and the hydrograph.

d) The one hydrograph was shown to compare it with the subsurface response we could (partially) observe with our GPR measurements after the event and prior to the irrigation. As we do not have information on subsurface response for other events, we did not consider to include more hydrographs, and referred to former studies instead, reporting similarly fast runoff responses in the area. We will check if the inclusion of more hydrographs will improve the story line of the revised manuscript.

Stated by authors, the runoff ratio of 4% during the rapid peak (that consists of event water) cannot be explainable by direct rainfall on saturated areas. Nevertheless, the experimental observations and the fact that rapid flow downslope of the sprinkling site can be detected do not, in my opinion, convincingly explain the rapid stream response that is observed during rainfall in the study catchment. And this is why: I think that the method of the sprinkling experiment, with the use of a sprinkling intensity of 30mm/h, is simply not appropriate for investigating if preferential flow reaches the stream and generates the hydrograph under natural conditions.

I do understand that the purpose was to initiate the preferential flow with this rate, but I disagree with the authors that any explanation for the stream response can be done in this way. Such high intensities will of course lead lateral flow out of the sprinkling site as soon non-vertically oriented preferential flow paths are filled, or some low permeable lense or layer is reached. It would be interesting to report the natural rainfall intensities in the area and for the particular shown runoff event. As only a total of around 20 mm occurred. I am wondering if natural intensities would be indeed able to establish, supply, and connect a preferential flow system from the hillslope to the stream under unsaturated conditions on that time scale. This would be very important, since initiation of preferential flow strongly depends on intensities. I do not doubt that PF flow might be active at the hillslope and connecting to the stream in events with 30 mm/h rainfall intensity over hours, but when do such events occur? I think that there are no data and results provided in this manuscript that showed that preferential flow at natural rainfall intensities, amounts, and duration can explain the observed behaviour.

We agree with the reviewer, that the observations of the irrigation experiment alone are not sufficient to extrapolate our findings to natural rainfall events. The conclusion, that the experimentally initiated preferential flow processes are also relevant under natural conditions, was mainly based on the strong subsurface response after the natural rain event seen in the GPR data. The stronger subsurface response despite lower intensity is an indicator, that lateral preferential flow occurs in the shallow subsurface of the hillslope during natural events. To better convey this line of argumentation, we will present and discuss the observations from the natural event and the irrigation event separately. We will also include the intensities of natural rain events in the area and quantitatively compare the observed subsurface responses with regard to the areal share of activated flow paths. Finally, we will discuss the validity of these finding more carefully under a methodological aspect.

Furthermore, early work of McDonnell and various follow up studies showed that much of the water in preferential flow actually is not event water. This adds further doubt to the authors' interpretation, since the observed peaks in the stream in the area are dominated by event water.

We will reference and discuss the work of McDonnell et al. and others reporting on that topic in the revised manuscript. We will also add data on the isotopic composition of water seeping into the installed piezometers. In the current version of the two companion papers, the data were presented in the second part (MS2) to discuss the interaction between preferential flow paths and matrix. In the revised version, the data will be shifted to MS1 to support the discussion of the event water and pre-event water fractions. We will furthermore clarify, that we cannot be sure that the first peak is to be attributed solely to lateral preferential flow.

Leaving the interpretation of the catchment scale interpretation and the lateral preferential flow aside, the paper does not provide much novelty from a process perspective. The depth distribution data of soil moisture is interesting from methodological perspective, but work on this was published previously (see the Allroggen references). The discussion about the processes is not supported by the results of the work, and the methodological discussion is overly long, repetitive, lacks nearly any citations, and I was not struck by a major scientific message. That just leaves a case study of a sprinkling experiment, which is - in the current form - not sufficient to be published in HESS. That said, the dataset is very interesting and could supply interesting insights, but a clear idea of what should be presented, and what conclusions can be supported by the data is needed.

We understand, that our results and conclusions were not convincingly presented in the current version of the manuscript. The comments of the three reviewers, however, are helpful in pointing out shortcomings in the story line and we are confident to convey our key points and the novelty of our results more clearly in the revised manuscript.

As stated earlier, we developed a common theme linking the two companion papers. This theme or concept follows the idea, that a system is defined by its form (i.e. spatial structure) and function (i.e. processes and behavior). Based on this concept, the questions to pursue are: What information is needed or most suitable to understand the investigated system. And how much information on both, form and function, is necessary?

The irrigation experiment will be presented as an exemplary study on hillslope

hydrology. In this context, the power of time-lapse GPR measurements as a non-invasive, temporally and spatially flexible method for hydrological monitoring will be discussed.

Following, an in-depth analysis is needed that goes beyond the very nice visual presentation of soil moisture data and calculated flow velocities.

We had originally hoped to obtain soil moisture changes from GPR data. However, due to the highly heterogeneous radar reflections and the absence of defined horizons, only a qualitative interpretation was possible. Due to the open system and the unknown fluxes from the irrigation area towards the GPR-monitored downhill area, assumption about mass input or the ratio between diffusive and advective soil water flow remained highly ambiguous. To avoid over-interpretation, we chose to remain at the level of recorded arrival times of the signal, response patterns and dynamics.

We will elaborate the analysis of the GPR response patterns to quantitatively compare the natural event and the irrigation with regard to the spatial share of the activated hillslope cross-section. Furthermore, the dynamics of the TDR and GPR data will be analyzed more thoroughly.

**Answers to line-by-line comments**

- The syntax is often awkward (except in the introduction) and could highly benefit from a very careful revision (or editing by a native speaker)
  Will be done

- P1L10: 'hillslope-scale connectivity'. There is no evidence provided for connectivity. There is evidence that some water moves downward from the sprinkling site. This neither indicates connectivity in this case nor connectivity under natural conditions.
  Will be changed:
  - before: The experiment revealed a fast establishment of hillslope-scale connectivity despite unsaturated conditions...
  - now: The experiment revealed fast establishment of preferential flow paths at the hillslope-scale despite unsaturated conditions.

- P1L11-12: 'These processes' is somewhat vague.
  Will be changed to 'These preferential flow processes'

- P2L13: Change: 'is on' to 'was on'
  Will be done

- P2L26: Change: 'At minimal...' to: 'GPR provides...'
  Will be changed:
  - before: At minimal invasive cost it provides information on subsurface structures in electrically resistive soils.
  - now: GPR provides information on subsurface structures in electrically resistive soils at minimal invasive cost.

- P3L7ff. I am not sure, if introducing multi-modal transit time distributions here and in the following parts is beneficial for the manuscript. If the authors introduce it and eventually mention it at several points, they should start calculating them for the mentioned events, rather than doing

a simple two component end member mixing (although this seems not to be possible with the isotope sampling). References are needed after 'flow process'.

In the revised manuscript, we will avoid the term 'multi-modal transit time distributions' to keep the focus on hillslope processes.

- P3L16ff. References for the claims are needed. Why vague?
  This refers to the fact that many studies do not specify subsurface flow processes in more detail, while the term itself subsumes a wide variety of different processes. The sentence was meant to introduce the following paragraph, where we state that the specific characteristics of subsurface flow processes are a crucial component in the hydrological behavior of a landscape. A less vague formulation will be used.

- P3L16ff. The research need that is outlined here, is not convincingly derived and link to the majority of the introduction. The introduction exclusively deals with preferential flow processes, before some double-peaked hydrograph behaviour is mentioned. After that, the research needs is only related to these hydrographs. As mentioned earlier, I miss the well thought out story line. I also missed references to important connectivity work, since the connectivity (hillslope-stream) part seemed to be an important part of the paper.
  The introduction will be revised and focused on investigation of subsurface flow in the context of the concept of 'form and function' applied to hydrological systems. These two aspects will be merged to elaborate the research question of how the observation of responses can be used to characterize a hillslope with regard to its form and function. The use of concepts like connectivity, double-peak hydrographs and multi-modal transit time distributions will be carefully checked and adapted to the revised story line.

- P6L10: What is the uncertainty that is associated with this very simple hydrograph separation, especially when accounting for the high variability of stable isotopes in rainfall?
  Uncertainties will be provided and discussed

- P7L3: Please provide the relative location of the irrigation site, when hillfoot is 0 and hilltop is 1. What is the length of the hillslope?
  Will be provided

- P7L3: Language and syntax are improvable.
  We will improve the language of the manuscript and consult a native speaker for final editing.

- P7L10: What is the return period of the sprinkling event (intensity, amount)?
  Will be provided

- P7L11: Here it is stated that it is not intended to mimic natural conditions, but then the authors used the results to infer flow processes under natural conditions. This does not fit together.
  The conclusion, that similar processes also occur under natural conditions

is based on the observation of similar GPR reflection patterns and dynamics in response to the natural storm event. In the revised manuscript, we will elaborate this line of thought and clarify the causal connection between observations and conclusions. This will be done by stating clear hypotheses in the beginning. We will furthermore discuss the natural and irrigation responses separately, compare the results, and put the observations in relation to each other.

- P7L17ff: Information about size and height above ground of the samplers are needed. What were the intensities of the natural rainfall events? This could be a good comparison to the experiment.
  Will be provided

- P9L26-28: 'The...' Better delete? Not needed. Another suggesting: better not start sentences with 'Figure 2 shows/presents/' or 'This/that is presented in Figure 2'; rather report the finding or what the reader show see and then reference the figure.
  We are not sure what this comment is referring to, as there is no reference to a figure in the annotated paragraph. We will, however, take this comment as a general recommendation to be applied to the whole manuscript.

- P10L1: 'Similar'. So what is the difference to the cited work?
  The same procedure was used and the text will accordingly be changed to 'as presented by...'.

- P10L14: 'suggested' Is certain or uncertain?
  The GPR data certainly show changes in GPR reflection patterns, which is interpreted as changes in water content. Soil moisture is not the only possible factor causing differences in GPR reflection patterns, but under the given circumstances/time scale the only reasonable one. The term will be rephrased for clarification.

- P10L14: I am also wondering where the separation between mobile and stagnant water was made?
  In contrast to the TDR data, GPR data does not provide absolute soil moisture values, and all interpretations are based on differences between different measurements. As the last GPR measurement was chosen as reference, water which was still there during the last measurement will not be visible in the structural similarity patterns. We therefore can only observe changes in soil moisture, and infer information on advectively and non-uniformly flowing mobile water only. The sentence will be rephrased for clarification.

- P10L31: Why was the 2% value chosen?
  The 2% threshold was chosen based on the variability within presumably constant soil moisture measurements. This will be clarified in the revised manuscript.

- P10L4ff: 'were calculated for the entire depth': Do you mean, that if you give the value for 10 cm it is the velocity for 0-10 cm, if you report the velocity for 20 cm depth, it is the velocity from 0-20 cm?
  That is right. This will be clarified in the revised manuscript.

- P10L9: 'main stem'. Delete?
  Will be changed.

- P12L13ff: How are these hydrographs separated? What method was chosen? This needs to be mentioned in the methods. What are the uncertainties associated with the isotope hydrograph separation?
  The required method descriptions and uncertainties will be provided.

- P12L13ff: Why was only one event chosen for analysis?
  As mentioned earlier, we chose to show only one hydrograph, because the emphasis of this part of the discussion was on the timing between rain event and the observed subsurface GPR reflection patterns (prior to the irrigation experiment). We chose to show hydrographs of several subcatchments to prove that the behavior is not specific to the investigated Holtz river, and referenced former studies to show that the observed response was not a unique event. In the revised manuscript, we will emphasize the results and discussion of the subsurface responses rather than the hydrograph. However, we will also check if the presentation of more hydrographs benefits the story line.

- P13: Strange font size/type of '20 mm' in the figure captions of fig.3.
  Will be changed.

- P13L2: 0.5 L. This was over the whole experiment? So more or less unimportant in the water balance.
  This is what we conclude later on in the manuscript. We also consider this an important finding, as surface runoff is often mentioned as a possible reason for an immediate runoff response or the event water signal in the river.

- P13L2 'relatively': Please provide a objective description, rather than using subjective terms like small, relative, best with numbers. Do that throughout the manuscript.
  Thank you for pointing this out. We will comply with this request.

- P13L8: Can these 20% be the uncertainty of the water balance? Or what is the actual uncertainty and how does that influence the results and interpretation? I am not sure what 'average storage deficit of 20%' means.
  The average storage deficit refers to difference between the mean storage increase in the four core area TDR profiles and the water input, which will be clarified. There is an uncertainty in this average water balance, which is mainly due to the low spatial coverage of four TDR profiles at the core area. It is likely that four profiles are not sufficient to fully represent the water balance of the entire irrigation site, which is why we chose to show single profiles instead of average values. We will mention and discuss this aspect in the revised manuscript.
  The water balance uncertainty of single profiles, however, is smaller, yet not representative for the entire core area. Here, the change between mass balance overshoot in the beginning and deficit later on is an indicator for a high portion of mobile water in the soil. This differentiation and the respective uncertainties will be mentioned in the revised manuscript.

- P14: Fig4. Uncertainty needed.
  Will be provided.

- P15L4-7: Move to methods.
  Will be changed.

- P15L13: 'generally low dynamics'. Please change.
  Will be changed to 'The temporally and spatially low dynamics'

- P15L31: 'all fast dynamics'
  Will be changed to 'no changes in GPR signal could be observed anymore'

- P15L32-33: Is there a possibility to differentiate if it is in the matrix or left the monitored area? How about working more with the water balance?
  There are several reasons why we can not calculate a water balance from GPR data. The most important one is that we can only observe structural differences between measurements. We can not tell how much water is left in the mapped transects at the end of the monitoring period, nor can we quantify the differences in terms of water content.
  In the revised manuscript, we will include the calculation of storage changes for the TDR profiles at the downhill monitoring area, to provide an estimation on how much water remains in the matrix. A water balance can not be calculated here either, though. Reasons are the unknown water input, i.e. lateral flow from the irrigation area, and the sparse spatial coverage vs. high heterogeneity of the monitored soil volume.

- P17L2: This is more discussion of processes then results. How was this inferred? It seems like a statement, but not supported by data. There is no evidence for this.
  The paragraph will be moved to the Discussion and the line of thought will be elaborated more clearly. Nevertheless we consider it as one of the results of the GPR measurements, that the natural rain event as well as the irrigation experiment caused soil moisture changes in the entire monitored soil profile down to a depth of 4.2 m. These patterns were shown to be patchy and do not indicate a water table.

- P17L4: Again, there is no proof for this. It is just speculative. How deep below the soil surface is the bedrock?
  This statement is based on the comparison of the GPR reflection patterns caused by the natural rain event and the artificial irrigation. It will be moved to the Discussion section and discussed more carefully.
  We do not have reliable information on the depth of the bedrock interface and it is presumably characterized by a gradual transition between the deposit layer and solid bedrock. At the investigated hillslope, drilling was usually inhibited by high stone content at a depth of 1.7 to 2m. This information will be added to the methods section.

- P18: Can velocity distributions be estimated? They might be more interesting/informative than just the first response time. One could see how the majority of the water moves, as the response time of the preferential flow should be somehow consistent with the time of the hydrograph response, especially if the preferential flow should explain the delivery of

event water to the stream.

Unfortunately, soil moisture changes on their own are not sufficient to calculate or estimate velocity distributions. To do so, tracer applications and the recording of breakthrough curves are required. We agree, that the estimation of velocity distributions would be very helpful in the investigation of the role of preferential flow at the catchment-scale. However, response velocities also bear some very important information, especially in such high spatial coverage as provided by the GPR measurements.

- P22L6: This is commonly observed. Soil moisture variability increases with decreasing moisture contents
  The statement was meant to explain the observation, rather than being a finding of the experiment. An adequate citation will be added.

- P22L12: 'high portion of mobile water'. Please quantify. This is important.
  Will be provided in the Results section.

- P22L13: 'Velocities over...' But this only relates to the fastest component. The water itself should travel with a velocity distribution, which was not estimated. I would argue that the median or average speed to the 'mobile' water (the matrix water is mobile, too) is less than 10-3 m/s. Give the numbers for MaiMai.
  We agree that the median or average *real* velocity of the water surely is much lower than the given maximum *response* velocity of $2.2 \cdot 10^{-3} \, ms^{-1}$. The response velocities shown in Figure 8 range between $10^{-4} \, ms^{-1}$ and $10^{-3} \, ms^{-1}$ (based on GPR and TDR measurements). Any water arriving after the first response cannot be used to quantify volumes or velocities, because (a) we can not reliably quantify the newly arriving water based on the soil moisture state of the monitored soil volume and (b) we do not know the 'start time' of the water, which might be somewhere between irrigation start and measuring time.
  The intention here was to investigate the potential of preferential flow for the immediate runoff response. We use the maximum response velocities to show that the response velocities occurring in preferential flow paths are sufficient to route water from the hillslopes to the river within the short time between rain event and discharge response. The fast decrease in soil moisture after irrigation stop shows that a large portion of the water (approx. 50% at the core area, see Figure 4) is highly mobile and leaves the monitored depth at velocities that exceed the potential matrix flow velocities.

  This will be clarified in the revised manuscript and the line of argumentation will be elaborated.

  Numbers for MaiMai will be provided.

- P22L16ff. This is true, but it still does not show water transport of volumes to the stream that might explain the hydrograph response. Is there impermeable/less permeable material below the field site? How deep is the bedrock? Was an attempt made to observe the subsurface stormflow there?

We cannot exactly locate the depth of the bedrock interface, but maximum possible drilling is usually at depths between 1.5m to 2m at the investigated hillslope. The application of GPR measurements was clearly motivated by our will to identify preferential flow layers if they existed. This will be discussed in the revised manuscript.

- P22L16ff. In addition: The activation of the preferential flow is highly dependent on the rainfall intensities. They might never be active under natural conditions when rainfall intensities are an order of magnitude lower (over event scale).
  Based on the strong signal observed in the GPR measurements between natural rain event and irrigation, we conclude that there was an even stronger response of preferential flow caused by the natural event despite lower intensity and shorter duration. This finding was thus not solely based on the experimental observations. The line of argumentation was obviously not clear enough and will be elaborated in the revised manuscript. This will be done by a separate discussion of the experimental results and the natural response (including the GPR responses) and the quantitative comparison of the two different responses.

- P22L24-25: Connectivity at hillslope scale was not shown in this experiment. The data simply showed that some preferential flow out of the sprinkling site occurred.
  We agree that the experimental results alone do not prove hillslope-scale connectivity. As mentioned above, this statement is based on a synthesis of experimental results and the observed response triggered by the natural rain event. Unfortunately, we can not translate the GPR signal into quantitative soil moisture changes and provide a mass balance of the monitored transects. But the strength of the natural signal in relation to the experimentally initiated response, clearly indicates a higher amount of water despite lower local input. We will qualitatively compare the observations from natural rainfall event and irrigation and also discuss the effect of additional vertical input during the natural event in contrast to the exclusively lateral input during irrigation. We will also check whether the use of the term connectivity is appropriate in this context.

- P22L30: Every hydrograph is likely a result of multi-modal distribution of travel times...
  The catchment-scale response, as well as double-peak hydrographs and transit time distributions in particular, will be discussed more carefully in the revised manuscript. The entire section will therefore be restructured.

- P23L20: How were the riparian zones and wetland patches delineated? What is the uncertainty? Is there a contraction/extension during events that could change this?
  The wetland patches are very small and restricted to the source area of the river. A larger patch which is not in direct proximity of the river was mapped in the field once. The rest was estimated based on field observations, by applying a buffer of 2 m to the river sections associated with wetland patches. This is just an estimate, and we can not exclude contraction/extension of this area. However, consider its impact minor

with regard to the overall areal share. This will be clarified in the revised manuscript.

- P23L22ff. Again, there is no evidence shown in this work that there was any connectivity between hillslope and stream. This is even more true when natural conditions are considered.
  As mentioned above, we will elaborate the line of argumentation regarding the observed hillslope processes more clearly in the revised manuscript (see replies to comments on P22L16ff and P22L24-25). We will also focus more on hillslope processes rather than catchment-scale interpretations. By doing so, we will stay within the actual scope of our dataset and avoid over-interpretation.

- P24L26ff. This methodological discussion is overly long, repetitive, and lacking references to work of others.
  The methodological discussion will be significantly shortened and streamlined to emphasize the novelty of the method rather than discussing general aspects of soil moisture measurements and subsurface exploration.

- P24L27-P25L3: Here I disagree. As said, the experiment was not designed to mimic natural conditions. This is completely fine, since it seems to me that the original idea was to employ the GPR system. Nevertheless, the setup does not allow inferring natural flow processes that explain hydrograph response.
  The experiment was designed to visualize potential preferential flow paths. The natural rain event, however, allowed us to put our observations in the context of naturally occurring responses. The analysis of similarities and differences between the response patterns and dynamics allows for conclusions on naturally occurring processes.

  We agree that the current presentation is misleading and needs clarification. We will revise the discussion of the general experimental setup. Furthermore, we will shift the focus of the entire manuscript towards an evaluation of the methods and the approach.

- P25L10: Explain: 'sufficiently steady state'
  The phrase refers to the preceding paragraph, stating that natural systems barely reach steady state conditions. For the investigation of preferential flow processes, however, the absence of advection can be assumed as steady state, even though comparably slow diffusive movement is still going on. In other words: To identify the dynamics of advective flow processes initiated by a temporally limited input, the amount of water moving slowly through the system does not matter. While the first one causes fast changes in soil moisture, the latter is visible as a trend and can easily be ignored.

  The paragraph will be rephrased and the term avoided.

- P25L24: Sat. conditions are most likely occurring at the bedrock interface. Technically the periglacial layers are not 'soils' anymore. Try to have a look at the work of Uhlenbrook et al. on periglacial slopes.
  We agree that the sentence 'Saturated conditions are very improbable in such soils.' needs to be rephrased or even deleted as indeed saturation is possible above a confining layer. However, the GPR data did not show

such a saturated area within the top 4 m, while on the other hand the rock became too hard to further penetrate with our power drill at a depth of max. 2 m. We therefore struggle to identify both the form/structure and the function/response of such an impermeable layer in the investigated hillslope.
We will also be more careful with the use of the word 'soil'.

- P27L31ff. This seems to be unnecessary - no important point was made in this section.
The methodological discussion will be shortened and the section removed.

- P29L4: The resulting connectivity was not shown and remains highly speculative.

  P29L5ff: This is speculative. Furthermore, the link to transit time distributions is somewhat strange.

  P29L20: Again, I think this was not shown here
  Summarizing the three comments above, we are very sorry that some key points of the manuscript were not well conveyed in its current version. As mentioned earlier, the manuscript will be revised and restructured to better elaborate these points. Furthermore, we will check the conclusiveness of our findings and conclusions and formulate them more carefully.

---

## Author Comment (AC3)

**Response to comments by reviewer#3**

We highly appreciate the thorough review and constructive comments of the anonymous reviewer. We are happy to implement and discuss the cogent remarks on data analysis and presentation of our findings and developed a concept to tackle the basic criticism regarding the story line and the conclusiveness of our results. This concept is based on the comments by all three reviewers and provides a common theme as well as overarching yet standalone story lines for the two companion papers. We will shortly present the main idea of the new concept, the revised hypotheses as well as the restructured story line of the first manuscript (MS1). As these improvements are relevant for our response to the comments by all reviewers, this part is included in all replies. Afterwards, we will address the general and specific comments made by reviewer#3, with our replies inserted in the original review.

**Conceptual framework**

To better elaborate the methodological aspects of the study and to provide a common theme for the two companion papers, we want to employ the concept of form vs. function. The original term 'form follows function' was first established in architecture and soon was adopted by biologists. It refers to the idea, that form and functionality are closely correlated, influence each other and co-evolve. We suggest to transfer the same idea to hydrological systems. This allows us to separate and analyze their two main characteristics: Their form, which is equivalent to the spatial structure and static properties, and their function, equivalent to internal responses and hydrological behavior. While this approach itself is not particularly new to hydrological field research, we want to employ this concept to explicitly pursue the question of what information is most advantageous to understand a hydrological system.

Accordingly, we developed different categories to organize and describe the data presented in the two manuscripts: Structural data summarizes all sorts of data which focus on direct exploration of form, e.g. soil cores. Response dynamics, on the other hand, are observations of function. They represent processes deprived of their spatial context and include soil moisture dynamics and discharge responses. In between these two categories are flow-relevant structures and response patterns, which may contain information on both, form and function of the system.

In the presented study, we apply this concept to subsurface flow within a hillslope. The first part of the study (MS1) methodologically focuses on function: We observed response patterns and dynamics from a natural rainfall event and during an irrigation experiment. The results are used to infer hydrological processes and the spatial organization of the monitored system. Based on these findings, the informative power and conclusiveness of the data will be discussed.

The second manuscript (MS2) focuses on form and starts off with a thorough structural exploration of the subsurface. It then proceeds towards observations of flow-relevant structures and response patterns and analyzes the information gain along this path.

**Hypotheses and story line**

The hypotheses of both manuscripts can be aligned according to the form/function framework and will clearly be stated at in the beginning. The hypotheses of MS1 will focus on the potential of response observations for hillslope hydrological field research and the application of time-lapse GPR measurements in this context:

- **H1.1** Response observations (discharge, TDR & GPR data) are sufficient to characterize subsurface flow ywithin the hillslope.

- **H1.2** Response patterns can be used to deduce flow-relevant structures in the subsurface.

- **H1.3** Time-lapse GPR measurements visualize subsurface flow dynamics and patterns and can replace hillslope trenches.

The story line will be streamlined and arranged along these hypotheses. This will help to make the manuscript easier to follow and to better elaborate the important and novel key points of our study. In the following, new or restructured sections are marked in brown.

**1 Introduction**

- Concept of form and function: form and function are the main defining features of a system. Applied to catchments, the concept describes any kind of spatial structure (from topography to macropores) as form, and the sum of all processes defining the hydrological behavior of the catchment as function. Both strongly influence and determine each other.

- Example subsurface flow at the hillslope: structures and heterogeneity control flow patterns and velocities and thus the occurrence of preferential flow. Which one is better suited to characterize a hillslope, form or function?

- Focus on function: What does it need to describe subsurface flow and preferential flow at the hillslope? How to observe subsurface flow processes? What do response patterns and dynamics tell us about form and function of a hillslope?

- Methodological challenge of preferential flow: former approaches at different scales: plot, hillslope- and catchment-scale.

- Hypotheses as stated above

**2 Methods**

- Study site description

- Hydrological response monitoring: Hydrograph and surface water isotopes

- Hillslope-scale irrigation experiment

  - Setup
  - Process monitoring
  - Piezometer isotope sampling

- Data analysis

  - TDR data analysis
  - GPR data analysis
  - Comparison natural event vs. irrigation: Distinguish the signals and calculate areal share of activated cross section
  - Response velocity calculation

**3 Results**

- Response to the natural rainfall

  - Hydrograph and surface water isotopes
  - Subsurface response patterns (green GPR reflection patterns)

- Irrigation experiment

  - Core area water balance
  - Soil moisture dynamics
  - 2D time-lapse GPR
  - Soil and piezometer isotopes
  - Combination of TDR and GPR
  - Response velocities

- Comparison of natural event and irrigation: Areal share and signal strength of GPR measurements before and after irrigation

**4 Discussion**

- Process interpretation

  - Interpretation of artificially induced response observations during and after the irrigation
  - Interpretation of the natural response observations after the rainfall event prior to the irrigation experiment
  - Identification of (flow-relevant) structures from response patterns

- Methodological discussion

**5 Conclusions**

- **H1.1** 'Response observations (discharge, TDR & GPR data) are sufficient to characterize subsurface flow within the hillslope.'

  → Processes can be identified and characterized without any concrete information about spatial structures. However, observations are limited in spatial (and temporal) resolution and interpretations beyond observation scale remain speculative.

- **H1.2** 'Response patterns can be used to deduce flow-relevant structures in the subsurface.'

  → Response patterns allowed to develop a conceptual description of the flow paths network, which is linked to subsurface structures. However, actual structural features, such as the deposit layer or the bedrock interface, could not be located.

- **H1.3** Time-lapse GPR measurements visualize subsurface flow dynamics and patterns and can replace hillslope trenches.'

  → Time-lapse GPR measurements lack the quantitative power and the direct link to structures (obtained by excavation) of trenches. Their spatial and temporal flexibility as well as their non-invasive character, however, are very advantageous for the investigation of highly dynamics and spatially distributed flow processes. Thus, the application of time-lapse GPR measurements in combination with soil moisture measurements are a powerful tool for the observation of hydrometric responses. Depending on the research question, the method can replace labor-intense trenches and even increase the observation density due to its spatial flexibility.

The titles of the two manuscripts will be adapted accordingly. They will be rephrased to emphasize the methodological aspects of the two papers, while keeping the focus on hillslope processes. The final versions of the titles are still subject to discussion. A possible suggestion for the title of this manuscript is:

FORM AND FUNCTION IN HILLSLOPE HYDROLOGY:
IN SITU CHARACTERIZATION OF SUBSURFACE FLOW BASED ON
RESPONSE OBSERVATIONS

We hope to have given a good overview over the anticipated revisions of the manuscript. While the elaboration above was meant to provide the 'big picture', our answers to the first reviewer's general and specific comments will illustrate how this concept will help to mitigate the reviewer's concerns. In the following, our replies are inserted into the original text by reviewer#3 and marked in orange.

**Answers to the general comments**

This study is designed to investigate subsurface flow processes, and in specific preferential flow, at two different scales. The authors chose a multi-method approach including quantitative soil moisture measurements using TDR (time domain reflectometry) and qualitative soil moisture measurements using GPR (ground penetrating radar) during an irrigation experiment at the hillslope scale. The experiment was complemented with common hydrometric and tracers techniques (interpretation of the rainfall runoff graph and stable water isotopes of rainfall and runoff) at different catchment scales. The study addresses an important topic in the hydrological sciences. The manuscript is clearly structured following common scientific standards. The data are presented in (mostly) clearly arranged, high quality figures. The combination of TDR and GPR measurements provide a very promising approach to monitor subsurface processes. While this approach is relatively novel, it bears a number of uncertainties and disadvantages, which are discussed at length (although the methodological discussion lacks a bit in references).

However, I do have some concerns. First of all, clear objectives were not formulated in the beginning of the manuscript and it follows that the rest of the story misses a central theme.

We will formulate clear hypotheses (see above) and state them at the beginning of the revised manuscript. The story line will be revised and adapted accordingly.

I do understand though, that the catchment scale analyses revealed that the system responded with a double-peak hydrograph to water input with the first, steeply and quickly responding hydrograph consisting of a mixture of new and old water and the second, dampened hydrograph consisting dominantly of old water. The results from the irrigation experiment were then used to explain the hillslope processes that may potentially lead to this catchment scale response. This leads to my second and third concerns: (2) I feel that only one hydrograph separation to calculate the event and pre-event water fractions of a catchment is not representative. Many studies have shown in the past that the event water fraction may vary largely in one catchment depending on various factors, e.g. water input and/or antecedent moisture conditions [e.g. Munoz-Villers & McDonnell, 2012]. Hence, I suggest including more data and/or an uncertainty analysis to add representability to the HS.

The one hydrograph was shown to temporally relate the observed subsurface response after the natural rain event (the green portion of the GPR data shown in Fig. 8) to the discharge behavior. The timing of the two observations, the subsurface response on the one hand and the discharge response on the other, allows us to draw conclusions on the processes within the hillslope. This line of thought was not properly conveyed. We will improve the story line by presenting and discussing the responses to the natural rain event and the irrigation separately. A quantitative comparison of the GPR response patterns of both events (the natural and experimental) will emphasize the focus on subsurface flow processes rather than catchment-scale responses. We agree that the fraction of event water in the first peak is likely to vary with rainfall characteristics and antecedent conditions and will avoid general statements based on this single event. We will furthermore check, whether the inclusion of more events will benefit the revised story line.

And (3) the authors clearly point out that they did not intend to mimic natural conditions with the irrigation experiment, however, they use the results to explain the natural response at the catchment scale. While this shortcoming is mentioned in the methodological discussion, I don't feel that this justification is sufficient to link the observed processes in the irrigation experiment to the natural conditions at the catchment scale.

The conclusions about relevant processes under natural conditions was backed by the observations of the subsurface response to the rain event prior to the experiment. This was not clearly elaborated and will be improved as described above. We will also shift the focus of the manuscript towards the methodological aspects of GPR measurements for hydrological monitoring and the informative power of response observations for the characterization of hillslopes.

At last, I feel that the novelty of this study is not clearly conveyed to the reader. I think it is the combined use of TDR and GPR measurements which bear a large potential to move forward in subsurface flow process understanding. However, this message needs to be presented more clearly.

With a stronger focus on the methodological aspects of the experiment, we are confident to improve the presentation and convey the novelty of our findings.

**Answers to the detailed comments**

**Objectives and experimental approach**

- I think that this subsection needs a bit more structure and/or more precise phrasing. Maybe name all objectives at the beginning of the paragraph and then list the approaches that you chose to address them in addition to the explanations why you chose said approaches (advantages vs disadvantages of others).

  This will be done. Clear hypotheses and research questions will be stated at the end of the introduction of the revised manuscript. We furthermore developed a concept to organize and subsume the approaches and methods presented in the two companion papers. This will strongly improve the structure of the introduction as well as the story line of the manuscript.

- Page 3, line 24: the role of what? Please add information because otherwise this sentence and subsequent paragraph come a bit out of the blue, i.e. raise further questions, e.g. why is it necessary to use a multi-scale approach?

  We will fully revise this subsection and clarify that sentence.

- Page 3, line 25: What are the conventional hydrological methods that you mention? I suppose the TDR measurements that you chose as approach in your experiment? Please add this information to the text or rephrase.

  The term 'conventional hydrological methods' refers to TDR measurements and piezometers. We will clarify this in the revised manuscript and fully revise this subsection.

- Page 3, line 27: you only propose GPR, why do you not mention the other techniques? (Maybe use this 3 comments to elaborate your objectives.)

  We agree and will mention other techniques in the revised manuscript. We will also fully revise this subsection.

**Methods**

**Hydrological response monitoring**

- Page 6, line 22: How exactly was rainfall water sampled for isotope analysis? Was a sequential sampler used or is it a bulk sample? Please add information.
  This information will be provided.

- Line 24 (and everywhere else in the ms): I suggest to use 'hydrograph separation' instead of 'mixing model', the latter implies that an EMMA was performed which is not the case.
  We will make this correction, thank you for pointing this out.

**Process monitoring**

- Page 8, line 3: Please add information why different TDR sensors were used.
  We wanted to increase the measurement frequency by measuring with three sensors in parallel. As only these three sensors were available to us, we decided that the benefit of higher frequency measurements out-weighed the disadvantages caused by the different sampling volume of one of the sensors.

- Page 8, line 6: Which 3 TDR tubes were used? Sounds like they were mentioned before which they aren't. Please rephrase or add information for better understanding.
  The sentence refers to the three different sensors, which were mentioned above. As this was obviously not phrased clearly enough, we will rephrase the paragraph: 'Soil moisture was measured manually. To enable parallel measurements and increase temporal resolution of the measurements, three probes were used in parallel. These probes differed slightly in their sensor design: Two TDR probes had an integration depth (i.e. sensor length) of 0.12 m and one probe had an integration depth of 0.18 m. These sensors were successively lowered to different depths into the 16 access tubes, where they measured the dielectric permittivity of the surrounding soil in the time domain through the PVC of the access tubes. Given a mean penetration depth of 5.5 cm and a tube diameter of 4.2 cm this yields an integration volume of 0.72 L and 1.05 L, respectively. The manual measurements were conducted in 0.1 m depth increments and followed a flexible measuring routine to cover active soil profiles with higher frequency.'

**Results**

- Figure 3:

  there is too much information in this Figure. I suggest moving some of the information to e.g. a separate table. For example, the information of the catchment sizes and the % of the precipitation amount should be placed

somewhere else, maybe together in one table. You may mark the 7th and 72th hr in the RR-Graph and refer the reader to the table. It also remains unclear to me why the information about the irrigation experiment (i.e. the mean structural similarity attribute) are placed in the rainfall-runoff and chemographs of the headwater catchments. Which y-axis does the mean structural attribute refer to? And what do the vertical bars mean? Please provide this information in the legend. I suggest considering making two different graphs, for example below each other in the manner of Figure a) catchment scale and b) hillslope scale. This should help to understand this figure faster and identify/extract important information more easily.

We agree that this figure is very busy and loaded with information. To make it easier to grasp this information, we will divide the figure in two graphs, add another y-axis for the mean structural similarity attribute and more clearly explain the shown GPR dynamics. However, as we consider the timing of (the green portion of) the GPR signal relative to the hydrograph to be important for the overall interpretation, we would prefer to keep this information together in one figure. We will furthermore elaborate the importance of the timing in more detail in the revised manuscript.

Information from the legend will be provided in a separate table.

Please also provide information about where groundwater was sampled for d18O, e.g. in section 2.2.
We will provide this information.

- Page 12, lines 7: Since there are several peaks in Figure 3, it would make it easier for the reader if you added the date of the described rain/hydrograph event.
  The events will be labeled for clarification.

- Page 13, line 5: What is an 'overshoot' in mass recovery? Please add a bit more information to explain this.
  We will rephrase the sentence and explain the observation: 'All profiles showed a mass recovery of more than 100 % (i.e. higher storage increase than water input at measuring time, see Fig. 4) in the first 60 min of the irrigation period.'

- Page 14, lines 19-20 and Figure 5: The weak signal of soil moisture dynamics in TDR 13, 6 and 14 is apparent in Figure 5. While an explanation is provided for the signal in TDR 6, I am missing explanation for TDR 13 and 14. How do you explain this? I suggest including this in the discussion section 4.1.
  The behavior of TDR14 and TDR13 will be mentioned here ('TDR13 and TDR14 show a slight decrease in soil moisture.') and discussed in Section 4.

- Figure 5: Soil moisture change is interpolated over time for visual reasons (according to figure caption), I think it would be helpful to add this information somewhere in the methods section (e.g. 2.4.1. TDR data analysis).
  We agree and will add this information Section 2.4.1. 'While the data

was interpolated in time for better visualization, all data analyses were performed with the uninterpolated data.'

**Discussion**

- Process interpretation: What is the novelty about this study? I suggest elaborating this e.g. add a paragraph to emphasize this a bit more clearly. The strength of our two-part study is the discussion of various data sources and their contribution to process understanding in the context of the interplay of form and function (a biological concept here transferred to hillslope hydrology). In the first part (MS1), the focus is on the observation of response patterns with GPR and TDR measurements. The combined use of these methods and their evaluation in the context of form and function of a hillslope as well as an alternative to the classical hillslope trench is the novelty of this study. To emphasize this aspect more clearly, we developed the above-mentioned concept for the two companion papers, highlighting two different methodological approaches. Both approaches include GPR measurements, which allows discussing the method under different aspects. According to this new concept, the story line of the manuscript will be revised and better structured along our research objective and hypotheses.

- Page 22, lines 22-25: I do not understand the link between delayed signal in the intermediate depth and the network of preferential flow paths. Please add a bit more text.
  The term 'delayed signal in the intermediate depths' refers to the fact, that the soil moisture signal arrived in greater depth (i.e. below approximately 1 m) before it appeared in the intermediate depth (approximately 0.5 to 1 m). Thus, the water bypassed this intermediate layer without being detected. Two scenarios can explain this observation: a) The water passes through the intermediate depth outside of the monitored soil volume. b) The water passes through the intermediate depth through small preferential flow paths. If the volume of these flow path is comparably small in comparison with the monitored soil volume, they will only become visible (by means of soil moisture changes) if the water leaks into the surrounding matrix. While we can not distinguish between these processes by means of the data presented in MS1, both are preferential flow processes acting at different scales. This will be clarified in the revised manuscript.

- Page 23, lines 22-29: So, does that mean that the first peak is generated by overland flow (as observed by Wrede et al.) AND shallow subsurface flow? Or is all of the event water conveyed via subsurface flow? Please express your interpretation a bit more clearly.
  We can not make any statement about the importance of overland flow for runoff generation beyond our observation, that surface runoff is of minor importance at the hillslopes, and that the areal share of saturated areas in the catchments is small. We therefore assume that the fraction of surface runoff is small. On the other hand, our results from the hillslope experiment show that the slope response is indeed fast and a potential source for mobile water. We thus conclude that preferential flow in the shallow

subsurface to some extent contributes to the first peak. To address the question about the fraction of event and pre-event water from the hillsopes in more detail, we will also include the soil water isotope data in MS1 (currently presented in MS2).

We will clarify these interpretations and include the above-mentioned argumentation. We will also revise the story line in a manner to stronger focus it on the hillslope response and its investigation instead of catchment-scale behavior.

**Technical corrections**

- Figure 4: Can you move the y-axis name and unit away from the numbers, e.g. to the very top or remove 140 and insert 120 instead?
  Will be done.

**Typing errors**

- Throughout the whole manuscript use either one of Fig. X, Fig X or Figure X.
  Will be checked and adapted to the journal's specifications.

- Page 3, line 8: empirically
  Will be done.

- Page 9, line 23: Add 'of' in 'In case of the vertical profiles...'
  Will be done.

- Page 12, line 19: use 'to' instead of 'of' in '...up to 67.6 % to the event runoff...'
  Will be done.

- Page 13, Figure 3, legend: 'attribute' instead of 'atribute'
  Will be done.

- Page 14, line 3: delete 'to' in '...returned to similar to initial conditions...'
  Will be done.

- Page 14, line 15: insert '(' in 'Fig.5)'
  Will be done.

- Page 15, line 2: use plural instead of singular: 'flow paths'
  Will be done.

- Page 19, Figure 6 caption, line 1: time-lapse GPR
  Will be done.

- Page 22, line 13: I suggest to use '...match...' instead of '...fitted with...'
  Will be changed to 'agreed'.

- Page 24, line 22: delete redundant 'the'
  Will be done.

- Page 27, line 17: substitute 'is' with 'are' in 'The dynamics of preferential flow are often characterized...'
  Will be done.

- Page 29, line 14: stable isotopes instead of staple isotopes
  Will be done.

---

## Author Response (AR1)

**General comments on the revised manuscript**

We highly appreciate the thorough review and constructive comments of the anonymous reviewers. We implemented the cogent remarks on data analysis and presentation of our findings and developed a concept to tackle the basic criticism regarding the story line and the conclusiveness of our results. This concept is based on the comments by all three reviewers and provides a common theme as well as overarching yet standalone story lines for the two companion papers. We will shortly present the main idea of the new concept, the revised hypotheses as well as the restructured story line of the manuscript at hand. Afterwards, we will address the general and specific comments made by the three reviewers, with our replies inserted in the original review. Please note that this document is closely related to our first response to the reviewers' comments. We followed the anticipated measures to a large extend and consider this document an updated version of the previous document.

**Conceptual framework**

To better elaborate the methodological aspects of the study and to provide a common theme for the two companion papers, we want to employ the concept of form vs. function. The original term 'form follows function' was first established in architecture and soon was adopted by biologists. It refers to the idea, that form and functionality are closely correlated, influence each other and co-evolve. We suggest to transfer the same idea to hydrological systems. This allows us to separate and analyze their two main characteristics: Their form, which is equivalent to the spatial structure and static properties, and their function, equivalent to internal responses and hydrological behavior. While this approach itself is not particularly new to hydrological field research, we want to employ this concept to explicitly pursue the question of what information is most advantageous to understand a hydrological system.

Accordingly, we developed different categories to organize and describe the data presented in the two manuscripts: Structural data summarizes all sorts of data which focus on direct exploration of form, e.g. soil cores. Response dynamics, on the other hand, are observations of function. They represent processes deprived of their spatial context and include soil moisture dynamics and discharge responses. In between these two categories are flow-relevant structures and response patterns, which may contain information on both, form and function of the system.

In the presented study, we apply this concept to subsurface flow within a hillslope. The manuscript at hand methodologically focuses on function: We observed response patterns and dynamics from a natural rainfall event and during an irrigation experiment. The results are used to infer hydrological processes and the spatial organization of the monitored system. Based on these findings, the informative power and conclusiveness of the data will be discussed.

The companion manuscript focuses on form and starts off with a thorough structural exploration of the subsurface. It then proceeds towards observations of flow-relevant structures and response patterns and analyzes the information gain along this path.

**Changes in structure of the revised manuscript**

The hypotheses of both manuscripts can be aligned according to the form/function framework and will clearly be stated at in the beginning. The hypotheses of the manuscript at hand will focus on the potential of response observations for hillslope hydrological field research and the application of time-lapse GPR measurements in this context:

- **H1** Response observations (discharge, TDR and GPR data) are sufficient to characterize subsurface flow within the hillslope. (function described without form)

- **H2** Response patterns can be used to deduce flow-relevant structures in the subsurface. (function reveals form)

- **H3** Time-lapse GPR measurements visualize subsurface flow dynamics and patterns and can replace hillslope trenches. (form and function observation)

The story line was streamlined and arranged along these hypotheses to make the manuscript easier to follow and to better elaborate the important and novel key points of our study. In the following, the most important changes of each section are shortly summarized.

**Introduction**

- The catchment-scale perspective on preferential flow as well as the section about double-peak hydrographs were removed.

- The concept of form and function is introduced as the leading theme.

- Aims and focus of the study are clearly stated and the hypotheses named at the end of the introduction.

**Methods**

- All methods were subsumed under the form and function framework. To do so, an additional figure was introduced (Fig. 3).

- The methods were distinguished into stream-centered and hillslope-centered approaches. The hydrograph analysis and surface water stable isotope data belongs to the stream-centered approach, the hillslope-centered approach includes observations of subsurface responses during the natural rainfall event as well as the irrigation experiment.

- A new subsection was added to explicitly describe the GPR data processing with focus on the distinction between natural event and experimental irrigation to emphasize this aspect throughout the manuscript.

- Another new subsection deals with the stable isotope sampling of pore water and piezometers, which had been part of the companion paper in the original version.

**Results**

- The separation in catchment-scale and hillslope-scale data was replaced by the two main categories of observations under natural conditions (also including subsurface responses) and under experimental irrigation.

- A new subsection and figure were introduced, showing the subsurface response patterns of the natural rainfall event (Fig. 6). Section *3.1.2 Hillslope-centered approach: Subsurface response patterns* emphasizes the subsurface response under natural conditions.

- Pore water and piezometer isotope data are shown in a newly introduced section, also including a new figure.

- A new subsection and figure directly and quantitatively compare response patterns under natural and experimentally irrigated conditions: Section *3.3 Comparison of natural event and irrigation response patterns*, including Fig. 13.

- The figure comparing TDR and GPR data was removed from this manuscript and shifted to the companion paper.

**Discussion**

- The discussion was drastically shortened and restructured.

- The revised process discussion emphasizes the subsurface responses (rather than catchment-scale interpretations) and the synthesis of natural and experimental observations.

- The revised methodological discussion focuses on the form and function framework, as well as the role of GPR measurements within this framework.

- The new structure of the revised discussion is as follows:

    - 4.1 Process interpretation
        * 4.1.1 Irriagation experiment
        * 4.1.2 Natural rainfall event observations
        * 4.1.3 Synthesis: Function of the invetsigated hillslope
    - 4.2 Methodological discussion
        * 4.2.1 Form and function in hillslope hydrology
        * 4.2.2 2D time-lapse GPR measurements as link between form and function

**Conclusion**

- The conclusion was widely rewritten and adapted the the form and function framework.

The titles of the manuscript was adapted accordingly. It was rephrased to emphasize the methodological aspects of the paper, while keeping the focus on hillslope processes:

**Form and Function in Hillslope Hydrology: Characterization of Subsurface Flow Based on Response Observations**

We hope to have given a good overview over our thorough revisions of the manuscript. While the elaboration above was meant to provide the 'big picture', our answers to the reviewer's general and specific comments will illustrate how this concept helps to mitigate the reviewer's concerns. All line and figure numbers refer to the new manuscript with all changes tracked, which is appended at the end of this document. The color code of the changes indicates the reviewer on who's comments the changes are based (Reviewer#1, Reviewer#2, Reviewer#3), general changes and text fragments that where moved to another location are indicated in brown. In the following, our replies are inserted into the original text by the reviewer and marked in the respective color.

**Answers to the general comments by Reviewer#1**

This paper presents very interesting data from a plot/hillslope scale sprinkling experiment combining TDR observations with time-lapse GPR to investigate soil moisture dynamics. Sadly, the promised (in the title) investigation of subsurface flow and the link to catchment scale is not convincingly presented, interpretations are often overdrawn and not supported by the results, and the paper currently reads more like a patchwork of data and ideas and not like a well aligned story. During the read, it felt as if the authors were not sure of what to present and what the key message of the work can/should be.

To set the frame for a more clear story line, we elaborated a common theme for the two companion papers: The theme focuses on the informative value of observations of 'form and function' for the hydrological investigation of hillslopes. While the manuscript at hand uses process observations (i.e. response patterns and dynamics) to characterize the investigated hillslope, the companion manuscript starts with structural exploration of the subsurface only and complements these with the observed responses. This frame allows us to formulate clear and distinct hypotheses for both manuscripts and to align the story lines along the overarching and coherent theme without relying on the other manuscript too much.

For this manuscript this meant a thorough revision of the methodological discussion: The general aspects of subsurface process monitoring as presented in the original version was drastically shortened. Instead, the discussion was focused on the value and shortcoming of response observations and the lack of structural information. Furthermore, the potential of time-lapse GPR measurements is discussed in this context.

Finally, we restructured the presentation and discussion of the hydrological response data in a way to to more clearly elaborate the logical link between observations and conclusions. Namely, the separation in hillslope-scale and catchment-scale flow dynamics was transferred into a separation of observations of the irrigation experiment and of the natural rainfall event. These changes helped to focus on the relevant aspects of the study and the novelty of the presented approach.

The data is visually very well presented, but the actual analysis of the experimental data remains poor, eventually supplying temporal evolution of soil moisture (although the method combination gives an interesting depth distribution to 4 m below surface) and flow velocities for the first respond[a)]. Moreover, these velocities are not really used[b)]. From a water balance perspective it is interesting, if the occurring amounts of PF can be, by any means, linked to the generated runoff volume in events[c)]. I also think that the inclusion of one event based hydrograph separation is a little bit thin[d)].

a) In original manuscript, we focused on the temporal evolution of the responses and deliberately minimized the discussion of spatial response patterns, as this was to be discussed in the companion paper. While we chose to keep the general separation of the two parts, the foci were adapted to the new concept of 'form and function' of a hydrological system. This allowed us to discuss certain aspects of the response patterns in more detail in the manuscript at hand, without impairing the story line and arguments of the companion manuscript.

b) The response velocities were used to describe the activation of preferential flow paths. This information is surely not as meaningful as full velocity distributions. However, in the context of the subsurface and stream-centered observations from the natural rainfall event, this information is helpful to link results. Comparing these response velocities with the timing of the natural subsurface response at the hillslope and the runoff response to the rainfall event, shows that the observed flow paths are capable of causing this response. In the revised manuscript, we elaborated the line of argumentation and discussed its explanatory power as well as the calculated response velocities in more detail. Please refer to the answers to the line-by-line comments for details.

c) Unfortunately, the GPR signal can not be translated into quantitative soil moisture changes. The structural similarity attributes indicate areas of structural changes and can only be used to obtain spatial patterns and qualitative estimates of soil moisture dynamics. Although strong dissimilarity speaks for large changes in soil moisture, a quantitative relationship cannot be derived. To comply with the reviewers remark, we improved the interpretation of the GPR data, e.g. by discussing the patterns of the natural event separately (Sec. 3.1.2) and by quantifying the spatial portions of the GPR transects activated during by the natural rainfall event and the irrigation experiment (Sec. 3.3). We will also include isotope data sampled from piezometers at the irrigation site, which provide a basis for the discussion of flow proceses within the hillslope and the link between hillslope observations and the hydrograph.

d) The one hydrograph was shown to compare it with the subsurface response we could (partially) observe with our GPR measurements after the event and prior to the irrigation. As we do not have information on subsurface response for other events, we did not see an advantage in including more hydrographs, and referred to former studies instead, reporting similarly fast runoff responses in the area.

Stated by authors, the runoff ratio of 4% during the rapid peak (that consists of event water) cannot be explainable by direct rainfall on saturated areas. Nevertheless, the experimental observations and the fact that rapid flow downslope of the sprinkling site can be detected do not, in my opinion, convincingly explain the rapid stream response that is observed during rainfall in the study catchment. And this is why: I think that the method of the sprinkling experiment, with the use of a sprinkling intensity of 30mm/h, is simply not appropriate for

investigating if preferential flow reaches the stream and generates the hydrograph under natural conditions.

I do understand that the purpose was to initiate the preferential flow with this rate, but I disagree with the authors that any explanation for the stream response can be done in this way. Such high intensities will of course lead lateral flow out of the sprinkling site as soon non-vertically oriented preferential flow paths are filled, or some low permeable lense or layer is reached. It would be interesting to report the natural rainfall intensities in the area and for the particular shown runoff event. As only a total of around 20 mm occurred. I am wondering if natural intensities would be indeed able to establish, supply, and connect a preferential flow system from the hillslope to the stream under unsaturated conditions on that time scale. This would be very important, since initiation of preferential flow strongly depends on intensities. I do not doubt that PF flow might be active at the hillslope and connecting to the stream in events with 30 mm/h rainfall intensity over hours, but when do such events occur? I think that there are no data and results provided in this manuscript that showed that preferential flow at natural rainfall intensities, amounts, and duration can explain the observed behaviour.

We agree with the reviewer, that the observations of the irrigation experiment alone are not sufficient to extrapolate our findings to natural rainfall events. The conclusion, that the experimentally initiated preferential flow processes are also relevant under natural conditions, was mainly based on the strong subsurface response after the natural rainfall event seen in the GPR data. The stronger subsurface response despite lower intensity is an indicator, that lateral preferential flow occurs in the shallow subsurface of the hillslope during natural events. To better convey this line of argumentation in the revised manuscript, we present and discuss the observations from the natural event and the irrigation event separately. We included the intensities of natural rain events in the area and quantitatively compare the observed subsurface responses with regard to the areal share of activated flow paths.

Furthermore, early work of McDonnell and various follow up studies showed that much of the water in preferential flow actually is not event water. This adds further doubt to the authors' interpretation, since the observed peaks in the stream in the area are dominated by event water.

In the revised manuscript, we reference and discuss the work of McDonnell et al. and others reporting on that topic. We will also add data on the isotopic composition of water seeping into the installed piezometers. In the current version of the two companion papers, the data were presented in the companion paper to discuss the interaction between preferential flow paths and matrix. In the revised version, the data will be shifted to the manuscript at hand to support the discussion of the event water and pre-event water fractions.

Leaving the interpretation of the catchment scale interpretation and the lateral preferential flow aside, the paper does not provide much novelty from a process perspective. The depth distribution data of soil moisture is interesting from methodological perspective, but work on this was published previously (see the Allroggen references). The discussion about the processes is not supported by the results of the work, and the methodological discussion is overly long, repetitive, lacks nearly any citations, and I was not struck by a major scientific message. That just leaves a case study of a sprinkling experiment, which is - in the current form - not sufficient to be published in HESS. That said, the

dataset is very interesting and could supply interesting insights, but a clear idea of what should be presented, and what conclusions can be supported by the data is needed.

We understand, that our results and conclusions were not convincingly presented in the original version of the manuscript. The comments of the three reviewers, however, were helpful in pointing out shortcomings in the story line and we are confident that the revised manuscript clearly conveys our key points and the novelty of our results.

As stated earlier, we developed a common theme linking the two companion papers. This theme or concept follows the idea, that a system is defined by its form (i.e. spatial structure) and function (i.e. processes and behavior). Based on this concept, we pursued the questions of what information is needed or most suitable to understand the investigated system. And how much information on both, form and function, is necessary?

The irrigation experiment will be presented as an exemplary study on hillslope hydrology. In this context, the power of time-lapse GPR measurements as a non-invasive, temporally and spatially flexible method for hydrological monitoring will be discussed.

Following, an in-depth analysis is needed that goes beyond the very nice visual presentation of soil moisture data and calculated flow velocities.

We had originally hoped to obtain soil moisture changes from GPR data. However, due to the highly heterogeneous radar reflections and the absence of defined horizons, only a qualitative interpretation was possible. Due to the open system and the unknown fluxes from the irrigation area towards the GPR-monitored downhill area, assumption about mass input or the ratio between diffusive and advective soil water flow remained highly ambiguous. To avoid over-interpretation, we chose to remain at the level of recorded arrival times of the signal, response patterns and dynamics.

We elaborated the analysis of the GPR response patterns to quantitatively compare the natural event and the irrigation with regard to the spatial share of the activated hillslope cross-section. Furthermore, the dynamics of the TDR and GPR data are analyzed more thoroughly in the revised manuscript.

**Answers to line-by-line comments**

- The syntax is often awkward (except in the introduction) and could highly benefit from a very careful revision (or editing by a native speaker)
  We revised the syntax during revision, and will have the manuscript copy edited prior to final publication.

- P1L10: 'hillslope-scale connectivity'. There is no evidence provided for connectivity. There is evidence that some water moves downward from the sprinkling site. This neither indicates connectivity in this case nor connectivity under natural conditions.
  Obsolete due to a complete revision of the abstract.

- P1L11-12: 'These processes' is somewhat vague.
  Obsolete due to a complete revision of the abstract

- P2L13: Change: 'is on' to 'was on'

This was changed accordingly. P. 3, l. 14 in the new version of the manuscript.

- P2L26: Change: 'At minimal...' to: 'GPR provides...'
  Was changed:
  - before: At minimal invasive cost it provides information on subsurface structures in electrically resistive soils.
  - now: GPR provides information on subsurface structures at minimal invasive cost. P. 3, l. 28 in the new version of the manuscript.

- P3L7ff. I am not sure, if introducing multi-modal transit time distributions here and in the following parts is beneficial for the manuscript. If the authors introduce it and eventually mention it at several points, they should start calculating them for the mentioned events, rather than doing a simple two component end member mixing (although this seems not to be possible with the isotope sampling). References are needed after 'flow process'.
  We avoided the term 'multi-modal transit time distributions' in the revised manuscript to keep the focus on hillslope processes.

- P3L16ff. References for the claims are needed. Why vague?
  Obsolete due to a complete revision of the paragraph.

- P3L16ff. The research need that is outlined here, is not convincingly derived and link to the majority of the introduction. The introduction exclusively deals with preferential flow processes, before some double-peaked hydrograph behaviour is mentioned. After that, the research needs is only related to these hydrographs. As mentioned earlier, I miss the well thought out story line. I also missed references to important connectivity work, since the connectivity (hillslope-stream) part seemed to be an important part of the paper.
  The introduction was thoroughly revised and adapted to the concept of 'form and function'. The focus was kept on the investigation of subsurface flow at the hillslope and distracting aspects avoided.

- P6L10: What is the uncertainty that is associated with this very simple hydrograph separation, especially when accounting for the high variability of stable isotopes in rainfall?
  The uncertainty is given in Section 3.1.1 Stream-centered approach: Hydrograph and surface water isotopes. P. 18, l. 3ff in the new version of the manuscript.

- P7L3: Please provide the relative location of the irrigation site, when hillfoot is 0 and hilltop is 1. What is the length of the hillslope?
  The information is given as percentage of the hillslope. P. 10, l. 3-4 in the new version of the manuscript.

- P7L3: Language and syntax are improvable.
  We will have the manuscript copy edited prior to publication.

- P7L10: What is the return period of the sprinkling event (intensity, amount)?
  As the sprinkling event was not meant to mimic natural conditions, we

chose not to include the return period. Intensity and amount of the irrigation are given in the manuscript to provide a basis for comparison.

- P7L11: Here it is stated that it is not intended to mimic natural conditions, but then the authors used the results to infer flow processes under natural conditions. This does not fit together.
  The conclusion, that similar processes also occur under natural conditions is based on the observation of similar GPR reflection patterns and dynamics in response to the natural storm event. In the revised manuscript, we elaborated this line of thought and clarified the causal connection between observations and conclusions. To do so, we more clearly presented and discussed the observations made at the hillslope after the natural rainfall event. We introduced additional data analyses and figures (Fig. 6 and Fig. 13 in the fully tracked manuscript) to clearly convey the link between natural responses and the irrigation experiment.

- P7L17ff: Information about size and height above ground of the samplers are needed. What were the intensities of the natural rainfall events? This could be a good comparison to the experiment.
  The sprinkler information is given on page 10, line 10ff, the Intensities are given on page 10, line 25ff of the revised manuscript.

- P9L26-28: 'The...' Better delete? Not needed. Another suggesting: better not start sentences with 'Figure 2 shows/presents/' or 'This/that is presented in Figure 2'; rather report the finding or what the reader show see and then reference the figure.
  We applied the recommendation to the whole manuscript.

- P10L1: 'Similar'. So what is the difference to the cited work?
  The text was changed to 'as presented by...'. P. 13, l. 32.

- P10L14: 'suggested' Is certain or uncertain?
  The GPR data certainly show changes in GPR reflection patterns, which is interpreted as changes in water content. Soil moisture is not the only possible factor causing differences in GPR reflection patterns, but under the given circumstances/time scale the only reasonable one. We changed the term to indicated. P. 14, l. 13.

- P10L14: I am also wondering where the separation between mobile and stagnant water was made?
  In contrast to the TDR data, GPR data does not provide absolute soil moisture values, and all interpretations are based on differences between different measurements. As the last GPR measurement was chosen as reference, water which was still there during the last measurement will not be visible in the structural similarity patterns. We therefore can only observe changes in soil moisture, and infer information on advectively and non-uniformly flowing mobile water only. We removed the word 'mobile' here (as any water from the natural rain event was visible) and improved the description of our data analyses in more detail, e.g. in Section 2.5.3 Comparison natural event vs. irrigation based on 2D GPR data, P. 14, l. 23.

- P10L31: Why was the 2% value chosen?
  The 2% threshold was chosen based on the variability within presumably constant soil moisture measurements. This information is given in revised manuscript, p. 15, l. 7-8.

- P10L4ff: 'were calculated for the entire depth': Do you mean, that if you give the value for 10 cm it is the velocity for 0-10 cm, if you report the velocity for 20 cm depth, it is the velocity from 0-20 cm?
  That is correct. This stated in the sentence: 'Response velocities are therefore integrated values describing processes in the entire soil column above', P. 15, l. 14.

- P10L9: 'main stem'. Delete?
  This was changed throughout the manuscript.

- P12L13ff: How are these hydrographs separated? What method was chosen? This needs to be mentioned in the methods. What are the uncertainties associated with the isotope hydrograph separation?
  The method description is given in Section 2.3 'Stream-centered approach', P. 9, l. 5ff.. The sources of uncertainty are shortly given in p. 18 l. 3ff.

- P12L13ff: Why was only one event chosen for analysis?
  As mentioned earlier, we chose to show only one hydrograph, because the emphasis of this part of the discussion was on the timing between rain event and the observed subsurface GPR reflection patterns (prior to the irrigation experiment). We chose to show hydrographs of several subcatchments to prove that the behavior is not specific to the investigated Holtz river, and referenced former studies to show that the observed response was not a unique event. In the revised manuscript, we emphasized the results and discussion of the subsurface responses following the natural rainfall event rather than the hydrograph. We also highlightd the comparison and the link between the two obsrevations. To keep the story line focused on the hillslope processes, we chose not to include more events.

- P13: Strange font size/type of '20 mm' in the figure captions of fig.3.
  Was changed, see Fig. 5 in the revised manuscript.

- P13L2: 0.5 L. This was over the whole experiment? So more or less unimportant in the water balance.
  This is what we conclude later on in the manuscript. We also consider this an important finding, as surface runoff is often mentioned as a possible reason for an immediate discharge response or the event water signal in the river. We therefore chose to keep it the way it was. P. 19, l. 15.

- P13L2 'relatively': Please provide a objective description, rather than using subjective terms like small, relative, best with numbers. Do that throughout the manuscript.
  Thank you for pointing this out. We changed the wording. P. 19, l. 17.

- P13L8: Can these 20% be the uncertainty of the water balance? Or what is the actual uncertainty and how does that influence the results and interpretation? I am not sure what 'average storage deficit of 20%' means.
  The average storage deficit refers to difference between the mean storage

increase in the four core area TDR profiles and the water input. This was clarified in the revised manuscript, P. 20, l. 1. The uncertainty in the average water balance is mainly due to the low spatial coverage of four TDR profiles at the core area. It is possible that four profiles are not sufficient to fully represent the water balance of the entire irrigation site, which is why we chose to show single profiles instead of average values in Fig. 7. This source of uncertainty is mentioned in the revised manuscript (Fig. 7).

- P14: Fig4. Uncertainty needed.
  Now Fig. 7. We chose to show the data of all four core area profiles separately rather than mean values, in order to visualize the variability amongst the profiles, which is the biggest source of uncertainty.

- P15L4-7: Move to methods.
  The statement on the analysis and interpretation of time-lapse 2D GPR measurements was removed from this location of the manuscript and intensified in the methods section, e.g. Section 2.5.3 Comparison natural event vs. irrigation based on 2D GPR data.

- P15L13: 'generally low dynamics'. Please change.
  The sentence was deleted.

- P15L31: 'all fast dynamics'
  Was changed to 'no changes in GPR signal could be observed anymore', P. 23, l. 31-32.

- P15L32-33: Is there a possibility to differentiate if it is in the matrix or left the monitored area? How about working more with the water balance?
  There are several reasons why we can not calculate a water balance from GPR data. The most important one is that we can only observe structural differences between measurements. We can not tell how much water is left in the mapped transects at the end of the monitoring period, nor can we quantify the differences in terms of water content. See p. 23, l. 33 to p. 24, l. 2.

- P17L2: This is more discussion of processes then results. How was this inferred? It seems like a statement, but not supported by data. There is no evidence for this.
  We consider it as one of the results of the GPR measurements, that the natural rain event as well as the irrigation experiment caused soil moisture changes in the entire monitored soil transects down to a depth of 4.2 m. The paragraph was removed and the line of thought elaborated more clearly in the new Section 3.3 *Comparison of natural event and irrigation response patterns* as well as the Discussion.

- P17L4: Again, there is no proof for this. It is just speculative. How deep below the soil surface is the bedrock?
  This statement is based on the comparison of the GPR reflection patterns caused by the natural rain event and the artificial irrigation. It was elaborated more clearly in the new Section *3.3 Comparison of natural event and irrigation response patterns* as well as the Discussion.

We do not have reliable information on the depth of the bedrock interface and it is presumably characterized by a gradual transition between the deposit layer and solid bedrock. At the investigated hillslope, drilling was usually inhibited by high stone content at a depth of 1.7 to 2m.

- P18: Can velocity distributions be estimated? They might be more interesting/informative than just the first response time. One could see how the majority of the water moves, as the response time of the preferential flow should be somehow consistent with the time of the hydrograph response, especially if the preferential flow should explain the delivery of event water to the stream.
  Unfortunately, soil moisture changes on their own are not sufficient to calculate or estimate velocity distributions. To do so, tracer applications and the recording of breakthrough curves are required, but the piezometers did not yield enough water for continuous sampling. We agree, that the estimation of velocity distributions would be very helpful in the investigation of the role of preferential flow at the catchment-scale. However, response velocities also provide some very important information, especially in such high spatial coverage as provided by the GPR measurements.

- P22L6: This is commonly observed. Soil moisture variability increases with decreasing moisture contents
  Here, the variability (amongst the four TDR profiles) actually increases with increasing soil moisture contents, which shows that the processes are spatially discrete and heterogeneous.

- P22L12: 'high portion of mobile water'. Please quantify. This is important.
  The percentage is given in the results section. P. 20, l. 3ff.

- P22L13: 'Velocities over...' But this only relates to the fastest component. The water itself should travel with a velocity distribution, which was not estimated. I would argue that the median or average speed to the 'mobile' water (the matrix water is mobile, too) is less than 10-3 m/s. Give the numbers for MaiMai.
  We agree that the median or average *real* velocity of the water surely is much lower than the given maximum *response* velocity of $2.2 \cdot 10^{-3}\, ms^{-1}$. The response velocities shown in Fig. 12 (in the fully tracked revised manuscript) range between $10^{-4}\, ms^{-1}$ and $10^{-3}\, ms^{-1}$ (based on GPR and TDR measurements). Any water arriving after the first response cannot be used to quantify volumes or velocities, because (a) we can not reliably quantify the newly arriving water based on the soil moisture state of the monitored soil volume and (b) we do not know the 'start time' of the water, which might be somewhere between irrigation start and measuring time.
  The intention here was to investigate the potential of preferential flow for the immediate runoff response. We use the maximum response velocities to show that the response velocities occurring in preferential flow paths are sufficient to route water from the hillslopes to the river within the short time between rain event and discharge response. The fast decrease in soil moisture after irrigation stop shows that a large fraction of the water is

highly mobile and leaves the monitored depth at velocities that exceed the potential matrix flow velocities.

This was clarified in the revised manuscript.

Velocities for MaiMai are provided in the revised manuscript. P. 32, l. 19-20.

- P22L16ff. This is true, but it still does not show water transport of volumes to the stream that might explain the hydrograph response. Is there impermeable/less permeable material below the field site? How deep is the bedrock? Was an attempt made to observe the subsurface stormflow there?
  Unfortunately, we cannot locate the depth of the bedrock interface, but maximum possible drilling is usually at depths between 1.5m to 2m at the investigated hillslope. The application of time-lapse GPR measurements was clearly motivated by our will to identify preferential flow layers if they existed, however, we could not find any. This was added to the Discussion section. The companion paper furthermore explicitly deals with the identification of subsurface structures and their relevance for flow processes. P. 42, l. 33.

- P22L16ff. In addition: The activation of the preferential flow is highly dependent on the rainfall intensities. They might never be active under natural conditions when rainfall intensities are an order of magnitude lower (over event scale).
  Based on the strong signal observed in the GPR measurements between natural rain event and irrigation, we conclude that there was an even stronger response of preferential flow caused by the natural event despite lower intensity and shorter duration. This finding was thus not solely based on the experimental observations. The line of argumentation was obviously not clear enough and was elaborated in the revised manuscript. In the revised manuscript, we present and discuss the experimental results and the natural response (including the GPR responses) separately and compare the two different responses by means of a newly introduced measure. See Sections 2.5.3, 3.3, and changes in Section 4.1

- P22L24-25: Connectivity at hillslope scale was not shown in this experiment. The data simply showed that some preferential flow out of the sprinkling site occurred.
  We agree that the experimental results alone do not prove hillslope-scale connectivity. As mentioned above, this statement is based on a synthesis of experimental results and the observed response triggered by the natural rain event. Unfortunately, we can not translate the GPR signal into quantitative soil moisture changes and provide a mass balance of the monitored transects. But the strength of the natural signal in relation to the experimentally initiated response, clearly indicates a higher amount of water despite lower local input. We introduced a method to quantitatively compare the observations from natural rainfall event and irrigation (Sec. 2.5.3 and 3.3).

- P22L30: Every hydrograph is likely a result of multi-modal distribution of travel times...

The entire section was deleted and the discussion restructured to focus the manuscript on hillslope processes rather than catchment response.

- P23L20: How were the riparian zones and wetland patches delineated? What is the uncertainty? Is there a contraction/extension during events that could change this?
  The wetland patches are very small and restricted to the source area of the river. A larger patch which is not in direct proximity of the river was mapped in the field once. The rest was estimated based on field observations. This is just a rough yet conservative estimate, and we can not exclude contraction/extension of this area. This information was included in the revised manuscript. P. 34, l. 32ff.

- P23L22ff. Again, there is no evidence shown in this work that there was any connectivity between hillslope and stream. This is even more true when natural conditions are considered.
  As mentioned above, we elaborated the line of argumentation regarding the observed hillslope processes more clearly in the revised manuscript (see replies to comments on P22L16ff and P22L24-25). We will also focus more on hillslope processes rather than catchment-scale interpretations. By doing so, we will stay within the actual scope of our dataset and avoid over-interpretation.

- P24L26ff. This methodological discussion is overly long, repetitive, and lacking references to work of others.
  The methodological discussion was significantly shortened and streamlined to emphasize the novelty of the method rather than discussing general aspects of soil moisture measurements and subsurface exploration.

- P24L27-P25L3: Here I disagree. As said, the experiment was not designed to mimic natural conditions. This is completely fine, since it seems to me that the original idea was to employ the GPR system. Nevertheless, the setup does not allow inferring natural flow processes that explain hydrograph response.
  The experiment was designed to visualize potential preferential flow paths. The natural rain event, however, allowed us to put our observations in the context of naturally occurring responses. The analysis of similarities and differences between the response patterns and dynamics allows for conclusions on naturally occurring processes.

  We agree that the presentation in the original manuscript was misleading. We shortened the discussion of the general experimental setup and shifted the focus of the entire manuscript towards hillslope processes and an evaluation of the approach within the form and function framework.

- P25L10: Explain: 'sufficiently steady state'
  The paragraph was removed and the term avoided in the revised manuscript.

- P25L24: Sat. conditions are most likely occurring at the bedrock interface. Technically the periglacial layers are not 'soils' anymore. Try to have a look at the work of Uhlenbrook et al. on periglacial slopes.
  The paragraph was deleted and the wording and interpretation handled more carefully throughout the manuscript. We also corrected the use of

the work 'soil' accordingly.
We will also be more careful with the use of the word 'soil'.

- P27L31ff. This seems to be unnecessary - no important point was made in this section.
  The methodological discussion was shortened and the section removed.

- P29L4: The resulting connectivity was not shown and remains highly speculative.

  P29L5ff: This is speculative. Furthermore, the link to transit time distributions is somewhat strange.

  P29L20: Again, I think this was not shown here
  Summarizing the three comments above, we are very sorry that some key points were not well conveyed in the original version of the manuscript. As mentioned earlier, we revised the manuscript and restructured it to better elaborate these points. We checked and avoided the use of the terms 'connectivity' and 'transit time distributions' and focused our presentation and our conclusions on the hillslope processes. In the revised manuscript, hydrograph and surface water isotope observations are discussed more carefully and with a clear relation to the hillslope observations.

**Answers to the general comments Reviewer#2**

This paper presents an interesting experimental approach, combining different types of measurements and techniques (soil moisture monitoring, time-lapse GPR, discharge and tracer measurements) to investigate mechanisms of rapid subsurface flow. Such combinations are often very promising as they can yield new interpretation of observations and new insight. The visual presentation of results, i.e. the figures, is very well done, the figures are informative and mostly well structured. The language is mostly suitable, with sentences sometimes being quite long. Here a language revision would certainly help to make the text more pleasant to read.

However, I do not think that the authors achieve their objective of investigating and providing new insights into rapid subsurface flow processes. As they themselves state in the Conclusions, the link between processes and structures is still missing, many statements remain speculations and thus, an exciting new insight is not really presented.
In their original version, the two companion papers were separated by their foci on temporal dynamics on the one hand (manuscript at hand), and spatial structures on the other (companion manuscript). Therefore, the discussion of the link between processes and structures was deliberately avoided here and left to the companion paper.
In the course of the revision, the foci of the two papers was shifted towards a more methodological perspective. While we pursue the question of how much information we can get from pure response observations in the manuscript at hand, the companion manuscript starts off with structural explorations of the subsurface. We subsumed this conceptual approach under the term 'form and function', applied to hydrological systems. According to this concept, the link between processes and structures is still part of the findings of the companion

manuscript. However, the informative value of observed response dynamics and patterns and the usefulness of structural information is discussed more thoroughly also in the revised manuscript at hand.

The experiments were no doubt labor-intensive and comprehensive, and I can understand that the authors would like to present them in their entirety. I have the impression, though, that the authors were not quite sure about the focus of the manuscript. Title and abstract speak of rapid subsurface flow and identifying preferential flow features whereas in the introduction and then later in the discussion the focus shifts to double-peak hydrographs as if the explanation of these had been the aim of the paper. The abstract does not indicate this. In between also (multi-modal) transit-time distributions are mentioned several times but surely the experiment was not designed to investigate transit time distributions. The discussion is lengthy and focuses on methodological aspects.

We understand that the story was not well presented in the original version of the manuscript. Based on the comments of the three anonymous reviewers, we decided on a common theme for the two companion papers and streamlined the story lines of both manuscripts. The revised manuscript will focus on the investigation of subsurface flow processes within the hillslope with a methodological perspective. The presented hydrographs are still a minor part of the study to provide a temporal framework for the subsurface flow processes observed after the rain event. The discussion of double peaks and (multi-modal) travel time distributions, however, was minimized or avoided.

At the moment I doubt that the manuscript presents sufficient new data and findings beyond speculations to justify a publication. Maybe it would be an option to combine this manuscript and the mentioned companion paper by Jackisch et al as they present the missing link between processes and structures as stated in the Conclusions, last paragraph.

The companion paper by Jackisch et al. presents many experiments and results which were not introduced in the manuscript at hand and does not include all of the data presented here. The presentation of the entire set of methods, results and conclusions clearly exceeds the frame of one paper. We therefore stuck to the separation in two manuscripts and revised both of them. We clearly elaborated their specific and distinct perspectives, while also allowing slightly more overlap than in the original version so that the links between processes and structures can be pointed out.

**Answers to line-by-line comments**

- P. 2, L 3: preferential flow and rapid subsurface flow are not per se synonyms
  Now p.3, l.4: The annotated sentence was deleted and the use of both terms checked throughout the manuscript.

- P. 3, L 12: if the explanation of double-peak hydrographs is indeed the motivation, this should be mentioned in the abstract
  The story line of the manuscript was revised in a way, that the focus is on the hillslope response. Catchment-scale aspects and the discussion of double-peak hydrographs was reduced and the abstract was revised accordingly.

- Section 2.1/2.2 and 3.1: I was a little confused that first it seemed to be a study focusing on the irrigation experiment whereas later on also longer-term behavior is described as result. I would only focus on description of the event and potentially, move descriptions of the general hydrological behavior of the catchments to the site description.

  As suggested, the first paragraph of Section 3.1 was removed and the necessary information is given in the site description now.

- Section 2.3.2: I had difficulties to exactly understand the TDR measurements with access tubes and would recommend to describe the soil moisture monitoring more clearly. Could you insert the prongs of the moisture probe via the access tubes in different depths?

  The used Trime Pico IPH probes (IMKO GmbH) do not have prongs, but aluminum plates as TDR wave guides. These plates are mounted on opposite sides of the probe, pressing against the access tube and measuring through the PVC access tube. We clarified our description of the procedure. P. 12, l. 1-9 of the revised version of the manuscript.

- P. 8, L 1: predominantly

  Was corrected. P. 11, l. 7.

- P. 8, L 6: which three TDR probes? Unclear

  This will be clarified three lines above:

  - before: 'Two versions of the TDR sensor were employed, with either 0.12 m or 0.18 m integration depth.'

  - now: 'Soil moisture was measured manually. To increase temporal resolution of the measurements, three probes were used in parallel. While these probes were identical with regard to measuring technique and manufacturing, they differed slightly in sensor design: two TDR probes had an integration depth (i.e. sensor head length) of 0.12 m, one probe had an integration depth of 0.18 m.' P. 12, l. 1ff.

- Fig. 2: I would not show the location of the piezometers here as you do not discuss any results. That may only be confusing. You can still mention in the text that you installed them but they measured any transient water tables. Also, it would help if you could indicate the three diverging transects in the figure.

  Now Fig. 4: We included the isotope data from water seeping into these piezometers in the revised manuscript, which were presented in the old version of the companion paper before. Due to these changes, the piezometers' positions are of relevance in the revised manuscript. We therefore still include them in the map of the revised manuscript.

  The three diverging TDR transects are included in Fig. 8 of the revised manuscript (formerly Fig. 5) and the central transect is indicated in the revised Fig. 4.

- P. 9, L1: ...transect of 6 piezometers 'intended to observe transient water levels'...

  The sentence was deleted and the description of the piezometers shifted to a new subsection (*2.4.3 Isotope sampling*).

- P. 9, L 28-29: add numbers of soil moisture profiles here to make sure it is clear to the reader which profiles you are referring to
  Numbers were added. P. 13, l. 26.

- P. 13, L 1: intensity of irrigation stated here is different to intensity stated in the methods section
  This discrepancy is based on the three different monitoring methods we applied: a flow meter, a tipping bucket, and 42 rain collectors. While the flow meter and tipping bucket are considered more reliable with regard to absolute amount, the rain collectors were mainly used to evaluate the homogeneity of the irrigation intensity. We related the measurements to each other and corrected this discrepancy. P. 10, l. 13 and p. 19, l. 15.

- P. 13, L 1: when did surface runoff cease?
  Surface runoff ceased approximately 20 min after the end of irrigation. We included this information in the revised manuscript. P. 19, l. 16.

- P. 13, L 4-5: I am not sure what is meant with 'overshoot'. Is this shown in Fig 4? Please explain more clearly.
  The sentence was rephrased and the observation explained more clearly: ' All profiles showed a mass recovery of more than 100 % (i.e. higher storage increase than water input at measuring time, see Fig. 7) in the first 60 min of the irrigation period.' P. 19, l. 18.

- P. 15, L 12: I would recommend to make the way times are stated consistent between text and figure (3.3 hours vs 03:18)
  This was done.

- Fig. 5: where is the map for TDR 5?
  Now Fig. 8: TDR 5 showed only very low soil moisture dynamics. It was left out to leave more space in the whole-page figure for the 'more interesting' profiles.

- Fig. 5: And the decreases in soil moisture shown e.g. in the lowest row of profiles is not really discussed in the text.
  The decrease in soil moisture can be attributed to the fading signal of the natural rain event. However, it is minimal compared to the soil moisture increase observed elsewhere and close to noise level. We added this information to the revised manuscript. P. 21, l. 14.

- I do not fully understand how the authors distinguish in the GPR maps between natural event water and irrigation water. Can that be done with the GPR signal? Please explain in a little more detail.
  The discrimination was made based on the dynamics of the GPR signal. We assume that the irrigation is causing any (significant) increase in soil water content after irrigation start. This assumption is not per se valid, but in our case supported by the increasing signal strength and the dynamics after irrigation start, as well as the dynamics of the TDR measurements. On the other hand, measurements taken before irrigation start show all areas activated by the natural rainfall event. Thus, areas where soil water content is decreasing during the irrigation are interpreted as flow paths that were only activated during the natural rainfall event,

but not during irrigation. We elaborated the description of this procedure and added a new subsection (*2.5.3 Comparison natural event vs. irrigation based on 2D GPR data*) to make it clear to the reader. It is also shortly summarized in the caption of Fig. 9.

- P. 17, L 2-3: is this a speculation or really supported by the results?
  We clearly see the fading signal after the natural rain event as well as the increase and decrease initiated by the irrigation. Also, the patterns are distributed across the GPR transects. While we cannot specify the size and characteristics of these distributed structures, a groundwater table or flow layer would have caused a clear and more organized signal in a specific depth. The paragraph was deleted and the observations described more clearly in the new section *3.3 Comparison of natural event and irrigation response patterns* as well as the Discussion.

- P. 17, L 5-6: maybe better say 'water that was supplied from upslope areas was more important...'
  The phrasing was changed as suggested. P. 26, l. 22.

- Fig. 6: there seems to be one map missing of transect 4 in the lower right corner where the legend is located?
  Now Fig. 9: The map was included.

- P. 22, L 13: agreed instead of fitted
  Was changed accordingly. P. 32, l. 16.

- P. 22, L 24-25: again, is this a speculation?
  This is the interpretation of the response patterns in both, GPR and TDR results. The 2D GPR reflection patterns clearly show that there is a heterogeneous soil moisture response, but no groundwater table or defined flow layer. The heterogeneity between the TDR profiles as well as the delayed signal in the intermediate depths of the core area profiles also imply preferential flow. We agree, that any interpretations beyond these findings are speculative. We left them to the companion manuscript, where they are backed by structural information. The annotated paragraph was deleted and the discussion of these observations moved to a new section (*4.1.3 Synthesis: Function of the invetsigated hillslope*), where they are discussed more carefully.

- P. 22, L 30: here now it sounds as if the investigation of double-peak hydrographs is the main objective of the paper. This comes as a surprise as it was not mentioned neither in the abstract nor in the title.
  The entire section was strongly revised and renamed (section *4.1.2 Natural rainfall event observations*) to keep the focus on hillslope-scale observations. The hydrograph is now only used as a reference to evaluate the hillslope-scale observations and the discussion of the catchment-scale observations was reduced to the necessary and helpful amount.

- I am not sure that from these measurements of soil moisture at discrete points in time lateral flow can really be inferred.
  We agree that discrete measurements in time alone are per se not useful to infer lateral flow. However, our experimental setup allows for a clear

separation between observations at the irrigated area and the downhill monitoring are. Due to the lack of water input at the downhill monitoring area, all increases in soil moisture downslope of the rain shield are the result of lateral flow or a combination of lateral and vertical flow. Thus, the spatial arrangement of measurements provides the necessary context to interpret the observed dynamics.

- P. 24, L 28-33: I do not think the reasoning here for the high input rates in convincing. I wonder how useful the observed patterns are for more realistic rainfall conditions.
  The discussion of the general experimental setup and monitoring was drastically shortened and the annotated paragraph deleted. Instead, the presentation and discussion of the subsurface responses following the natural rainfall event were improved to allow for an evaluation of the results of the irrigation experiment.

- P 26, L 11: soil moisture should be used with caution as indication for flow, and this has been discussed in the literature before
  The discussion of general methodological aspects was drastically shortened and the paragraph deleted.

- P26, L 19-24: this is not new
  As mentioned before, the discussion of general methodological aspects was drastically shortened and the paragraph deleted.

- Conclusions: I would not mention transit-time distributions here. That surely goes beyond the focus of the paper and may be a stretch to infer.
  The discussion of catchment-scale observations was reduced to the aspects which are important to evaluate the hillslope-scale interpretations. Thus, the discussion of transit-time distributions was widely removed from the revised manuscript. Thanks for pointing this out.

- P29, L 14: stable
  Was corrected. P. 42, l. 16.

**Answers to the general comments**

This study is designed to investigate subsurface flow processes, and in specific preferential flow, at two different scales. The authors chose a multi-method approach including quantitative soil moisture measurements using TDR (time domain reflectometry) and qualitative soil moisture measurements using GPR (ground penetrating radar) during an irrigation experiment at the hillslope scale. The experiment was complemented with common hydrometric and tracers techniques (interpretation of the rainfall runoff graph and stable water isotopes of rainfall and runoff) at different catchment scales. The study addresses an important topic in the hydrological sciences. The manuscript is clearly structured following common scientific standards. The data are presented in (mostly) clearly arranged, high quality figures. The combination of TDR and GPR measurements provide a very promising approach to monitor subsurface processes. While this approach is relatively novel, it bears a number of uncertainties and

disadvantages, which are discussed at length (although the methodological discussion lacks a bit in references).

However, I do have some concerns. First of all, clear objectives were not formulated in the beginning of the manuscript and it follows that the rest of the story misses a central theme.

We developed a theme to structure the manuscript more clearly and also improve the link to the companion paper. Based on this theme, we formulated clear hypotheses and state them at the beginning of the revised manuscript. The title, abstract and story line was revised and adapted accordingly.

I do understand though, that the catchment scale analyses revealed that the system responded with a double-peak hydrograph to water input with the first, steeply and quickly responding hydrograph consisting of a mixture of new and old water and the second, dampened hydrograph consisting dominantly of old water. The results from the irrigation experiment were then used to explain the hillslope processes that may potentially lead to this catchment scale response. This leads to my second and third concerns: (2) I feel that only one hydrograph separation to calculate the event and pre-event water fractions of a catchment is not representative. Many studies have shown in the past that the event water fraction may vary largely in one catchment depending on various factors, e.g. water input and/or antecedent moisture conditions [e.g. Munoz-Villers & McDonnell, 2012]. Hence, I suggest including more data and/or an uncertainty analysis to add representability to the HS.

The one hydrograph was shown to temporally relate the observed subsurface response patterns after the natural rain event (the green portion of the GPR data shown in Fig. 6 and 9) to the discharge behavior. The timing of the two observations, the subsurface response on the one hand and the discharge response on the other, allows us to draw conclusions on the processes within the hillslope. In the original version of the manuscript this line of thought was not properly conveyed. We improved the story line by presenting and discussing the responses to the natural rain event and the irrigation separately. A quantitative comparison of the GPR response patterns of both events (the natural and experimental) will emphasize the focus on subsurface flow processes rather than catchment-scale responses. We agree that the fraction of event water in the first peak is likely to vary with rainfall characteristics and antecedent conditions and avoided general statements based on this single event in the revised manuscript.

And (3) the authors clearly point out that they did not intend to mimic natural conditions with the irrigation experiment, however, they use the results to explain the natural response at the catchment scale. While this shortcoming is mentioned in the methodological discussion, I don't feel that this justification is sufficient to link the observed processes in the irrigation experiment to the natural conditions at the catchment scale.

The conclusions about relevant processes under natural conditions was backed by the observations of the subsurface response to the rainfall event prior to the experiment. This was not clearly elaborated in the original version of the manuscript and was improved as described above. We also shifted the focus of the manuscript towards the methodological aspects of GPR measurements for hydrological monitoring and the informative power of response observations for the characterization of hillslopes.

At last, I feel that the novelty of this study is not clearly conveyed to the

reader. I think it is the combined use of TDR and GPR measurements which bear a large potential to move forward in subsurface flow process understanding. However, this message needs to be presented more clearly.

We drastically shortened the basic methodological discussion and put a stronger focus on the methodological aspects related to the GPR measurements and the form and function framework described above. By doing so, the novelty of our methodological approach was emphasized and distracting side tracks and overly long discussions were avoided.

**Answers to the detailed comments**

**Objectives and experimental approach**

- I think that this subsection needs a bit more structure and/or more precise phrasing. Maybe name all objectives at the beginning of the paragraph and then list the approaches that you chose to address them in addition to the explanations why you chose said approaches (advantages vs disadvantages of others).

  To improve the story line of the manuscript, we developed a concept to organize and subsume the approaches and methods presented in the two companion papers. According to this concept, parts of the introduction, including the annotated subsection, were completely rewritten. In the revised manuscript we clearly state the aim and approaches of our study and name the three hypotheses at the end to clearly set the frame for the presented study.

- Page 3, line 24: the role of what? Please add information because otherwise this sentence and subsequent paragraph come a bit out of the blue, i.e. raise further questions, e.g. why is it necessary to use a multi-scale approach?

  We fully revised this subsection and deleted the annotated sentence.

- Page 3, line 25: What are the conventional hydrological methods that you mention? I suppose the TDR measurements that you chose as approach in your experiment? Please add this information to the text or rephrase.

  The information is given in the revised introduction. P. 4, l. 6-7 of the revised manuscript.

- Page 3, line 27: you only propose GPR, why do you not mention the other techniques? (Maybe use this 3 comments to elaborate your objectives.)

  In the revised manuscript we point out that we choose GPR due to its high spatial resolution and the short measurement times, which could not be achieved with other methods such as ETR. Beyond that, we decided not to go into too much detail and did not add a more thorough review about geophysical methods.

**Methods**

**Hydrological response monitoring**

- Page 6, line 22: How exactly was rainfall water sampled for isotope analysis? Was a sequential sampler used or is it a bulk sample? Please add information.
  This information is provided in the revised manuscript. P. 9, l. 1. and p. 13, l. 6.

- Line 24 (and everywhere else in the ms): I suggest to use 'hydrograph separation' instead of 'mixing model', the latter implies that an EMMA was performed which is not the case.
  The wording was corrected, thank you for pointing this out.

**Process monitoring**

- Page 8, line 3: Please add information why different TDR sensors were used.
  We wanted to increase the measurement frequency by measuring with three sensors in parallel. As only these three sensors were available to us, we decided that the benefit of higher frequency measurements out-weighed the disadvantages caused by the different sampling volume of one of the sensors. This information was added in the revised manuscript. P. 12, l. 1.

- Page 8, line 6: Which 3 TDR tubes were used? Sounds like they were mentioned before which they aren't. Please rephrase or add information for better understanding.
  The paragraph was completely revised for better understanding. P. 12, l. 1ff.

**Results**

- Figure 3:

  there is too much information in this Figure. I suggest moving some of the information to e.g. a separate table. For example, the information of the catchment sizes and the % of the precipitation amount should be placed somewhere else, maybe together in one table. You may mark the 7th and 72th hr in the RR-Graph and refer the reader to the table. It also remains unclear to me why the information about the irrigation experiment (i.e. the mean structural similarity attribute) are placed in the rainfall-runoff and chemographs of the headwater catchments. Which y-axis does the mean structural attribute refer to? And what do the vertical bars mean? Please provide this information in the legend. I suggest considering making two different graphs, for example below each other in the manner of Figure a) catchment scale and b) hillslope scale. This should help to understand this figure faster and identify/extract important information more easily.

Now Fig. 5: The reason why we put the GPR dynamics into the hydrograph/chemograph plot was to show the temporal relation between the different observations. We understand from the reviewers' comments, that this aspect was not clearly conveyed in the original version of the manuscript and revised the text accordingly. Thus, we also kept the different data together in one plot, but took some measures to make it easier to grasp: We removed the catchment sizes and event runoff portions from the legend and moved the information to an extra table. We also mention in the caption that the GPR data is shown to demonstrate the temporal relation between subsurface response and discharge response and refer the reader to another, new figure, where these data are shown and described in more detail. The new figure is also referenced in the legend. We furthermore added another y-axis for the GPR data. The figure is now better embedded In the context of the revised manuscript.

Please also provide information about where groundwater was sampled for d18O, e.g. in section 2.2.
This information is given in the methods section (p. 9, l. 2.) and was added to Fig. 2.

- Page 12, lines 7: Since there are several peaks in Figure 3, it would make it easier for the reader if you added the date of the described rain/hydrograph event.
  The events were labeled for clarification. The labels are also given in Fig. 5 of the revised manuscript.

- Page 13, line 5: What is an 'overshoot' in mass recovery? Please add a bit more information to explain this.
  The sentence was rephrased: 'All profiles showed a mass recovery of more than 100 % (i.e. higher storage increase than water input at measuring time, see Fig. 7) in the first 60 min of the irrigation period.' P. 19, l. 18.

- Page 14, lines 19-20 and Figure 5: The weak signal of soil moisture dynamics in TDR 13, 6 and 14 is apparent in Figure 5. While an explanation is provided for the signal in TDR 6, I am missing explanation for TDR 13 and 14. How do you explain this? I suggest including this in the discussion section 4.1.
  The behavior of TDR14 and TDR13 is mentioned in the revised manuscript. P. 21, l. 14-16 and P. 34, l. 5-7.

- Figure 5: Soil moisture change is interpolated over time for visual reasons (according to figure caption), I think it would be helpful to add this information somewhere in the methods section (e.g. 2.4.1. TDR data analysis).
  We agree and added this information: 'While the data was interpolated in time for better visualization, all data analyses were performed with the uninterpolated data.' P. 13, l. 21.

**Discussion**

- Process interpretation: What is the novelty about this study? I suggest elaborating this e.g. add a paragraph to emphasize this a bit more clearly.

The strength of our two-part study is the discussion of various data sources and their contribution to process understanding in the context of the interplay of form and function (a known concept here transferred to hillslope hydrology). In the manuscript at hand, the focus is on the observation of response patterns with GPR and TDR measurements. The combined use of these methods and their evaluation in the context of form and function of a hillslope as well as an alternative to the classical hillslope trench is the novelty of this study. To emphasize this aspect more clearly, we developed the above-mentioned concept for the two companion papers, highlighting two different methodological approaches. Both approaches include GPR measurements, which allows discussing the method under different aspects. According to this new concept, (methodological) discussion of the manuscript was completely revised.

- Page 22, lines 22-25: I do not understand the link between delayed signal in the intermediate depth and the network of preferential flow paths. Please add a bit more text.
  The interpretation is explained in more detail in the revised manuscript. P. 32, l. 30ff.

- Page 23, lines 22-29: So, does that mean that the first peak is generated by overland flow (as observed by Wrede et al.) AND shallow subsurface flow? Or is all of the event water conveyed via subsurface flow? Please express your interpretation a bit more clearly.
  We can not make any statement about the importance of overland flow for runoff generation beyond our observation, that surface runoff is of minor importance at the hillslopes, and that the areal share of saturated areas in the catchments is small. We therefore assume that the fraction of surface runoff is small. On the other hand, our results from the hillslope experiment show that the subsurface hillslope response is indeed fast and a potential source for mobile water. We thus conclude that preferential flow in the shallow subsurface at least contributes to the first peak.
  We clarified these interpretations and adapted the formulation of our findings accordingly throughout the discussion section.

**Technical corrections**

- Figure 4: Can you move the y-axis name and unit away from the numbers, e.g. to the very top or remove 140 and insert 120 instead?
  Now Fig. 7: This was changed accordingly.

**Typing errors**

The typing errors were corrected. Thank you very much for pointing them out.

**In situ investigation of rapid subsurface flow: Temporal dynamics and catchment-scale implicationForm and function in hillslope hydrology: characterization of subsurface flow based on response observations**

Lisa Angermann[1], Conrad Jackisch[2], Niklas Allroggen[3], Matthias Sprenger[4,5], Erwin Zehe[2], Jens Tronicke[3], Markus Weiler[4], and Theresa Blume[1]

[1]Helmholtz Centre Potsdam, GFZ German Research Centre for Geosciences, Section Hydrology, Potsdam, Germany
[2]Karlsruhe Institute of Technology (KIT), Institute for Water and River Basin Management, Chair of Hydrology, Karlsruhe, Germany
[3]University of Potsdam, Institute of Earth and Environmental Science, Potsdam, Germany
[4]University of Freiburg, Institute of Geo- and Environmental Natural Sciences, Chair of Hydrology, Freiburg, Germany
[5]University of Aberdeen, School of Geosciences, Geography & Environment, Aberdeen, Scotland, UK

*Correspondence to:* Lisa Angermann (science@lisa-angermann.de)

**Abstract.**

The term *form and function* was established in architecture and biology and refers to the idea, that form and functionality are closely correlated, influence each other and co-evolve. We suggest to transfer this idea to hydrological systems to separate and analyze their two main characteristics: Their form, which is equivalent to the spatial structure and static properties, and their function, equivalent to internal responses and hydrological behavior. While this approach is not particularly new to hydrological field research, we want to employ this concept to explicitly pursue the question of what information is most advantageous to understand a hydrological system. We applied this concept to subsurface flow within a hillslope, with a methodological focus on function: We conducted observations during a natural storm event and followed this by a hillslope-scale irrigation experiment. The results are used to infer hydrological processes of the monitored system. Based on these findings, the explanative power and conclusiveness of the data is discussed. The measurements included basic hydrological monitoring methods, like piezometers, soil moisture and discharge measurements. These were accompanied by isotope sampling and a novel application of 2D time-lapse GPR (ground-penetrating radar). The experiment revealed fast establishment of preferential flow paths at the hillslope-scale despite unsaturated conditions. We found that response observations were well suited to describe processes (and thus, the function) at the observational scale. Especially the combination of a stream-centered and hillslope-centered approach allowed us to link processes and put them in a larger context. Transfer to other scales beyond observational scale and generalizations, however, rely on the knowledge of structures (form) and remain speculative.

~~Preferential flow is omnipresent in natural systems. It links multiple scales from single pores to entire hillslopes and potentially influences the discharge dynamics of a catchment. However, there is still a lack of appropriate monitoring techniques and thus, process understanding. In this study, a promising combination of 2D time-lapse ground-penetrating radar (GPR) and soil moisture monitoring was used to observe preferential flow processes in highly structured soils during a hillslope-scale~~

irrigation experiment. The 2D time-lapse GPR data were interpreted using structural similarity attributes, highlighting changes between individual time-lapse measurements. These changes are related to soil moisture variations in the subsurface. In combination with direct measurements of soil moisture, the spatial and temporal characteristics of the resulting patterns can give evidence about subsurface flow processes. The response dynamics at the hillslope were compared to the runoff response behavior of the headwater catchment. The experiment revealed fast establishment of preferential flow paths at the hillslope-scale despite u These preferential flow processes substantially impact the overall catchment response behavior. While the presented approach is a good way to observe the temporal dynamics and general patterns, the spatial characteristics of small-scale preferential flow path could not be fully resolved.

**1 Introduction**
* * *
**1.1 Methodological challenge of capturing subsurface flow**

 The specific challenge of investigating preferential flow lies in its manifestation across scales, its high spatial variability, and pronounced temporal dynamics. A considerable number of experimental and model approaches have been proposed to investigate the issue (Beven and Germann, 1982; Šimůnek et al., 2003; Gerke, 2006; Weiler and McDonnell, 2007; Köhne et al., 2009; Beven and Germann, 2013; Germann, 2014). However, rapid flow in structured soils is still a challenge to current means of observation, process understanding and modeling.

In previous studies at the hillslope-scale, the focus  was often on lateral flow processes and the establishment of overall connectivity. Hillslope-scale excavations yield information on spatial extent and characteristics of preferential flow paths in 3D (Anderson et al., 2009; Graham et al., 2010), but are highly destructive and lack the temporal component. Hillslope-scale tracer experiments in contrast, resolve temporal dynamics and velocities (Wienhofer et al., 2009; McGuire and McDonnell, 2010), but lack the spatial information. Hillslope-scale experiments are usually very labor intensive and require high technical effort and most studies are concentrated on well-monitored trenches (McGlynn et al., 2002; Tromp-Van Meerveld and McDonnell, 2006; Vogel et al., 2010; Zhao et al., 2013; Bachmair and Weiler, 2012). Blume and van Meerveld (2015) give a thorough review of investigation techniques for subsurface connectivity and find experimental studies on this topic underrepresented in hydrological field research.

In recent years, a trend towards non-invasive methods for hillslope-scale observations emerged (Gerke et al., 2010), which has been an important improvement with regard to repeatability and spatial and temporal flexibility of observations (Beven and Germann, 2013). In this context various geophysical methods have been applied for subsurface exploration (e.g. Wenninger et al., 2008; Garré et al., 2013; Hübner et al., 2015). From all applied geophysical techniques ground-penetrating radar (GPR) is known as the tool providing the highest spatial and temporal resolution.  GPR provides information on subsurface structures at minimal invasive cost (e.g. Lambot et al., 2008; Bradford et al., 2009; Jol, 2009; Schmelzbach et al., 2011, 2012; Steelman et al., 2012). Its short measurement times and high sensitivity towards soil moisture predestine GPR for monitoring subsurface flow. Nevertheless, only few field studies exist which have successfully applied surface based GPR for the investigation of subsurface flow paths (Truss et al., 2007; Haarder et al., 2011; Guo et al., 2014; Allroggen et al., 2015b).

**1.2 Experimental approach with focus on responses and function**

In the present study, we aim to characterize preferential flow within a hillslope. We chose a combination of conventional hydrological methods and non-invasive GPR measurements to explore flow processes by means of observations at the hillslope (hillslope-centered approach according to Blume and van Meerveld, 2015). This hillslope-centered approach was supported by stream-centered process observations, including a basic hydrograph analysis and surface water stable isotope sampling during the natural rainfall event. The hydrological methods at the hillslope include surface runoff collectors, a dense network of soil moisture observation profiles, stable isotope samples and piezometers. They were accompanied by 2D time-lapse GPR measurements, which were chosen due to the method's high spatial resolution and short measuring times, allowing for high spatial and temporal flexibility (Allroggen et al., 2015a).

Besides the insights in preferential flow at the hillslope scale, we also want to use this example to analyze the explanatory power of the different observations. To do so, we developed the form and function framework to subsume our data and observations as shown in Fig. 1. The idea of the form and function dualism was established in architecture (*form follows function* Sullivan, 1896) and commonly used in biology (e.g. Thompson et al., 1942) and describes the link, mutual influence and co-evolution of the outer appearance and functional purpose of a (research) object.

Here, form includes all static properties and spatial structures, such as topography, geology and subsurface structures, but also porosity, hydraulic conductivity and stone content of the soil. Function summarizes all dynamics and processes, including soil moisture dynamics, discharge behavior and preferential flow. These two are closely related and co-evolve. Based on this idea, Sivapalan (2005) suggested, that patterns, responses and functions are the basic key to understanding and describing a hydrological system, as they incorporate the morphogenetic processes that led to the spatial structures. While this approach refers to the larger scale and the development of a general theory, our aim is to apply the form-function-framework to observations at the local scale.

Starting at the left side of the spectrum presented in Fig. 1, we focus on the observation of response dynamics and response patterns. The potential of the methods for the investigation of subsurface flow processes at the hillslope-scale and the characterization of typical runoff generation mechanisms are discussed and possible further improvements suggested. Based on these findings, the informative power and conclusiveness of the data will be discussed. To complement the functional perspective on the investigation of subsurface flow, the companion paper by Jackisch et al. (this issue) concentrates on the spatial characteristics of subsurface flow from the plot- to hillslope-scale with a specific focus on subsurface structures.

Following the form and function framework, the hypotheses focus on the potential of response observations for hillslope hydrological field research and the application of time-lapse GPR measurements in this context:

**H1** Response observations (discharge, TDR and GPR data) are sufficient to characterize subsurface flow within the hillslope. (function described without form)

**H2** Response patterns can be used to deduce flow-relevant structures in the subsurface. (function reveals form)

**H3** Time-lapse GPR measurements visualize subsurface flow dynamics and patterns and can replace hillslope trenches.

[Figure]

**Figure 1.** The concept of form and function applied to observations in hillslope hydrology. Four different categories which can be applied to data as well as the data sources.

These studies identified time shifts and amplitude variations along manually selected reflection horizons, which give evidence on vertical infiltration. Allroggen and Tronicke (2015) suggest a novel data analysis procedure for GPR data based on an attribute analysis of 2D time-lapse GPR transects. The attributes highlight specific areas of increased difference which are interpreted as local changes in soil moisture and thus related to subsurface flow paths. This method opens new perspectives for 5 the investigation of preferential flow at the hillslope-scale.

At the basin-scale, preferential flow is treated as an intrinsic property of the investigated system. Most experimental evidence stems from analyses of the dynamic water balance (McGuire and McDonnell, 2010), salt tracers (Sheng et al., 2014) and stable isotopes (Gazis and Feng, 2004; Gazis and Feng, 2014). The focus of these studies is on transit time distributions or the identification of source areas and storages throughout the catchment. Both are indicative of the spatial organization of 10 the landscape and skewed or even multi-modal transit time distributions suggest preferential flow processes. Hydrological dynamics of different observations throughout the catchment were correlated to emprically  relate catchment response to hillslope processes (Bachmair and Weiler, 2014). However, it is difficult to draw conclusions on relevant hillslope processes based on catchment-scale observations and most studies employ models to verify their hypotheses on relevant processes (e.g. Fenicia et al., 2014).

15 A particularly distinct example of a multi-modal transit time distribution is a double-peak hydrograph. These hydrographs are observed in many different catchments and have been subject to a number of research projects (e.g. Onda et al., 2001; , 2004; , 2007; , 2009; They are characterized by an almost instant response with steep rising and declining limbs, and a second prolonged response with a delay of several hours or even days after the onset of precipitation. In the literature, the first peak is often attributed to surface runoff or an immediate reaction of the riparian zone. The reasons for the second peak are often subsumed under the 20 vague term *subsurface flow processes*, which summarizes a wide variety of different processes.

The specific characteristics of subsurface flow processes are a crucial component in the hydrological behavior of a landscape. Understanding the process of the establishment of flow paths from the plot- to hillslope-scale on the one hand, and the relation between hillslope processes and catchment-scale dynamics on the other, is a major step towards understanding runoff generation mechanisms and the functional organization of catchments (Zehe et al., 2014).

25 **Objectives and experimental approach**

To fully understand the role of preferential flow  it is necessary to use a multi-scale approach. Methods are required which are capable of covering both, spatial patterns and temporal dynamics at multiple

scales. The combination of conventional hydrological methods and spatially and temporally flexible, non-invasive geophysical measurements has the potential to meet these requirements. We thus propose time-lapse GPR measurements as an alternative to established methods like trenches or dye tracer excavations.

In this study, we aimed at identifying and monitoring preferential flow processes across scales. The central element of the experimental work was a hillslope-scale irrigation experiment, which was specifically designed to observe capture both, vertical and lateral flow processes. Monitoring entailed a combination of soil moisture observations and 2D time-lapse GPR measurements. The experiment was complemented by plot-scale tracer experiments, a structural survey, a basic hydrograph analysis and stable isotope sampling.

This article is the first part of the companion paper and focuses on the temporal dynamics of preferential flow.

We compared the identified processes and dynamics with the hydrological response behavior to a natural summer storm event to evaluate their catchment-scale relevance. The potential of the methods for the investigation of subsurface flow processes at the hillslope-scale and the characterization of typical runoff generation mechanisms are discussed and possible further improvements suggested. To complement the temporal perspective on preferential flow, the second part by Jackisch et al. (this issue) concentrates on its spatial characteristics from the plot- to hillslope-scale in relation to subsurface structures.

**2 Methods**

**2.1 Study site**

[Figure]

**Figure 2.** Map of the investigated Colpach River catchment and the four gauged sub-catchments. The site of the hillslope-scale irrigation experiment is located in the Holtz 2 catchment and indicated in red.

The investigated area is located at the south eastern edge of the Ardennes Massif in western Luxembourg. It consists of a number of nested sub-catchments of the Colpach River catchment, which is part of the Attert River basin. The landscape of this area is characterized by Devonian schist bedrock (Colbach and Maquil, 2003). The soils are young and composed of eolian loess deposits and weathered schist debris. Under periglacial conditions, the weathered rocks were relocated by solifluction, causing an often horizontal or slope parallel orientation of the saprolite (Juilleret et al., 2011). The periglacial deposit layer (basal layer) is overlain by shallow top soil (upper layer). The soil is classified as Haplic Cambisol (CM, IUSS Working Group WRB (2006)).

The schist bedrock below is strongly inclined with almost vertical foliation and considered  impermeable but with fractures which can function as a complex flow network with local storage in the rock cracks when saturated (Van den Bos et al., 2006; Kavetski et al., 2011). The subsurface structures are of predominantly geogenic origin and considered temporally persistent.

Within this landscape, a typical hillslope consists of agriculturally used elevated plateaus and forested valleys with  steep slopes (15° to 25°). While the headwater catchments of the investigated area are usually narrow with marginal floodplains, the main Colpach River network is characterized by wider valleys with more pronounced floodplains.

The average annual precipitation between 2011 and 2014 was $965\,\mathrm{mm}$, the annual average air temperature was $8.8\,°\mathrm{C}$. These data stem from a meteorological station from ASTA (*administration des services techniques de l'agriculture de Luxembourg*) close to Roodt, approximately $2\,\mathrm{km}$ from the experimental site.

The experimental work conducted in the frame of this study focused on a north facing hillslope in the Holtz headwater catchment. The experiment was supplemented by hydrological data from five neighboring headwater catchments of different size. All sub-catchments as well as the location of the irrigation site are shown in Fig. 2.

**2.2 Experimental methods**

The experimental approach consists of two parts. The hillslope-centered approach concentrates on local observations at the hillslope. It includes soil moisture measurements, pore water isotope data, measurements of surface runoff etc.. These observations related to a hilslope-scale irrigation experiment and a natural summer storm event.

The stream-centered approach focuses on the discharge response and stream water stable isotope signal following two summer rainfall events. It represents the integrated response of a catchment. While hydrographs and stream tracer dynamics have been studied and discussed extensively elsewhere (Wrede et al., 2015; Martínez-Carreras et al., 2016, in the same area), we wanted to use these data to position our hillslope observation in the bigger picture of the catchment scale dynamics. An overview over approaches, methods and their foci is given in Fig. 3.

**2.3 Stream-centered approach**
* * *
The hydrological response behavior of several nested sub-catchments was investigated. At four locations v-notch or trapezoidal gauges were installed and equipped with CTD sensors (Decagon Devices Inc.). Water levels were measured every $15\,\mathrm{min}$. Precipitation was monitored with tipping buckets (Davis Instruments Corp.) in the Holtz 1 headwater. All data were logged with CR1000 data loggers (Campbell Scientific Inc.).

At the same locations and additionally close to the source of the Holtz River (Holtz 1 in Fig. 2), water samples were taken with auto samplers (ISCO 3700, Teledyne). The bottles of the auto samplers were pre-filled with styrofoam beads to avoid evaporation from the sample bottles. Samples were then transferred to glass bottles and analyzed in the laboratory at the Chair of Hydrology, University of Freiburg. The isotopic composition ($\delta^{18}O$ and $\delta^{2}H$) of the water samples was measured  by wavelength-scanned cavity ring-down spectrometry (Picarro L2120-iWS-CRDS). The results are given in the $\delta$-notation in [‰], describing the deviation of the ratio between heavy and light isotopes ($^{2}H/^{1}H$ and $^{18}O/^{16}O$) relative to the ratio of the Vienna Standard Mean Ocean Water (VSMOW). For liquid analysis the accuracy is usually given as $0.2\,‰$ for $\delta^{18}O$ and $1\,‰$ for $\delta^{2}H$ (Sprenger et al., 2015).

[Figure]

**Figure 3.** Overview over experimental methods in the stream-hillslope scheme, as well as the natural-irrigation scheme.

[revised manuscript text omitted]

5 ~~The catchment response of the investigated headwater catchments showed a strong seasonality. During winter, direct and flashy runoff reactions prevailed. In the vegetated season the pattern reversed to low base flow and relatively weak reactions to rain events. Even during this time, however, strong storm events triggered double-peak runoff responses, demonstrating the effect of a bimodal transit time distribution through the catchment. Figure 5 pictures the hydrologic response of the gauged sub-catchments 
[revised manuscript text omitted]

~~The high initial signals (i.e. localized low structural similarity to the reference measurement 24 h after irrigation start) in all transects and their decrease during the first measurements until 1.5 h after irrigation start indicated water remaining from the preceding natural rain event. While transect 1 showed only a weak signal of the natural event in the first measurement, transects 2 and 3 exhibited stronger and longer lasting signals.~~

  temporally and spatially low dynamics

The TDR measurements at the downhill monitoring area were complemented by the 2D time-lapse GPR measurements (see Fig. 4), yielding 2D images of structural similarity attributes referenced to the last measurement 24:00 h after irrigation start (Fig. 9). The first weak irrigation signals (shown in blue in Fig. 9) appeared in the first measurement after irrigation start (1:28 h), with transect 1 showing the clearest response. After about 3:23 h strong, localized signals occurred and increased in intensity over time. The general maximum was reached approximately 5:18 h after irrigation start, showing distinct activated flow paths. Most signals started to  decline after 6:45 h, which is 2:10 h after the end of the irrigation period.

In transect 2 some weak signals appeared in the depth below 2.5 m 1:30 h after irrigation start. At this time, the signal was close to noise level, but the pattern became stronger and more distinct in the following measurements. At transect 3, the persisting signal of the natural rain event made it difficult to identify the irrigation induced response. However, a weak irrigation signal appeared after 1:28 h and reached its maximum at 6:45 h after irrigation start. At transect 4 the structural similarity attribute values were generally low, which indicates a low deviation from the reference state. Either the mobile water showed  low dynamics (with regard to total mass over time), or water was less confined to specific structures and local changes are less pronounced. Both interpretations suggest that this transect was generally wetter due to its proximity to the river. Overall, the experiment does not appear to have affected this transect much.

The overview over all GPR measurements in Fig. 6B and 9 visualizes the dynamics of the hillslope section. The green natural rainfall signal faded from uphill to downhill, with the highest intensity and duration in transect 3. After irrigation start, the blue irrigation signal appeared, gradually propagating downhill and eventually overpowering the natural rain signal. 18:00 h after irrigation start (i.e. 13:25 h after irrigation ended),  no changes in GPR signal could be observed anymore. This suggests steady soil moisture conditions and thus, the absence of highly mobile water in all transects.

[Figure]

**Figure 8.** The soil moisture data measured at the TDR profiles at the irrigation site, showing the soil moisture dynamics in depth. The top four plots show all four core area profiles, columns are arranged according to the three diverging transects in downhill direction. Rows are approximately at the same contour line. Measurements were taken at $0.1\,\mathrm{m}$ increments. While data analysis was based on non-interpolated data, soil moisture measurements were here interpolated linearly for better visualization. The plots cover the time from irrigation start until 9:00 h after irrigation start to focus on the first soil moisture response. Arrows indicate the measurement times and installation depth of each TDR profile. Time is given in [h] after irrigation start.

The mobile water either left the monitored area, or dispersed by diffusion into the matrix surrounding preferential flow paths, where it remained beyond the reference measurement and thus would not have been visible by means of structural similarity attributes.

~~Besides the soil moisture dynamics, there were some spatial features revealed by the GPR measurements, which are relevant for understanding hillslope-scale flow processes. Firstly, both, the natural rain event and the irrigation caused advective flow in discrete structures more or less evenly distributed over the hillslope cross-section. An (ephemeral) groundwater body or specific flow layers could not be identified in the top 4.2 m of the subsurface. Secondly, the artificial irrigation had only a minor impact in comparison to the natural rain event, despite higher local input. The accumulated upslope areawas therefore more important than irrigation intensity or duration. The heterogeneity and disconnectivity of the preferential flow network and the methodological aspects of form are discussed separately by Jackisch et al. (this issue).~~

**3.2.4  Pore water and piezometer isotope responses**

The temporal dynamics of the stable isotope compositions of the pore water (selected depths shown as circles in Fig. 10A) partially traced the signals of the rainfall and irrigation water input (green lines and blue triangles). The high $\delta^2 H$ signal of the first minor rainfall event (dark green) was clearly visible in the top 10 cm below ground in the profile sampled at the downhill monitoring area 24:00 h before irrigation and 5:30 h after this rainfall event (Fig. 10A and C). Similarly, the isotope signal of the $2^{nd}$ rainfall event (light green) and the irrigation water (blue triangles) could be seen in the top 10 cm of the soil at the core area and the downhill monitoring area, respectively. Especially the isotope profile taken at the core area after irrigation showed an increase in $\delta^2 H$ in the top 0.85 m below ground, showing the influence of both, the irrigation and the event water. Below the depth of approximately 1.2 m of all profiles, the soil water isotope composition seemed not to be impacted by the rainfall events and irrigation.

Only few mL of water were seeping into the piezometers, with piezometer B being the only one that could be sampled more than once. Piezometer B was sampled first shortly after the irrigation ended (0:20 h) and showed a composition that was close to the irrigation water. The other two samples 1:32 h and 13:38 h after irrigation ended, showed a decrease in $\delta^2 H$, towards the composition of the soil water (red and orange diamonds in Fig. 10).

Piezometers A, C, G and H were sampled once 13:38 h after irrigation ended. The water sampled from piezometers at the core area (A and C, pink diamonds) showed the same composition as the irrigation water. Piezometers located at the downhill monitoring area (G and H, purple diamonds) in contrast, showed an isotopic composition similar to the rainfall water and different to the pore water in the depth profiles.

~~GPR and TDR are both sensitive to the electrical permittivity of the soil and hence, are expected to be well comparable. Nevertheless, a quantitative calibration of GPR data by means of TDR measurements was not possible. Due to the sensitivity of GPR measurements to installations in the soil, we kept a certain distance between the GPR transects and the access tubes and removed the TDR probes from the access tubes during the GPR measurements. Furthermore, the overlap in depth between the methods is small, as the TDR measurements extend from 0 m to maximum 1.7 m, while the GPR measurements extend~~

from approximately 0.7 m to 4.2 m. Last but not least, the GPR signal is influenced by a variety of factors and an unambiguous attribution of soil moisture values is arguable. These methodological aspects, combined with the high observed complexity of subsurface structures and flow paths make a quantitative correlation difficult.

However, a qualitative comparison of the results from both methods is possible, and shows a good agreement in space and time. The results from both methods showed similar soil moisture patterns and the dynamics suggested a continuous propagation of the irrigation signal towards greater depth (Fig. 11). In comparison to the TDR signal covering the top 1.7 m, the GPR irrigation signal was shifted slightly downward, which might have been due to inaccuracies in the GPR pulse velocity estimation, or the mere spatial distance between the two locations.

Comparison of the soil moisture response observed in the three diverging TDR transects with the closest positions along GPR transect 1 (see Fig. 9 for locations) show a correlation between signal strength in both datasets. GPR transect 1 shows a strong signal in the lateral position between 5 m to 7 m and 8 m to 10 m, leaving a gap in the area of the center TDR transect. TDR4, TDR5 and TDR6 are the downhill profiles of the center transect and show weak signals, too (see also Fig. 4 and Fig. 8), indicating that this area is generally less prone to lateral preferential flow.

**3.2.5 Response velocities**

The results of the calculated response velocities from TDR and GPR measurements are summarized in Fig. 12. The top row shows the depth distribution of observed response velocities for  all GPR transects. The bottom row shows the TDR-based results, separated in core area profiles and the three diverging transects. Additionally to the TDR-based response velocities, GPR-based velocities observed in 0.5 m wide sections of the GPR transects which were closest to the displayed TDR profiles, are shown in the plot.

At the core area, the dominating vertical response velocity was around $10^{-4}\,\mathrm{m\,s^{-1}}$, with a tendency to increasing velocities with depth  (Fig. 12). As response velocities were calculated for the entire soil profile above the measuring depth, this increase indicates a bypass of intermediate depths through preferential flow paths, and a limited and slow interaction with the matrix. The highest observed vertical velocity was $10^{-3}\,\mathrm{m\,s^{-1}}$ in the depth of 1.4 m below ground. The respective soil moisture signal was recorded in the very first measurement after irrigation start, which indicates that we might have even missed the first response.

Similar to the vertical response velocities, the dominant TDR-based response velocity at the downhill monitoring area was in the order of magnitude of $10^{-4}\,\mathrm{m\,s^{-1}}$ (Fig. 12). Response velocities of around $10^{-3}\,\mathrm{m\,s^{-1}}$ were observed in six profiles all over the downhill monitoring area, of which the highest values (1.0 to $1.6 \times 10^{-3}\,\mathrm{m\,s^{-1}}$, TDR4, TDR6, TDR11 and TDR17) are based on signals observed during the first profile measurements after irrigation start. The fastest response was observed in the top 0.5 m (TDR4, TDR6, and TDR10) of the soil and below a depth of 1 m (TDR6, TDR11, TDR17, and TDR18).

The GPR-based response velocities were calculated for the entire width of the GPR transects down to the maximum depth of approximately 4.2 m (Fig. 12 top), and for single sections close to the TDR profiles down to a depth of 1.7 m for comparison of the methods (Fig. 12 bottom). All GPR transects showed slight and localized irrigation water signals in the data collected 1:28 h after irrigation start (see Fig. 9). This translates to response velocities between $1.4 \times 10^{-3}\,\mathrm{m\,s^{-1}}$ to $2.2 \times 10^{-3}\,\mathrm{m\,s^{-1}}$,

depending on the distance to the irrigation area and signal depth. The data of the next measurements  suggest response velocities between $5.4 \times 10^{-4}\,\mathrm{m\,s^{-1}}$ to $8.2 \times 10^{-4}\,\mathrm{m\,s^{-1}}$. Given the fact that this is a conservative estimate due to the even sparser temporal resolution in comparison to the TDR measurements, response velocities are likely to be similar to or even higher than those calculated from TDR results.

**3.3 Comparison of natural event and irrigation response patterns in 2D GPR images**

The signal of the natural rainfall event could be observed throughout the entire transects. The highest signal density (i.e. areal share) was found between $0.9\,\mathrm{m}$ to $1.7\,\mathrm{m}$ depth in all transects (Fig. 13). The strongest response signal (i.e. lowest structural similarity) appeared below the depth of $2.5\,\mathrm{m}$ in transects 2 and 3 (Fig. 6A). Transect 4 showed generally higher structural similarity and thus, lower response signals.

The areal share of the irrigation signal was the highest in transect 1, with $30.5\,\%$, and decreased downhill, with $20.8\,\%$ in transect 2, $16.4\,\%$ in transect 3, and $6.0\,\%$ in transect 4 (Fig. 13). Transect 1 also shows irrigation signals in regions which were not (or not any more) active after the rain event. Especially the depth between $1.7\,\mathrm{m}$ to $2.2\,\mathrm{m}$, which showed a comparably low natural rain signal, was activated by the irrigation. Overall $39.7\,\%$ of the regions activated by irrigation in transect 1 have already been active before irrigation. In contrast, most of the irrigation signals observed in the three downhill transects consist of re-activated flow paths. Here, $72.5\,\%$, $86.1\,\%$ and $76.7\,\%$ of the irrigation patterns were activated both by the natural event as well as by the irrigation experiment (Fig. 6 and Fig. 13).

The natural rain signal in the GPR data was overpowered by the irrigation signal at 20:27 h, which is 3:23 h after irrigation start and about 22:45 h after the natural rain event. The dampened dynamics in transect 4 were due to its proximity to the river and therefore generally wetter conditions and less capacity for additional wetting.

The comparison of the two different response patterns shows that both, the natural rain event and the irrigation caused advective flow in discrete flow paths more or less evenly distributed over the hillslope cross-section. An (ephemeral) groundwater body or specific flow layers could not be identified in the top $4.2\,\mathrm{m}$ of the subsurface. Furthermore, the artificial irrigation had only a minor impact in comparison to the natural rain event, despite higher local input. Water that was supplied from upslope areas was therefore more important than irrigation intensity or duration.

[Figure]

**Figure 9.** Structural similarity attributes calculated from  time-lapse GPR data. All measurements were referenced to the last one 24:00 h after irrigation start, indicating changes in the GPR reflection patterns associated with soil moisture changes. A structural similarity attribute value of 1 indicates full similarty, lower values signify higher deviation from the reference state. Water from the preceding natural rain event (green) was identified by constant or increasing structural similarity attributes. Water from the experimental irrigation (blue) was identified by decreasing values after irrigation start by more than $0.15$. Within one column the rows give a sequence over time (after irrigation start). Columns proceed downhill, with increasing distance from the rain shield.

[Figure]

**Figure 10.** Stable isotope data from precipitation, irrigation, piezometers and pore water samples. A: Temporal dynamics of pore water and piezometer $\delta^2H$ in relation to water input by precipitation (bulk samples) and irrigation. Pore water data is shown only for the depths of piezometer filters (compare with panels B and C for depths) and the top soil (0.1 m below ground). B and C: Pore water and piezometer data over depths, separated by core area and monitoring area. Water input stable isotope data is indicated at the soil surface.

[Figure]

**Figure 11.** figure will be removed ~~Relative soil moisture changes at TDR11 (purple) and structural similarity attribute values (green and blue) in depth over time. The signal from the natural rain event preceding the experiment is shown in green, signal attributed to irrigation water in blue. The GPR data is extracted from a narrow section of transect 1, between 6.0 m to 6.1 m in direct proximity of TDR11. The location of TDR11 is also indicated in Fig. 9. Distance between TDR11 and the depicted section of GPR transect 1 was 0.35 m. Vertical lines indicate the irrigation period (red), TDR measurements (black) and GPR measurements (blue). Data was interpolated linearly in space and time.~~

[revised manuscript text omitted]

At the hillslope-scale, these preferential flow paths connected and created a diffuse network of advective flow in the voids of the young skeletal soil. The structures are related to the periglacial slope deposits, which are typical for the region and landscapes which formed under similar conditions (Juilleret et al., 2011). These temporally persistent structures cover the investigated catchment and therefore, bear the potential to connect the entire hillslope, and effectively drain it as soon as the input stops.

**Catchment-scale flow dynamics**

Double-peak hydrographs are likely the result of a multi-modal distribution of transit times through the catchment. They are caused by the spatial or functional organization of the landscape, which leads to the activation of different flow paths or different source areas and storages throughout the catchment. The reasons for the occurrence of double-peak hydrographs are therefore manifold, and depend on the specific settings in the respective catchment.

In literature, the first peak of double-peak hydrographs is often attributed to saturation-excess or infiltration-excess overland flow (Burt and Butcher, 1985), or the immediate reaction of the riparian zone (e.g. Masiyandima et al., 2003), and is usually not the main focus of investigations. The reasons described for the second peak are more diverse. Lateral flow along impermeable layers or a gradual connection of previously disconnected saturated areas (Becker (2005), cited by Graeff et al. (2009)) are processes in the shallow subsurface, which have been related to double-peak hydrographs. In other catchments, water percolating deeply through fissures in the bedrock (Onda et al., 2001), overflow of a deep subsurface storage (Zillgens et al., 2007), or rapid displacement of groundwater and pressure transduction (Wenninger et al., 2004), cause the prolonged second peak.

**4.1.2   Natural rainfall event observations**

The subsurface response patterns revealed by the GPR measurements after the natural rainfall events were similar to the irrigation induced patterns with regard to their patchiness, but showed a slightly different spatial distribution (Fig. 13). The GPR measurements 12:52 h after the $2^{nd}$ rain event showed the highest density of response signals in the depth between $0.8\,\text{m}$

to 1.7 m (Fig. 6A and 13). The higher response in the shallow depth could be interpreted as the signal of vertically infiltrating rain water, which did not occur during the irrigation. This depth also correlated with high stone content and the periglacial cover beds, which are characteristic for the area (e.g Juilleret et al., 2011). While no (transient) water table could be detected in the monitored depth, the patterns indicated a concentration of preferential flow paths in this depth.

The TDR measurements also showed high initial soil moisture and a strong reaction to irrigation in this depth. However, except for the slight decrease in soil water content in the most downhill located TDR profiles TDR6, TDR13 and TDR14 (Fig. 8), soil moisture values barely fell below the initial values measured between rainfall events and irrigation. This means that the signal of the natural rainfall events was already gone in the shallow subsurface, and that no information on the natural rain signal in this depth can be derived from the TDR data.

The timing of the response dynamics observed with the GPR measurements and the discharge response also shed light on the prevalent processes. The natural rainfall events ended at 21:35 h on June 20, 2013. The first GPR measurements were taken at 10:42h on June 21, 12:52 h later (Fig. 5). Located at the lower section of the hillslope, they were interpreted to show a declining soil moisture signal, which was mostly gone 37:22 h after the $2^{nd}$ rainfall event (i.e. 18:00 h after irrigation start, see Fig. 6B and 9). Following the hypothesis of a top to bottom drainage of the hillslope, and considering the downslope location of the study site, the recorded signal represented the tailing of the shallow subsurface flow response to the natural rainfall event.

At the time of the first GPR measurements, the first peak of the hydrograph was already gone, while the second peak was on its rising limb and reached its maximum 12:00 h later (24:52 h after rainfall event, Fig. 5). Thus, the following decline in subsurface response was observed after the first peak and coincided with the rise of the second peak. This timing provides strong evidence that the second hydrograph peak was not primarily caused by the activation of the observed preferential flow paths in the shallow subsurface.

Several studies investigated the double-peak hydrographs of the Weierbach catchment.  Wrede et al. (2015) used dissolved silica and electrical conductivity and found that the first peak was dominated by event water, while the second peak mainly consisted of pre-event water and strongly depended on antecedent conditions. Based on these observations, the first peak was attributed to fast overland flow from near-stream areas, while the second peak was attributed to subsurface flow where antecedent water was mobilized. Fenicia et al. (2014) came to a similar conclusion and identified a riparian zone reservoir as the origin of the first peak.

The stable isotope data collected during the rainfall event prior to the irrigation experiment, showed the same dynamics as observed by Wrede et al. (2015) (Fig. 5). The isotopic composition of the first peak suggested a mixture of event water and pre-event water, while the composition of the second peak indicated the dominance of pre-event water. A simple water balance revealed, that the total mass of the first peak of the event runoff accounted for about 4 % of the precipitation amount in the Holtz 2 catchment. Based on a rough delineation, the existing wetland patches in the catchment amounted to approximately 800 m$^2$ in the source area. Thus, the specific discharge of the first peak exceeded the existing wetland patches or riparian zones of the headwater catchment by a factor of 27, and suggests that overland flow from near-stream saturated areas could not solely explain the observed discharge response.

However, in the Holtz 2 catchment, the first peak of the summer storm runoff accounted for about 4 % of the bulk precipitation amount, based on a simple water balance. This portion exceeded the existing wetland patches or riparian zones of the headwater catchment by a factor of 50, and suggests that other processes had to be involved.

Here, the timing of the GPR dynamics and the discharge response shed light on the prevailing processes. The natural rain event ended at 22:00h on June 20th, 2013. The first GPR measurements were taken at 10:00h on June 21st, approximately 12 h later (Fig. 5). Located at the lower section of the hillslope, they were interpreted to show a declining soil moisture signal, which completely disappeared 24 h after the rainfall event. Following the hypothesis of a top to bottom drainage of the hillslope, and considering the downslope location of the study site, the recorded signal represented the tailing of the shallow subsurface flow response to the natural storm event. At this time, the first peak of the hydrograph was already gone, while the second peak was on its rising limb and reached its maximum 12 h later (36 h after storm event, Fig. 5). This timing provides strong evidence that the second peak was not caused by the activation of the observed preferential flow paths.

The strong signal in the shallow subsurface (0.7 m to 1.7 m below ground), caused by the natural storm event, and the higher response despite lower intensity and duration (i.e. accumulated local input per m$^2$ at the hillslope *versus* irrigation area) in comparison to the irrigation experiment (Fig. 6), showed that the strong signal from the natural rain event was caused by water from the entire hillslope draining through the shallow subsurface structures. The accumulated uphill area had therefore more impact on subsurface flow than the accumulated local input.

This finding in combination with the  high mobility of the water revealed by the irrigation experiment, suggests that the activation of preferential flow paths within subsurface structures was likely responsible for the first immediate peak of the hydrograph, and demonstrates the importance of lateral subsurface flow at the hillslope-scale.

Connectivity established quickly within preferential flow paths, and high response velocities bore the potential to route water from the hillslopes towards the river within very short time. Similarly, Van den Bos et al. (2006) stated, that the presence of preferential flow paths and steep slopes in the Colpach River catchment enable subsurface runoff, even at times when the soil and weathered zone are not yet at field capacity.

**4.1.3 Synthesis: Functioning of the investigated hillslope**

Comparison of the GPR data of the natural rainfall event and the irrigation reveals that the response in the shallow subsurface was stronger after the natural event, even though the input per square meter was much lower than during the irrigation (Fig. 6 and 13). The observed response could only be caused by the accumulated water input of the (entire) hillslope draining through the shallow subsurface. Thus, the hillslope is prone to a substantial amount of lateral flow, which quickly ceases after water supply stops.

In combination with this finding, the high potential response velocities (Fig. 12) and the high mobile water fraction revealed by the TDR and GPR measurements show, that fast, lateral subsurface flow is an important process in the investigated hillslope. The timing of the hydrograph dynamics and the declining response in the GPR measurements described above (Fig. 5), as well as the freely percolating rainfall event and irrigation water shown by the stable isotope data, give further evidence that the

activation of preferential flow paths within the shallow subsurface was contributing to the first immediate peak of the stream hydrograph.

Preferential flow paths established quickly, and high response velocities have the potential to route water from the hillslopes towards the river within few hours. The presence of preferential flow paths and the steep slopes in the Colpach River catchment were reported to enable subsurface runoff, even at times when the soil and weathered zone are not yet at field capacity (Van den Bos et al., 2006).

Many catchments reportedly showing double-peak hydrographs are headwater catchments with predominantly steep slopes and shallow soils, and in many cases with periglacial slope deposits (Burt and Butcher, 1985; Onda et al., 2001; Graeff et al., 2009; Birkinshaw and Webb, 2010; Fenicia et al., 2014; Wrede et al., 2015; Martínez-Carreras et al., 2016). Such systems are characterized by pronounced subsurface structures and therefore, prone to heterogeneous flow patterns and preferential flow at the plot- and hillslope-scale.

~~The gradual activation of lateral flow paths and the resulting establishment of hillslope connectivity has also been found in other landscapes. At the Panola research catchment, bedrock topography leads to a fill-and-spill behavior (Tromp-Van Meerveld and McDor In a subarctic soil filled valley (Spence and Woo, 2003), flow generation depends on the spatially variable soil storage capacity as a result of topography and the spatial heterogeneity of soil characteristics and ground frost. Flow paths within the hillslope establish and eventually connect large parts of the hillslope, causing a sudden and strong increase in subsurface flow.~~

The processes causing the second peak could not be resolved with  this study, but it is hypothesized that deep percolating water from hillslopes and plateaus caused the delayed response. This hypothesis is backed by a study comparing catchments of different geology (Onda et al., 2001), where the prolonged response of double-peak hydrographs was identified as an indicator for deeply percolating subsurface flow through bedrock fissures. This theory might also apply to the here investigated catchment with its fractured schist bedrock (Kavetski et al., 2011). Furthermore, deep subsurface storage overflow (Zillgens et al., 2007) and fast groundwater displacement (Graeff et al., 2009) are processes, which  may play a role in the behavior of the Colpach River catchment. These hypotheses are also backed by the isotopic composition of the second peak, suggesting the dominance of pre-event water (Fig. 5), and could explain the dependency of the occurrence of double-peak hydrographs on groundwater storage as described by Wrede et al. (2015) and Martínez-Carreras et al. (2016). They could only observe the second peak if the groundwater storage was sufficiently filled, resulting in a hysteretic threshold behavior for the occurrence of the second, delayed peak.
* * *
**4.2 Form and function in hillslope hydrology**

Similar to the categorization of methods and observation data (see Fig. 1 and 3), we also subsumed the results and findings under the form-function-framework. By doing so it becomes clear, that observations and results are not always linearly obtained. Function is not necessarily described best by mere process observations. We thus want to discuss our data with regard to their value for the findings of the different form and function categories.

As elaborated in Sec. 4.1.3, the response observations described the functioning of the investigated hillslope well. Response dynamics and their relation to each other across scales were a valuable source of information, shedding light on the characteristics of subsurface flow processes. Response patterns and spatially distributed point observations helped to develop a conceptual idea of the spatial organization and the functional network of the system. They were necessary to provide the spatial context to calculate response velocities and develop an idea of the establishment of flow paths. As such, response observations were the major key stone towards understanding the investigated system, supporting hypothesis H1: the function of a system can be described by response observations.

In addition to pure response observations, however, basic knowledge about structures and local characteristics strongly improved the interpretability of our data and reduced the ambiguity. This basic knowledge included the presence of periglacial slope deposits, the downslope position at the hillslope, and the hydraulic conductivity of the subsurface. This information was easily obtained from literature and in the field. It allowed us to close gaps in observational scales and link local observations at the hillslope to the overall system responses, and was thus the basis to more reliably relate the hillslope-centered and stream-centered observations.

Without this structural knowledge, the informative scope of response observations is limited to their observation scale. Transfer to other scales remains speculative. This fact is an important aspect, as most *in situ* response observations suffer from limitations in spatial resolution. Soil moisture measurements are restricted by their integration volume, and point measurements in general struggle to cover the entire domain. Here, more detailed information on (flow-relevant) structures might greatly improve our process understanding.

While we were able to describe processes in the investigated hillslope in great detail, new findings on structure are scarce. Any detail on the form of the investigated hillslope we concluded from our observations, were mere confirmations of previous knowledge. The only knowledge on form that was gained from the presented data, concerns the flow-relevance of structures. We found that substantial lateral flow was occurring in an intermediate depth, most likely associated with the periglacial slope deposits. We also found, that distinct preferential flow paths occurred in greater depth, which suggests that the bedrock might not be as impermeable as previously assumed (Fenicia et al., 2014).

A conclusive picture of subsurface structures could not be drawn from process observations, partially refuting hypothesis H2, stating that function helps to reveal form. However, the observed response patterns were helpful to characterize known structures with regard to their flow-relevance.

**4.3    2D time-lapse GPR measurements as link between form and function**

The interpretation the time-lapse 2D GPR measurements is difficult and requires a thorough understanding of the expressiveness of the recorded data. While changes in structural similarity at a short time scale can be attributed to variations in soil moisture content, these changes contain no information on where the soil moisture content increases or decreases (i.e. the direction of the change). The data are qualitative information only, and supportive measurements are necessary. Therefore, GPR measurements need to be accompanied by other methods, such as TDR measurements or trenches, to provide a reference for the observed changes.

In the correct setup, however, 2D time-lapse GPR measurements are a very powerful tool and a valuable source of information. Especially in highly heterogeneous hillslopes, the spatial context provided by the GPR measurements is crucial for the investigation of complex preferential flow networks. This spatial context can not be provided by ERT measurements with their comparably low spatial resolution, highly invasive excavated trenches or by point measurements. In contrast to dye tracer excavations, time-lapse GPR measurements yield the temporal component, which is necessary to cover the highly dynamic processes. They allow for repetitions in time and space and avoid the manipulating effect of a trench face on flow through the unsaturated zone (Atkinson, 1978).

Within the form and function framework, the biggest advantage of 2D time-lapse GPR measurements lies in the combination of high resolution spatial response patterns and dynamics. The method provides a direct link between flow-relevant structures and processes. It allows us to map response patterns in high temporal resolution, without manipulating the subsurface flow field. As such, this method has the potential to visualize the gradual establishment of flow paths, localize them and calculate response velocities (hypothesis H3).

**General experimental setup and monitoring**

Irrigation intensity and duration of the experiment were not chosen to mimic natural conditions, but to activate all potential flow paths. The limited input area and the artificial irrigation conditions both might have influenced flow patterns and processes. As such, the experimental settings might not fully allow for a direct inference of processes triggered by a natural storm event. However, the strong subsurface flow reaction to the natural storm event was generated by the accumulated input of the entire uphill area. As such, the high rates were reasonable choices to approximate the hydrological response in the shallow subsurface and could even have been stronger.

Furthermore, the sharp boundary purposely created by the rain shield, was also the advantage of this experimental design. It allowed for a separate investigation of lateral and vertical processes. Due to the separation of irrigation area and downhill monitoring area, lateral processes were easier to observe and interpret.

**Advective flow, timing and dynamics**

The temporal proximity to the natural storm event preceding the experiment, helped to evaluate the experimental observations with regard to their transferability and their importance at the larger scale. At the same time, its impact had to be considered for data interpretation. Investigating dynamics of a system, the definition of the reference state is crucial. Natural systems are characterized by ever changing conditions and barely establish steady state in the strict sense. Even if no soil moisture changes can be observed, a uniform and continuous movement of water can not be excluded.

However, as we were interested in fast preferential flow, a sufficiently steady state was established when all advective momentum was dispersed. To ideally visualize the soil moisture changes caused by the advective flow, the measurements showing the least dynamics in soil moisture were chosen as the reference.

Most TDR profiles returned to approximately initial conditions after 18:00 h18 h with only a marginal remaining soil moisture increase. Accordingly, the GPR signal stabilized approximately 18:00 h18 h after irrigation start, showing that the fast dynamics caused by the natural rain event and the irrigation vanished. The monitoring period was thus sufficient to cover the fast response to the irrigation for both methods. The fast response to the natural rain event, in contrast, was not visible in

the TDR measurements prior to the irrigation period. This discrepancy might be related to the differences in spatial coverage with regard to depth and the position along the topographic gradient. Most TDR profiles with an installation depth of more than 1 m are located uphill of GPR transect 1, where only a very weak signal of the natural rain was recorded.

**Monitoring methods**

The installed piezometers did not show a detectable response. Neither the piezometers, nor the GPR measurements indicated a connected water table or ephemeral groundwater body at depths of less than 4.2 m below ground. The highly conductive preferential flow network drained the soil very effectively, even during the high input of the experimental irrigation. Saturated conditions are very improbable in such soils. Flow through adjacent structures caused some water to seep into the piezometers, but did not indicate a water table. Under such predominantly unsaturated conditions, the piezometer response is very sensitive to their specific location. This observation shows that piezometers are not well suited for the observation of hydrological processes in highly structured and well drained soils.

GPR measurements are influenced by a variety of factors, such as repositioning accuracy, antenna coupling and inaccuracies in the data recording and processing. Therefore, special care was taken about exact positioning of the GPR antennas, by using an automated tracking station and wooden guides for exact positioning. While soil properties other than soil moisture were assumed stable over the course of the experiment, methodological factors, such as surface contact of the antennas and signal deviation below areas of large differences in radar velocity are presumably the only additional source of artifacts in the recorded radargrams.

While we deem the GPR signals to be reliable indicators for soil moisture changes, care has to be taken not to over-interpret the resulting data. Firstly, the structural similarity attribute is not linearly dependent on soil moisture changes. This fact, in combination with the depth attribution based on a constant GPR wave velocity, suggests that GPR images are to be interpreted as a qualitative map of changes rather than a tool to precisely locate and quantify preferential flow paths. Secondly, structural similarity attributes highlight relative changes, without specifying the direction of the change. Our interpretation that decreasing similarity indicates increasing soil moisture, is based on the assumption that the reference measurement was the driest.

However, we observed a fast and consistent evolution of the GPR patterns over time. Furthermore, the good temporal and spatial correspondence with TDR measurements also provides strong evidence that the observed signal is actually related to soil moisture changes. Taking all the above mentioned aspects into account, the GPR data provides valuable insight into subsurface flow processes, which is hard to obtain with other methods.

While soil moisture is a commonly used indicator for flow processes in the unsaturated zone, its measurement is difficult. Our challenge was to record full profiles as continuously as possible. The applied technique allowed for that and yielded a decent data set. However, besides the positioning of the probes and the limited temporal resolution implied by the measuring routine, the concept of soil moisture measurement itself exerts some limitations. The penetration depth of the TDR signal is subject to change with soil moisture. As such the absolute values may be largely ambiguous in terms of precise determination of the amount of water. Furthermore, the scale discrepancy between integration volume and flow path volume constrains the sensitivity to high velocities in small-scale flow paths, which will be discussed in Sec. **??**.

**Combination of monitoring methods**

Soil moisture measurements at the core area disclosed a gradual wetting of the top $0.4\,\mathrm{m}$ of the soil. This could mistakenly be interpreted as a continuous wetting front and described as vertical matrix flow based on 1D assumptions. The information on spatial patterns and flow processes obtained from measurements in greater depths and GPR transects, however, revealed that this simplification is not adequate to properly represent all processes.

5 Furthermore, a separate interpretation of single TDR profiles or even each of the three TDR transects (Fig. 4) on its own would have led to different conclusions about subsurface flow processes. The characteristics and extent of flow patterns and dynamics could only be inferred from the combination of a dense soil moisture monitoring network and the time-lapse GPR transects. A limited number of point observations would have been very likely to either miss hydrologically active structures or to overestimate their impact. The two dimensional monitoring put single measurements into spatial context and revealed the

10 process scale and adequate observation density needed for a realistic representation of the investigated processes.

In contrast to conventional trenches or point measurements, the presented combination of TDR profiles and 2D time-lapse GPR measurements provides the spatial context that is crucial for the investigation of complex preferential flow networks. In contrast to dye tracer excavations, time-lapse GPR measurements yield the temporal component, which is necessary to cover the highly dynamic processes. This combination of methods allows for repetitions in time and space and avoids the

15 manipulating effect of a trench face on unsaturated flow paths (Atkinson, 1978). In recent years, the combination of TDR and electric resistivity tomography (ERT) was successfully applied to monitor soil moisture dynamics at the hillslope-scale (Wenninger et al., 2008; Wenninger et al., 2013; Wenninger et al., 2015). Long measurement times and a limitation in electrode spacing, however, lead to a comparably low spatial and temporal resolution. Under the given circumstances and research questions, GPR measurements are less restricted in resolution and are therefore better suited for the observation of fast and

20 small-scale preferential flow.

**Integration volume and resolution and scale**

With a measurement principle based on the dielectric permittivity, TDR and GPR both take snapshots in time and describe the current state of the subsurface. The momentum of the water is not measured directly, and water fluxes are inferred indirectly from changes in soil moisture and reflection patterns. Response velocities can only be derived from the spatial and temporal

25 dynamics of soil moisture, and the separation of diffusive and advective movement is difficult.

While a vertically proceeding wetting front is relatively easy to observe and quantify by means of a 1D soil moisture profile, the identification and investigation of preferential flow paths is more difficult. Preferential flow paths are spatially and temporally discrete and in the case of the investigated hillslope expected to be smaller than the integration volume of both GPR and TDR. Thus, preferential flow paths will only become visible if their volume is comparably large, or if they interact with the

30 surrounding matrix and increase the overall soil moisture. Diffusive flow has better visibility and its importance is thus likely to be overestimated. Fraction and velocity of advective flow are likely underestimated due to their low or delayed soil moisture footprint.

The dynamics of preferential flow is are often characterized by a stepwise activation of single flow paths. Temporal interpolation between measurements is therefore not appropriate. The measurement frequency directly impairs the resolution

35 and detectable maximum velocity. The $2\,\mathrm{vol\%}$ threshold applied to identify the first response in soil moisture during this

experiment is rather high. We cannot exclude that the actual first response happens earlier without being detected. Furthermore, several locations already showed a soil moisture response in the very first measurement after irrigation start. Figure 12 shows that the setup of the GPR measurements was not capable of capturing response velocities faster than $2.2 \times 10^{-3}\,\mathrm{m\,s^{-1}}$. The calculated velocities from both, GPR and TDR measurements, are therefore rough estimates and likely to underestimate preferential flow velocities. To reliably quantify maximum response velocities, a higher measurement frequency would have been necessary.

The size of the integration volume is a trade-off between representativity and sensitivity, and the ideal integration volume strongly depends on the research objective. While a small integration volume is able to instantly detect small preferential flow paths as they establish, it requires a higher number of spatial repetitions to be representative for the investigated system. 2D transects, as provided by the time-lapse GPR measurements, are a good improvement over point observations and 1D profiles. Nevertheless, the resulting patterns can only be properly interpreted if the characteristics and scale of preferential flow paths are known.

**Multi-scale approach**

The experimental investigation of the runoff generation, and especially the origin of double-peak hydrographs, usually focuses on integrated signals such as environmental tracers (Zillgens et al., 2007; Zillgens et al., 2015) and water balance calculations. While many studies use models to prove or reject different hypothesis (Graeff et al., 2009; Graeff et al., 2014), the direct observations of subsurface flow processes in relation with double-peak hydrographs are sparse. Piezometers (Masiyandima et al., 200 are restricted to processes in the saturated zone. Distributed soil moisture measurements or temperature profiles (Birkinshaw and Webb, 201 are based on the assumption of continuous processes and often neglect lateral flow processes. The combined investigation of hillslope-scale processes and catchment-scale dynamics allows for conclusions on the relation between subsurface flow and runoff behavior.

Jackisch et al. (this issue) complement the hillslope-scale irrigation experiment with a structural survey as well as plot-scale samples and dye tracer experiments, to improve our understanding of small-scale processes and their dependency on subsurface structures. These two studies combine to a multi-scale perspective on preferential flow processes and the controlling structures.

**5 Conclusions**

The study site is an example for headwater catchments with steep slopes and young and highly structured soils,  typical for landscapes that formed under  periglacial conditions. Here, preferential flow paths quickly developed in the unsaturated zone within minutes after the onset of an intense  rain event, causing lateral flow across the hillslope.  In combination with the high response velocities of up to $10^{-3}\,\mathrm{m\,s^{-1}}$ or faster, and the large  fraction of mobile water, these flow paths have the potential to quickly route  water from the hillslopes towards the stream.

The strong dynamics and high spatial variability of preferential flow challenge the investigation of these processes. While we were able to describe the overall flow dynamics, the spatio-temporal resolution of our monitoring setup was not sufficient to reliably quantify the maximum response velocities. Our study has furthermore shown that  causes and importance of observations can only be evaluated if the necessary context is known. This context includes knowledge of the spatio-temporal patterns on the one hand, and relevant process scales on the other.

The spatio-temporal context is provided by a combination of quantitative point measurements (TDR), qualitative mapping of patterns and dynamics (2D time-lapse GPR) and the observation of integrated system response (hydrographs and  stable isotopes). Either of these approaches provides a substantial piece to the puzzle, while neither of them on its own would have provided the full picture. The experiment has shown that time-lapse GPR measurements are a powerful tool which provides new perspectives for the investigation of preferential flow processes in hillslopes. The methodology's flexibility and minimally invasive character allow for repetitions in time and space, and thus, the direct observation of processes under driven conditions. Depending on the research question, the method can replace labor-intense trenches and increase the observation density.

The observation of response patterns and dynamics by means of the TDR and GPR measurements were shown to suffice to characterize subsurface flow within the hillslope. Processes were identified and characterized without any concrete information about spatial structures. However, despite the high number of TDR observations and the 2D response patterns obtained from the time-lapse GPR measurements, our observations were methodologically limited in spatial and temporal resolution. Measuring intervals and integration volumes of the methods are restricted, and interpretations beyond observation scale remain speculative. Conclusions on or links to larger or smaller scales are not reliable. Here, more detailed information on spatial structures and their impact on flow processes might improve our understanding of the investigated area and will improve the ability to transfer findings to other scales by providing the physical basis behind the observed processes.

The observed response patterns, revealed by the GPR and TDR measurements, allowed to develop a conceptual description of the flow path network, which is linked to subsurface structures. Certain spatial characteristics of the flow path network such as layers prone to preferential flow could be inferred from the response patterns. However, actual structural features, such as the delineation of the deposit layer or the bedrock interface, could not be localized. All structure related conclusions are rather confirmations of previous knowledge than new findings. The topic of structural exploration will be taken up in the companion

paper by Jackisch et al. (this issue) to further elaborate and discuss the methodological aspects linking form and function in hillslope hydrology.

The combination of catchment-scale dynamics and hillslope-scale processes allows us to infer the overall functioning of the catchment. However, while these processes and dynamics could well be observed with the presented approach, there is a lack of process understanding at the smaller scale. Even though the experiment gave us a good idea of the general dynamics and patterns of the relevant processes, the applied resolution was not sufficient to reliably determine maximum velocities and describe discrete spatial properties of the preferential flow paths. Especially the link between processes and structures is still missing, as the relevant process scale is below the resolution of the applied methods.

The second part of this study presented by Jackisch et al. (this issue) focuses on this missing link by complementing the hillslope-scale irrigation experiment with plot-scale dye tracer experiments and structural exploration. The synthesis of the results from both parts provides detailed insights into hillslope processes and preferential flow.

*Acknowledgements.* We are grateful to Marcel Delock, Lisei Köhn and Marvin Reich for their support during fieldwork, as well as Markus Morgner and Jean Francois Iffly for technical support, Britta Kattenstroth for hydrometeorological data acquisition and isotope sampling and Barbara Herbstritt and Begoña Lorente Sistiaga for laboratory work. Laurent Pfister and Jean-Francois Iffly from the Luxembourg Institute of Science and Technology (LIST) are acknowledged for organizing the permissions for the experiments and providing discharge data for

5   Weierbach 1 and Colpach. We also want to thank the three anonymous reviewers, whose thorough remarks greatly helped to improve the manuscript. This study is part of the DFG funded CAOS project *From Catchments as Organised Systems to Models based on Dynamic Functional Units* (FOR 1598).

---

## Author Response (AR2)

We once again are grateful for the thorough reviews and comments on our manuscript. As elaborated below, we did not fully agree in all points, but happily implemented most of the recommendations made by the two reviewers. The suggestions made here and in the first review helped to improve the manuscript and we truly appreciate the effort put into it. We addressed all remarks by the two reviewers and inserted our answers to all comments into the original text in purple for reviewer#2 and orange for reviewer#3. All major changes to the manuscript based on suggestions by the two reviewers are also indicated in the respective color. Any changes made based on the editor's concerns, typo fixes or any other updates to the text are color coded in blue. Furthermore, Figure 3 and 10 were updated, based on the reviewers' comments, which is not apparent in the manuscript. For details, please see the answers below.

As elaborated in detail in the answers to the reviewer's comments below, we refrained from merging the two manuscripts into one. Thus, the main focus of the second revision of the presented manuscript include strengthening of the link between the two papers by including more references to the companion paper, as was asked by the editor. At the same time, we tried to mitigate the concerns raised by reviewer#3, regarding the lack of structural information, and included some more information on subsurface structures with reference to the companion paper.

We furthermore increased the focus on GPR as a methodological advancement in hillslope hydrology. This was already done in the last revision, and was now further emphasized by including a short review of previous GPR application for monitoring subsurface flow processes. Last but not least, we included a short elaboration on the basis on which we compared the results from different observations (natural conditions vs. artificial irrigation). We hope that these revisions suffice to mitigate the remaining concerns of the two reviewers.

**Answers to the general comments by reviewer#2**

The authors have put a lot of effort into the revision of the manuscript and made substantial changes. I like the common theme of 'form and function' and think it contributes substantially to the conceptualization of hydrological processes. It also gives the manuscript more structure. Overall, the manuscript reads better, and especially the discussion has improved, also by returning to the function and form concept. The figures are still very well done and contain a lot of information (sometimes almost too much).

I understand that the focus of the paper is now more to make a case for using GPR for subsurface investigation. For that the introduction should elaborate more on usage of geophysical methods and specifically GPR in hydrological process studies, advantages and challenges. There is a short paragraph on this (P. 3, L 25-32) with a mention of a few studies that have used GPR but no more information on these studies: what did they find? What were the shortcomings?

How is this study trying to overcome shortcomings or close gaps? How does it go beyond the cited studies?

As suggested, we provide more information on the previous use of GPR for soil moisture monitoring. A new paragraph in the introduction presents a short review of applications and reasons for us to use the novel analysis procedure presented by Allroggen & Tronicke, 2015 (p. 2, l. 29 ff.).

Overall, I am still not struck by a major scientific outcome of this study or a major progress in understanding the hydrological response of these small catchments better. The link between natural rainfall data and irrigation data with respect to elucidating hydrological functioning remains weak. The irrigation setup (intensity) is not comparable to natural conditions, and the natural rainfall events are too few to really say something about the functioning of the hillslopes and catchments under various conditions. My recommendation would be to put more emphasis then on the GPR part and show how this study advanced the usage of GPR in hydrological process studies.

We agree with the referee that the GPR application in this study is novel and needs to be emphasized. We already attempted to add this emphasis in the last round of revisions especially in the hypotheses, discussion and conclusions. We will now further emphasize this aspect by providing a short review of the use of GPR for soil moisture monitoring as described above.

We furthermore included a short section comparing the water input of our irrigation with the water input of a natural rainfall event on the entire hillslope (p. 8, l. 15 ff.). While this comparison does not justify the irrigation experiment to mimic natural conditions, it nevertheless provides a basis for interpreting and comparing the soil moisture responses made under natural and artificial conditions.

**Answers to the specific comments comments**

- Abstract, L 9: explanatory

  This was changed as suggested.

- Intro: Again, in the introduction and throughout the manuscript, the terms 'subsurface flow' and 'preferential flow' are still used interchangeably which I find problematic. It would be good here to state early that the authors' hypothesis is that subsurface flow at their sites happens through preferential flow features. Maybe there is evidence from other studies from the sites that the authors could cite? This has become a very well studied catchment.

  In the revised introduction we clearly state, that preferential flow is a special case of subsurface flow, which is related to heterogeneous subsurface structures (p. 2, l. 3 ff.).

- Intro: I would recommend to merge the two sections of the intro into one coherent text and also to move the paragraphs on form and function up, before the description of experimental approach.

  We merged the two sections as suggested and added two paragraphs to give a short review on GPR for hydrological monitoring and link this part of the introduction to our methodological approach. As this provides more elegant transition to the following paragraph in comparison to the

former version of the manuscript, we abstained from moving the form and function paragraph to a different position.

- P. 8, L 12: Please do not write 'etc' but instead, list all the measurements/observations types you collected.
  This was changed as suggested.

- P. 8, L 21: Please state somewhere clearly when (date) the experiments were done, when the analyzed natural rainfall event occurred, when the irrigation experiment was carried out, that would make it easier to understand the sequence of experiments.
  This information is given on page 7, line 20 as well as in Fig. 5, which depicts the timing of the rainfall event, the irrigation and different observations. However, we also inserted this information at the beginning of the methods section, to emphasize the temporal relation between the experiment and other observations.

- P. 8, L 22: what is a ctd sensor? better write pressure transducer and specify the sensor type in parentheses
  This was changed as suggested.

- I would revise the caption of Fig. 3; e.g. 'Experimental methods applied in the stream-centered and hillslope-centered approaches, divided into the sampling during the natural rain event and the irrigation experiment.'; also: What does background mean?
  The caption was changed as suggested. We also added a short explanation of the category 'background' to clarify the figure.

- P. 9, L 1: missing word 'bulk samples of the'?
  This was changed as suggested.

- P. 9, L 3: piezometer sampling in the stream-centered approach is not listed in Fig 3
  The information was added to the schematic

- After reading the description of experiments, I would recommend a separation into rain event and irrigation, rather than the separation into stream- and hillslope-centered. That might be easier to follow.
  In the original version of the manuscript, the separation was made only between rain event and irrigation, as suggested. However, it didn't become clear then, that there were also hillslope-centered observations made during the natural rainfall event, which led to confusion elsewhere. To pronounce the hillslope-centered GPR-measurements during the natural storm event, we introduced the concept of hillslope- vs. stream-centered approach and organized our methods accordingly (see Fig. 3). In order to avoid the previous confusion, we prefer to stick to the new categorization, which is more flexible and better suited to present our methods.

- P. 10, L 4-5: 'plot had a slope..' and '..at the hillslope is..'
  This was changed as suggested.

- P. 19, L 4: subsequent instead of consequent
  This was changed as suggested.

- P. 19, L 12: '... was ... vanishing'
  This was changed as suggested.

- P. 33, L 7: '... preferential flow ...'
  This was changed as suggested.

**Answers to the general comments by reviewer#3**

Generally, the manuscript is in much better shape now. The authors really put a lot of effort into improving the manuscript. They did fairly address my critique on the lack of a story line, the small sample size in hydrograph separation and the justification of linking the observed processes in the irrigation experiment to the natural conditions and all other detailed comments. I think this analysis deserves to be published, however, I remain conflicted about the separation of the data into two companion paper. There are three main reasons for my concern:

1. The authors made it clear that they use the form and function framework to present the different kinds of data in different manuscripts, i.e. the response dynamics and patterns are used to deduce the flow path in the first companion paper and the flow path is used to explain the structure in the second paper. As a reader, I feel this separation is an unnecessary artificial construct that makes the whole story lengthy. Why not include the data on structure right here in this manuscript? As a reader, I expect to back up the findings made here (flow path was deduced from response dynamics and patterns) with the data collected about the structure as presented in Jackisch et al.
   The form and function framework was introduced in the first place in order to formulate and test different approaches to understanding the investigated system. Thus, the separation of the data according to this framework and the two companion papers is indeed an artificial construct, as we do have an enormous amount of data on both sides. The methods, data, and discussion of both approaches however, exceed the frame of one paper and we consider it worthwhile to discuss the two different approaches. To complete the information needed for the process interpretation in the presented paper, we included some more details on subsurface structures in the methods section as well as in the discussion, linking back to the companion paper.

2. I think that a paper that appears in a series of companion paper should still be stand-alone which I feel is not the case with both these manuscripts.
   In the revised version of the manuscripts we included full descriptions of the experimental setup and all methods needed, rather than referring to descriptions in the companion paper in case of an overlap. Both papers have their own, separately valid findings. We certainly agree, that the lecture of both manuscripts provides more insight and conclusions as described above. Both papers however, have their own story lines and findings, and provide - in the revised version - all necessary information to elaborate these findings. The newly added information on subsurface structures mentioned above will further support this approach.

3. looking at the second manuscript by Jackisch et al., I find a lot of overlap with the Angermann et al. manuscript. I suggest to put the key findings of the second companion paper into this manuscript by Angermann et al. I feel that merging these two manuscripts will make a more balanced story of exploring subsurface flow in the context of the form and function framework and will improve the manuscript.

   We originally wrote only one manuscript containing both parts, but found that the multitude of methods and results clearly exceeded the frame of one single paper. We therefore decided to split it in two parts. Both companion papers still present a large amount of different methods, and combining them would require a strong reduction on both sides. However, the plot-scale experiments in the form-part are a crucial part to explain and interpret the observations made during the irrigation experiment. The isotope and hydorgraph observations are required to fully understand flow dynamics and relate the artificial observations to natural conditions. We feel that the required reduction would not benefit the findings and the story, and decided to stick with the current separation.

   In the course of the last revision, we actually extended the description and discussion of the hillslope-scale irrigation experiment, especially in the form part. This was done in response to critique by some of the reviewers, who demanded both manuscripts to be more stand-alone. This increased the methodological overlap between the two companion papers, but improved the story lines of the single manuscript.

**Answers to the specific comments**

There continue to be some minor issues. I am being picky but I offer these comments to help improve readability.

- Page 13, lines 27-28: if you could not sample in high frequency but still calculated a weighted average, how many samples did you take? Is it a within-event average or an intra-event average?

  As described in the methods section, we collected bulk samples during the rain events. While we could not display the temporal dynamics of the stable isotope composition, bulk sampling of one event at least provides a weighted average and are therefore superior to single samples in time. The formulation was adapted to make this more clear.

- Figure 10 is too complicated, especially part A. There are too many symbols. Please revise. Maybe don't use time on the x axis, but same axes as in B and C and denote the time of sampling for each subplot.

  Panel A shows the same data also shown in panel B and C, but the temporal development instead of depth profiles. The depths profiles are best presented and perceived when plotted vertically. Similarly, time is best understood when plotted on the x-axis, and switching the axes in panel A would be counter intuitive. Also, panel A is about the temporal relation between the different samples from water input, pore water, and piezometers. Reducing the information in this panel would render its information content useless. To make the figure easier to read, we changed the headings and note that Panel A shows the same data as shown in panel B and

C. We also mention in the caption that it shows the influence of the water input on pore and piezometer water.

- P. 29 lines 26-31: It remains unclear, which information on form is discussed. For example, I can't find any information about the hydraulic conductivity in this paper. I expect that form characteristics are part of the companion paper, but I think they would make a valuable addition to this paper as well (see my major concern above). Especially when reading the companion paper, I wonder if it would make sense to include the key findings presented in that paper to include in this paper, and make one from the two paper.
As elaborated above, the two papers present more methods and results than appropriate for one paper. The combination of the irrigation data with hydrograph and tracer data yields different insights than the combination of the irrigation data with the plot excavations. We therefore prefer to leave the separation as it is in the current version of the manuscripts. We did, however, include some more information, such as hydraulic conductivity measured in the catchment, in the methods section and refer to the companion paper for more detail. We also link back to the findings on subsurface structures from the companion study in the revised discussion, to improve the link between the two papers.

**Typing errors**

- Page 5, line 32: 'Bulk samples of the ? .' Add a word!

- Page 6, line2: 'after the they' ? Delete 'the'.

- Page 6, line 12: The plot had a slope (Add 'a'.)

- Page 11, line 10: Add a preposition after 'Comparison'.

- Page 13, line 7: miximum?

- Page 27, line3: preferential flow? (Add 'flow'.)

- Page 30, line 13: Add a preposition after 'interpretation'

- Page 31, line3: 'under pergiglacial conditions', insert space.

The typing errors have been corrected. Thank you for pointing them out.

[revised manuscript text omitted]

---

## Author Response (AR3)

**Comments on the third revision of the manuscript 'Form and function in hillslope hydrology: characterization of subsurface flow based on response observations' by L. Angermann et al.**

Once again we'd like to thank the editor and the reviewers for their feedback on our manuscript. We are happy that we could improve our manuscript based on the comments by all reviewers involved in the revision during the last months. Please find below the comment by reviewer#2 and our response in purple. The abstract of the manuscript was revised accordingly and the changes are shown in the annotated version of the manuscript at the end of this document.

**Answers to the comment by reviewer#2**

The authors have responded to the comments raised by the reviewers; they have not considered all of them but then gave reasonable justifications for not doing so. At this point I do not have specific comments anymore. However, I still have the same concern as before. As a stand-alone paper, results and conclusions seem somewhat vague with limited gain of knowledge on hydrologic function. This is actually reflected in the abstract. The authors state that their goal is to ' ... explicitly pursue the question of what information is most advantageous to understand a hydrological system'. This is a very interesting question (although it is a new spin of the study that did not become evident in previous versions of the manuscript) that is unfortunately not picked up again, neither in the conclusions nor in the abstract. There is only one sentence in the abstract that relates to the results of the study: '... The experiment revealed fast establishment of preferential flow paths at the hillslope-scale despite unsaturated conditions ... '. The rest of the abstract is very vague without explicitly stating how the research question was answered and what was learned. Of course, the setup and results of the experimental work cannot be changed anymore, but I would strongly recommend to improve the abstract in this respect and make it more informative and also more concrete even though results may not be very exciting or clear.

Besides our findings on processes, we also conclude in the manuscript, that response observations (patterns and dynamics, i.e. function) were well suited to understand processes at the observational scale and that the combination of a hillslope-centered and stream-centered approach allowed us to understand the establishment of flow paths within the hillslope. We further conclude, that generalizations and transfer of knowledge to other scales than the observational scale(s) require a more detailed knowledge on subsurface structures and refer the reader to the companion paper, where this aspect is investigated and discussed thoroughly.

To mitigate the reviewer's concern about the discrepancy between the story line of the manuscript and the findings presented in the abstract, we elaborated these two aspects of our findings more clearly. In the revised version of the abstract, we emphasized that the fast establishment of preferential flow paths as well as their relation to larger scale discharge response are the main findings of our study on flow processes at the hillslope. Afterwards, we also emphasized our

conclusions on the applied methods and the *form and function* approach. We shortly described what we could learn from the response observations (i.e. functional approach) and which aspects remained vague. We hope that the revised abstract eliminates the concerns raised by reviewer#2 and better represents the story line of the manuscript.

[revised manuscript text omitted]